# Flexible, scalable, high channel count stereo-electrode for recording in the human brain

Keundong Lee[1,10], Angelique C. Paulk[2,3,10], Yun Goo Ro[1,10], Daniel R. Cleary[1,4], Karen J. Tonsfeldt[1,5], Yoav Kfir[6,7], John S. Pezaris[6,7], Youngbin Tchoe[1], Jihwan Lee[1], Andrew M. Bourhis[1], Ritwik Vatsyayan[1], Joel R. Martin[1], Samantha M. Russman[1], Jimmy C. Yang[6,7], Amy Baohan[6,7], R. Mark Richardson[6,7], Ziv M. Williams[6,7], Shelley I. Fried[6,7], U. Hoi Sang[1], Ahmed M. Raslan[8], Sharona Ben-Haim[4], Eric Halgren[9], Sydney S. Cash[2,3] & Shadi. A. Dayeh[1] ✉

Over the past decade, stereotactically placed electrodes have become the gold standard for deep brain recording and stimulation for a wide variety of neurological and psychiatric diseases. Current electrodes, however, are limited in their spatial resolution and ability to record from small populations of neurons, let alone individual neurons. Here, we report on an innovative, customizable, monolithically integrated human-grade flexible depth electrode capable of recording from up to 128 channels and able to record at a depth of 10 cm in brain tissue. This thin, stylet-guided depth electrode is capable of recording local field potentials and single unit neuronal activity (action potentials), validated across species. This device represents an advance in manufacturing and design approaches which extends the capabilities of a mainstay technology in clinical neurology.

Brain disorders severely interfere with quality of life and can lead to major socioeconomic disparities[1,2]. A major therapeutic approach for a wide variety of neuropsychiatric diseases involves invasive recordings from both the cortex and subcortical structures and/or direct electrical neuromodulation of those structures. For treating medically intractable epilepsy, for example, it is commonplace for recordings to be made using stereotactically placed electrodes (sEEG or depth electrodes). Similarly, electrodes of this type are used to target the thalamus, substantia nigra and other subcortical structures for the control of seizures, Parkinson's disease, and essential tremor as well as a growing number of other disorders[3–13]. Future applications of these electrodes could be to understand memory disorders and assist in memory restoration[14–16] while other uses could be the development of brain computer interfaces to restore movement and communication in the setting of trauma, amyotrophic lateral sclerosis and stroke[17]. Electrodes of this type are implanted through small openings in the skull and penetrate the brain parenchyma at varying depths depending on the surgical target, and allow for subcortical recordings and, sulcal depth evaluation, with deep structural reach that is not attainable by surface electrodes. Currently, arrays of electrodes are hand-assembled 0.8–1.27 mm diameter cylinders comprised of 8–16 contacts each 1.5–5 mm in length.

[1]Integrated Electronics and Biointerfaces Laboratory, Department of Electrical and Computer Engineering, University of California San Diego, La Jolla, CA 92093, USA. [2]Department of Neurology, Harvard Medical School, Boston, MA 02114, USA. [3]Center for Neurotechnology and Neurorecovery, Department of Neurology, Massachusetts General Hospital, Boston, MA 02114, USA. [4]Department of Neurological Surgery, University of California San Diego, La Jolla, CA 92093, USA. [5]Department of Obstetrics, Gynecology, and Reproductive Sciences, Center for Reproductive Science and Medicine, University of California San Diego, La Jolla, CA 92093, USA. [6]Department of Neurosurgery, Harvard Medical School, Boston, MA 02114, USA. [7]Department of Neurosurgery, Massachusetts General Hospital, Boston, MA 02114, USA. [8]Department of Neurological Surgery, Oregon Health and Science University, Portland, OR 97239, USA. [9]Department of Radiology, University of California San Diego, La Jolla, CA 92093, USA. [10]These authors contributed equally: Keundong Lee, Angelique C. Paulk, Yun Goo Ro. ✉e-mail: sdayeh@ucsd.edu

The manufacture of clinical electrode arrays has only incrementally advanced since their initial development in the early 1950s because of the limitations in hand assembly and wiring of these implantable devices. In addition, the construction of these electrodes limits their spatial resolution; they are only able to record local field potentials (LFPs) over relatively large areas (e.g., multiple mm) and are unable to record from small, discrete neuronal populations, let alone individual neurons (e.g., action potential activity). A variety of modifications of this electrode have been used to record highly local sites in the brain. For example, platinum-iridium microwires extruding from the tip of depth electrodes enable recording of single and multi-unit activity from up to 9 microwires[18]. This configuration only allows recording from the tip. Dixi Medical has produced a depth electrode with extensible microwires from the body of the array (personal communication with Dixi Medical). Neither of these approaches allows more than a few channels to be recorded, neither afford grid-like high spatial resolution in that developing a spatial map of multiple action potential sites of origin is not possible, and the devices are still hand-made. Other electrodes that can record single units from the human brain and afford high-resolution spatial mapping of single-cell activity include the Utah array[19] and Neuropixels[20,21] with up to hundreds of channels[22,23]. These devices, currently used in research, are limited by the silicon (Si) manufacturing technology and the brittleness of Si. They are also currently only able to access superficial cortical layers of the brain in humans, though there are advances in these devices enabling recording from deeper structures used in non-human primates[24].

To increase the spatial resolution and channel count of electrodes that can record from either the lateral gray matter or deep brain structures, recent engineering approaches have focused on rolling or adhering conformable and photolithographically defined polyimide electrodes around or on medical-grade tubing used in clinical depth electrodes[25–28]. Previous research has well-established the transformation of thin-film electrodes to depth electrodes and demonstrated successful high-quality recording chronically[27]. However, these hybrid integration approaches impose a limitation on the size of the electrode such that the starting diameter is pre-determined by the clinical depth electrode diameter.

To address these various limitations and go well beyond current capabilities, we developed an entirely different manufacturing method for thin-film electrodes enabling reproducible, customizable, and high throughput production of electrodes (1) to be implanted in the operating room using similar brain implant techniques to standard clinical depth electrodes, and (2) to reach deep brain structures and achieve high spatial resolution and channel count with a much thinner electrode body. This advanced manufacturing process exploits (1) titanium (Ti) sacrificial layers employed in the microfabrication of free-standing microelectromechanical system (MEMS) devices. A stylet inserted where the Ti sacrificial layer is removed assists in hardening and implanting the depth electrode—similar to the standard clinical SEEG electrode implantation procedures—and is subsequently removed. (2) This MEMS process is implemented on relatively large ($18 \times 18$ cm²) glass substrates (Fig. 1a) allowing us to produce multiple copies of the SEEG devices using materials that are typical for the manufacturing of display screens. Therefore, this unique manufacturing method of thin-film based and clinical-grade depth microelectrode array, termed a micro-stereo-electro-encephalography (µSEEG) electrode, enables flexibility in design, scalability afforded by the display screen manufacturing, which is cost-effective, and does not involve manual assembly typical for standard SEEG electrodes. The µSEEG dimensions can be made custom for application-based contact spacing and channel count. Here, we illustrate the flexibility of our design by manufacturing and testing µSEEG electrodes ranging from a few millimeters to tens of centimeters long, 1.2 mm wide, and 15 µm thick. The manufacturing is compatible with electrode

materials that can be used to produce microscale electrode contacts with low electrochemical impedance. We demonstrate the µSEEGs with two low-impedance contact materials: (1) the platinum nanorod (PtNR) contact technology (Fig. 1b) we developed[29,30] and (2) the poly(3,4-ethylenedioxythiophene) polystyrene sulfonate (PEDOT:PSS) electrode technology[31–34] to record broadband neuronal activity including single units (action potentials) and LFPs in rats, pigs, non-human primates (NHPs), and humans. We also test and demonstrate the flexibility of the manufacturing process, which can involve either polyimide or parylene C as the device substrate, both of which are biocompatible. This newly integrated µSEEG electrode induced less tissue damage than cylindrical clinical electrodes in a 2-week rat implant ($n = 1$). Such a flexible, high channel count system paves the way for expanded and more efficacious neuronal recordings and neuromodulation across the spectrum of neuropsychiatric diseases.

## Results

### Manufacturing µSEEG electrodes

To fabricate µSEEG electrodes, we first coated the glass substrate with a sacrificial polyimide layer and followed by the deposition of titanium etch-mask layer that was patterned with circular openings (Supplementary Fig. 1). The circular openings were used to etch the 1st PI layer that were deposited on top of this layer. At the end of the fabrication process, the entire device layer was flipped, and O₂ plasma was used to create an array of holes in the 1st PI layer. During this process, the titanium etch-mask layer protected the first polyimide layer, except for the circular openings. We then coated the glass substrate with two polyimide layers (1st and 2nd PI layer) and an interleaved Ti sacrificial layer (Fig. 1c). When the sacrificial titanium layer is dissolved in a later stage in the process, the two polyimide layers form the structural enclosure (sheath) for the insertion of the stainless-steel stylet. Above the second polyimide layer, we deposited and patterned the metal trace layer with 10 nm/250 nm chromium/gold stack (deposited and patterned twice for a total trace thickness of 520 nm). Both the width and spacing of the metal traces are 3 µm. As it approaches the connectorization where the PCB is attached, the width of the metal traces expands to 20 µm and its spacing becomes 5 µm. After metallization, a film of platinum–silver alloy was deposited to form the PtNR contacts. The µSEEG electrode has advantages in its form factors including channel count, contact size, and impedance over the previously reported electrodes due to the advanced MEMS technique employed in the fabrication process (Supplementary Table 1)[25–28]. This was followed by a top-most polyimide layer (3rd PI layer, Fig. 1c) coating. The next step involved exposing the microcontacts and defining the electrodes. To achieve this, we induced holes in the 3rd PI layer to expose the platinum–silver alloy films. Simultaneously, we etched the shape of the electrode and additional larger holes (Fig. 1d) into the polyimide layer for mechanical stabilization of the stylet. A nitric acid (HNO₃) etch at 60 °C dissolved the silver from the platinum–silver alloy and exposed the PtNR contacts (Supplementary Fig. 1). The resulting structure is then peeled off from the substrate, flipped, and temporarily adhered to another host glass substrate. At this point, the very first sacrificial polyimide layer was etched by O₂ plasma exposing the titanium etch-mask layer that was pre-patterned with circular openings. Continuation of the O₂ plasma etching through the circular openings drilled through the 1st PI layer, which exposed the titanium sacrificial layer underneath. A final buffered oxide etching dissolved the titanium layers (both sacrificial and etch-mask layer) after which the device is rinsed with flowing deionized water.

The stylet is inserted through the mechanical stabilization holes (Fig. 1d) and the sheath formed by the two polyimide layers (Fig. 1e) to the tip of the electrode (Fig. 1f, g; stylet insertion process illustrated in Supplementary Fig. 2). At the very tip of the electrode, an array of holes was etched in the 1st PI layer around the sacrificial layer (marked with a red arrow in Fig. 1h). As the 2nd PI layer is coated to fill these holes, the

interface between the 1st and 2nd PI layers has effectively a larger surface area than a planar one and as a result, better adhesion between the 1st PI layer and the 2nd PI layer is established. The greater mechanical stability afforded by the array of holes prevents the stylet from piercing through the tip when the stylet reaches this interface (Supplementary Fig. 3). The tip of the stylet is mechanically polished to a rounded shape to minimize damage during insertion (Supplementary Fig. 4).

The µSEEG electrode was manufactured with a U-shaped neck between the electrode array proper and a continuation of the thin film, providing additional length for the metal traces. Once the straight edges of the U-shaped electrode are flipped, the total length of the µSEEG electrode becomes 28 cm, on par with the length of a standard clinical sEEG electrode (Fig. 1j) but with a total thickness of approximately 15 µm. Overall, the µSEEG electrode after the stylet insertion had ~1/10 the cross-sectional area of a typical clinical depth electrode while matching its length and its ability to reach to deep brain structures (Fig. 1j).

### µSEEG electrodes are robust to tearing and can be implanted and extracted without deformation, producing less damage than clinical electrodes

As these devices must be robust for longer-term implant periods, mechanical strength and resilience against tear were assessed using pull measurements with both the µSEEG electrode and, to compare with a clinical lead, on a 1.2-mm diameter PMT depth electrode anchored on two polyurethane tube regions around a Pt contact. The tensile strengths (critical forces) were 1 MPa (16 mN) for the µSEEG electrode and 14 kPa (48 mN) for the PMT electrode (Supplementary Fig. 5). To evaluate the reliability of the µSEEG electrodes for longer-term implant periods, we performed an accelerated aging test and a bending test showing negligible degradation over 150 days with 84,000 cycles of lead bending (Supplementary Fig. 6).

Since electrode and contact integrity should also be maintained during implantation of the µSEEG with the stylet without any deformation or loss of function, an acute implantation was first assessed on a phantom brain model. The displacement of a µSEEG and the surrounding phantom brain medium before and after stylet extraction was less than 10 µm (Supplementary Fig. 7, $N = 6$). Electrochemical impedance spectroscopy before and after stylet insertion showed relatively stable 1-kHz impedances, changing from $33.0 \pm 2.5 \text{ k}\Omega$ to $35.0 \pm 3.7 \text{ k}\Omega$ (Supplementary Fig. 8), indicating that there was no substantial damage to the device during stylet insertion. The electrodes were extracted in these phantom experiments and all animal and human experiments without any mechanical deformations or tears.

Finally, to test the amount of tissue damage caused by these devices, we implanted rats with one chronic µSEEG electrode with 1.89 mm recording length on one hemisphere and a clinical electrode on the other hemisphere for 14 days ($N = 4$ electrodes). Insertion of the µSEEG electrode resulted in decreased astrocyte scarring, as measured by significantly lower GFAP positive area as compared to the clinical electrode as shown in Supplementary Fig. 10. We observed no significant difference in the number of Neun-positive cells surrounding the lesion between the µSEEG and the clinical grid, but there was a small non-significant improvement with the µSEEG electrode (2-way ANOVA, F (1, 72) = 3.290, $p = 0.0739$). We also imaged the PtNR µSEEG electrodes upon extraction from the NHP brain and observed minimal changes compared to non-implanted ones, demonstrating the stability of the µSEEG electrode in tissue (Supplementary Fig. 11, $N = 3$).

### µSEEG flexible design is scalable for multiple acute and chronic applications

To demonstrate the flexibility in the manufacture, design, and use of µSEEG to record neurophysiologically relevant neural activity in multiple settings and species, we tested devices with working neural recording lengths ranging from 1.89 to 7.65 mm, made from either parylene C or polyimide, with microelectrode contacts composed of either PEDOT:PSS or PtNRs (Fig. 1k, Supplementary Fig. 12, and Supplementary Table 3). We transitioned to all polyimide µSEEG electrodes after we observed that parylene C µSEEG develop cracks in the parylene C layers and in the PEDOT:PSS layers after stylet insertion, whereas polyimide µSEEG did not suffer from any cracks. The crack was caused by mechanical damage to the parylene C layers applied by the stylet during insertion, which then propagated to the PEDOT:PSS layer (Supplementary Fig. 9). In addition, PtNRs contacts did not suffer any delamination from the µSEEG whereas PEDOT:PSS suffered from delamination after stylet insertion in a substantial subset of electrodes, therefore reducing product yield.

All designs used have microelectrode contacts (also called channels, each 30 µm contact diameter for PtNRs and 20 µm contact diameter for PEDOT:PSS) with a center-to-center spacing of 60 µm (Fig. 1k, Supplementary Fig. 12, and Supplementary Table 3). While the diameter of the microcontacts can be adjusted, we have found that a contact diameter of 30 µm provides the most reproducible and optimal results for PtNRs based on our most recent optimization efforts. Therefore, we decided to use 30 µm as a diameter for the PtNR electrodes. We created two short versions: (1) a short 64-channel µSEEG; (2) a short 32-channel µSEEG. The short 64-channel µSEEG includes 64 microelectrode contacts along a recording length of 3.80 mm. Side flaps are incorporated to help with the stabilization of the array upon insertion. This design is intended for use in the intraoperative setting and resembles other microelectrode arrays (often called laminar arrays), which were designed to capture activity across the cortical layers[35]. (Fig. 1k–n, Supplementary Figs. 7 and 8, and Supplementary Table 3). The architecture of this system is formatted for use in smaller animals or in recording from the neocortex of humans or larger animal species—such as for use in a brain computer interface. The short 32-channel µSEEG (Fig. 1k, l, right; Supplementary Fig. 12 and Supplementary Table 3) includes 32 microelectrode contacts along a recording length of 1.92 mm intended for use in chronic recordings in smaller animals.

We also made a longer version designed for accessing deeper structures (simultaneously with lateral cortex) in larger animals. This long µSEEG includes 128 microelectrode contacts at a 60 µm center-to-center spacing along a recording length of 7.65 mm at the tip of the entire array. This configuration most closely resembles clinical depth electrodes (Fig. 1j, o and Supplementary Fig. 12), although the spacing of the contacts or the incorporation of contacts with diameters larger than 100 µm in a future µSEEG design can be varied for specific end use. Our custom acquisition board connects to a 1024-channel electrophysiology control system, provided by Intan Technologies LLC[29]. Depending on the channel counts of the electrode, we utilized 1–8 RHD2164 chips for impedance measurement and for recordings.

### µSEEG electrodes record local field potential events both acutely and chronically

To test the capabilities of the µSEEG electrode in capturing relevant neural activity[36] we recorded from the rat barrel cortex in both the acute and chronic settings. Acute recordings from rat S1 cortex under anesthesia were performed with both a surface µECoG array[30] and the 64-channel µSEEG (Fig. 2a–c and Supplementary Figs. 12–15). When contralateral whiskers were selectively deflected by a directed air puff stream, we found LFP voltage responses ($z$-scored relative to 0.5 s before stimulus delivery) and increases in high gamma power (HGP; power between 65 and 200 Hz) on both the µECoG and µSEEG (Fig. 2d and Supplementary Figs. 14 and 15). At different depths along the µSEEG electrode and at different channels in the µECoG grid, whisker deflection induced significantly greater LFP and HGP responses than baseline (0.5 s before stimulation; Wilcoxon rank-sum test; $p < 0.001$;

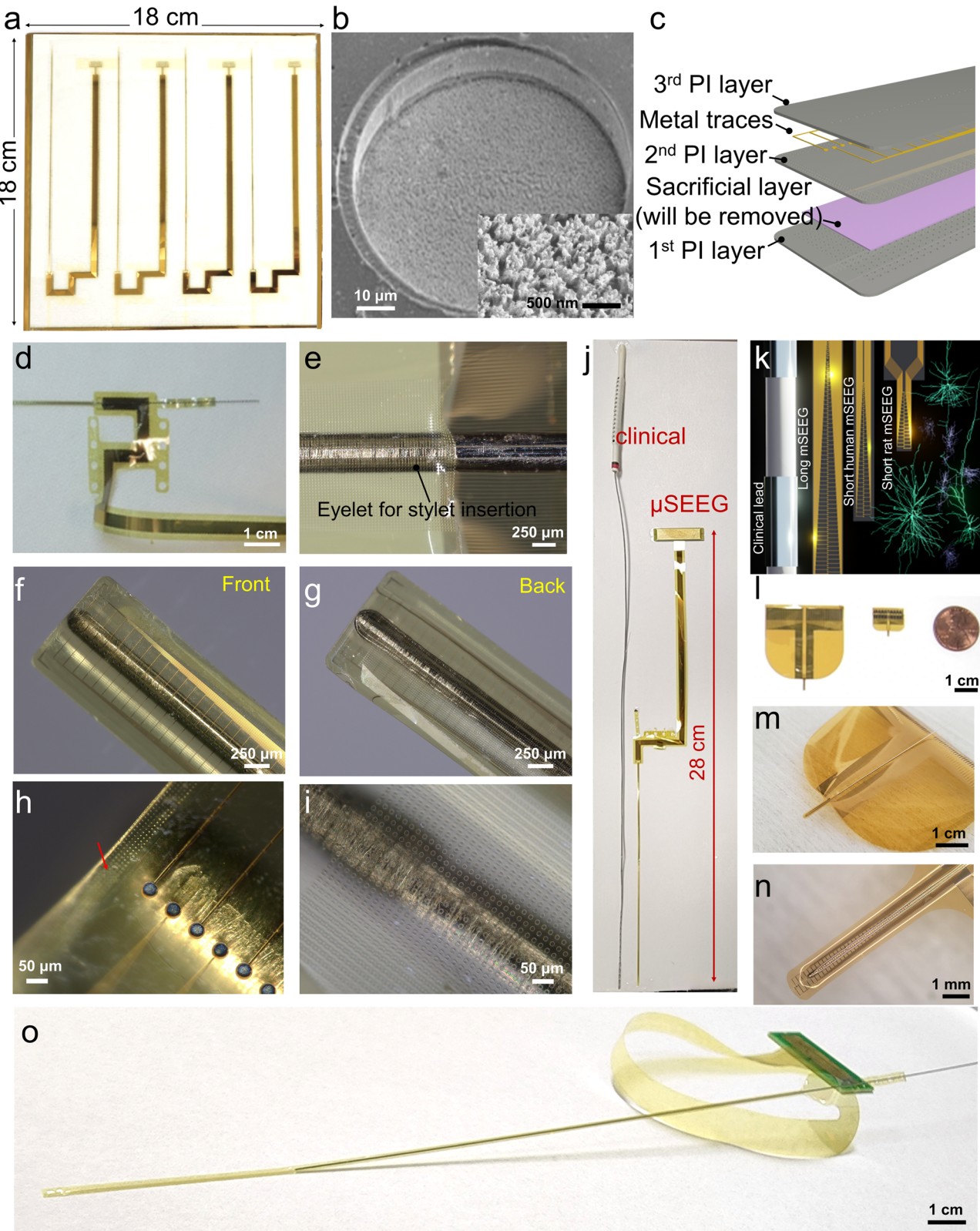

**Fig. 1 | μSEEG electrode arrays. a** Photograph of a single glass substrate plate with four μSEEG electrodes. **b** Scanning electron microscope (SEM) image of a single PtNR contact; Inset is a magnified image showing the PtNRs. **c** Structural composition of the μSEEG array. **d** Photograph showing the 'neck' of the array where the U-shape pattern is flipped to provide metal trace extension and circular holes are present to stabilize the inserted stainless-steel stylet. **e** Optical microscope (OM) image of the region of insertion of the stylet in the inflatable "sheath" of the μSEEG electrode. OM images of **f** front, **g** back side of the μSEEG electrode. **h, i** Magnified OM images at the tip **f** front and **g** back layers. The red arrow indicates the micro-hole arrays that interlock the 1st and 2nd PI layers. **j** Long 128-channel μSEEG electrode and comparison with a clinical electrode. **k** Diagram of the relative scale of human cortical neurons relative to a clinical SEEG lead and μSEEG electrodes[45–47]. **l** Flexibility in manufacturing procedure to produce short 64-channel μSEEG electrodes (left) or short 32-channel μSEEG electrodes (right) and photographs showing **m** overall and **n** tip of the 64-channel μSEEG electrodes. **o** A perspective view of the long μSEEG electrode with partially inserted stylet illustrating the flexibility and slenderness of the electrode body.

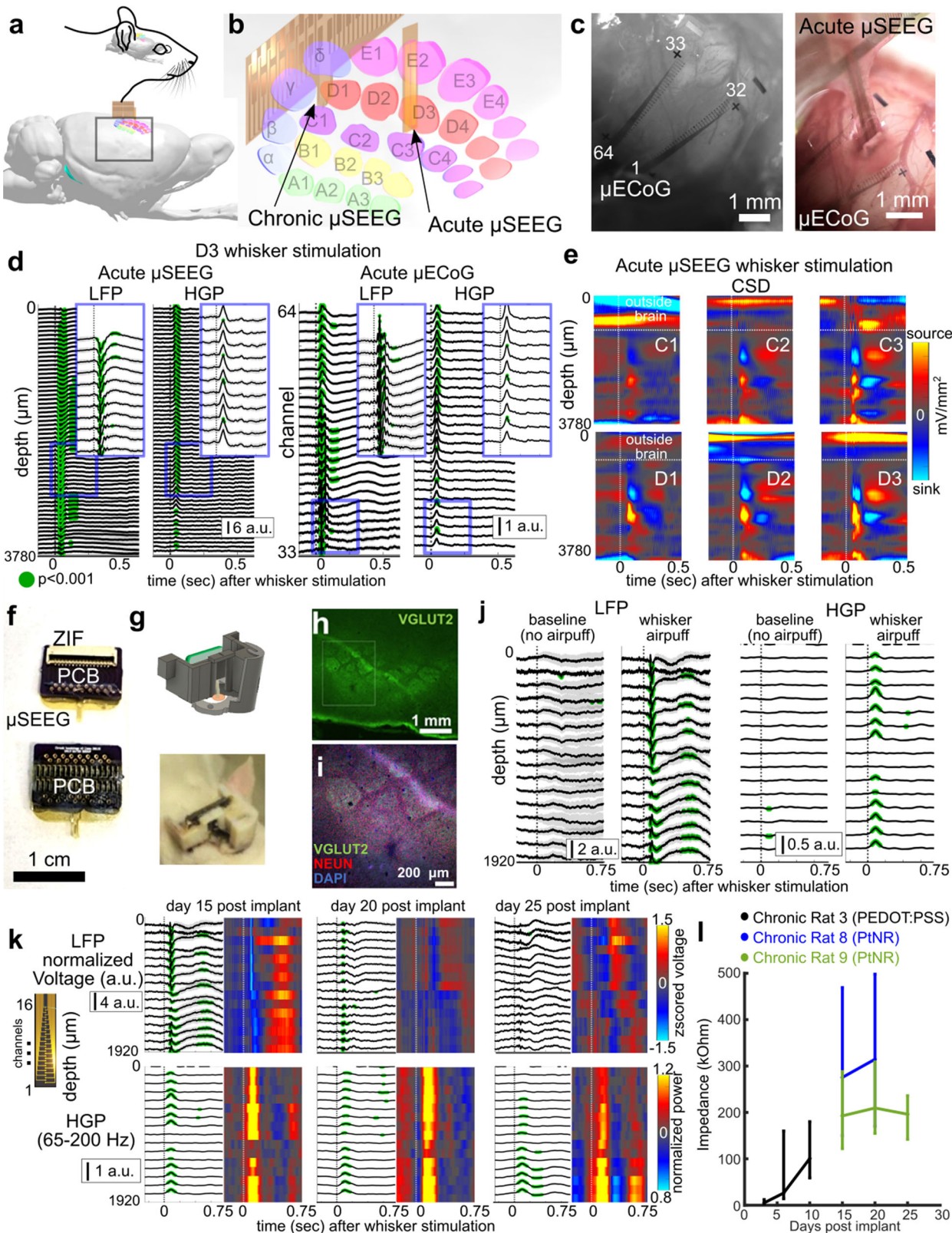

*n* = 39 trials), with some deflections showing reversals in voltages along the depth electrode, also reflected in the current source density (CSD) analysis (Fig. 2e and Supplementary Figs. 16 and 17). Furthermore, we found sensory specificity in the responses, with stronger neural responses (in the LFP, HGP and CSD) with stimulation closer to D3, C3, and even E3 (Fig. 2e and Supplementary Figs. 14–18), allowing us to estimate the location of the electrodes relative to columns of the barrel cortex. The concurrently implanted µECoG surface microelectrode, used to confirm we were recording from the barrel cortex, reflected similar D3, C3, and even E3 whisker-selective voltage and HGP dynamics in response to sensory stimulation (Supplementary Figs. 19–21).

After confirming that µSEEG electrodes could detect sensory stimulation-induced neural activity acutely, we developed a 3D-printed

**Fig. 2 | µSEEG electrodes can be used for acute and chronic implantations and recordings. a**, **b** Location and 3D reconstruction of possible locations of the acute and chronic implantation of µSEEG electrode devices for recording from the rat barrel cortex[48]. **c** Images of the implanted µECoG electrode (left) and the µSEEG electrode (right). Note some contacts are outside brain tissue on the µSEEG electrode. **d** Example voltage responses across the µSEEG electrode (left) and the µECoG electrode (right) to whisker stimulation at different whisker locations, with the insets zoomed-in views of the voltages and high gamma power (HGP, 65–200 Hz). Green dots indicate a significantly different from 0.5 s before (which is baseline) air puff stimulation to the whisker (Wilcoxon rank-sum test) per channel and across trials. Number of trials >10. **e** Increasing responses as air puff stimulation

is closer to the C3 and D3 whiskers as indicated by the current source density (CSD). **f** 32-channel µSEEG electrode for chronic rat recordings and a custom printed circuit board (PCB) with zero-insertion-force (ZIF) connector which electrically connects the device to the recording system via flexible flat cable (FFC). **g** 3D-printed headstage for the 32-channel µSEEG electrode. **h**–**i** Electrode location localization as visualized using histology. **j** Example voltage and high gamma power (65–200 Hz) responses without stimuli (baseline) versus with stimuli (whisker air puff). **k** Responses to air puffs at three different time points post implant in one rat. **l** Impedance measures at 1 kHz across multiple days and multiple rats; vertical bars are standard deviation from average values. For (**d**), (**j**), and (**k**), a.u. arbitrary units in z-scored voltage for the LFP and normalized High Gamma Power (HGP).

headstage for a chronic implantable version of the short 32-channel µSEEG electrode (Fig. 2f, g and Supplementary Fig. 22). We implanted the device successfully in nine rats (Supplementary Table 2) with rat barrel cortical responses in three of the nine rats with histological confirmation of the electrode location (Fig. 2g–k). We implanted the devices for 25 days and recorded at three or more time points following implantation to test recording quality and impedance changes over time (Fig. 2f–l). Impedance fluctuated across days per rat but was still low enough to record voltage responses ($63.5 \pm 49.1$ kΩ, $268.0 \pm 115.2$ kΩ, and $198.0 \pm 48.7$ kΩ for rat 3, 8, and 9, respectively) across rats and across days post implant (Fig. 2l). We recorded voltage responses and changes in high gamma power with whisker stimulation, which was not evident when performing sham controls (trials with no air puffs; Fig. 2j). Furthermore, we observed similar voltage responses and HGP recorded by functional microcontacts across the days in individual rats (Fig. 2k). This result was confirmed by calculating the correlation between averaged responses across days per channel. In particular, we found that activity during whisker stimulation was more correlated per channel across days (Pearson's linear correlation average rho across channels: $0.2 \pm 0.06$ (std), maximum average: $0.73 \pm 0.10$ (std)) compared to sham controls (Pearson's linear correlation average rho across days per channel: $0.17 \pm 0.03$ (std), maximum average: $0.56 \pm 0.13$ (std)). These differences were also reflected in the high gamma power measures (Pearson's linear correlation average rho across days per channel: whisker stimulation: $0.33 \pm 0.28$ (std), maximum average: $0.70 \pm 0.15$ (std); sham controls: $0.21 \pm 0.07$ (std), maximum average: $0.54 \pm 0.04$ (std)).

### µSEEG electrodes acutely record stimulus and anesthesia-induced dynamics across species

Demonstrating that µSEEG electrodes can be used to record clinically relevant dynamics in the human brain requires both scaling up the devices for use in recording from larger brains as well as demonstrating that µSEEG electrodes record clinically and neurologically relevant neural dynamics[37,38]. A major goal was to test the µSEEG while modeling settings and paradigms that could be used in acute or chronic clinical mapping of activity[6,39]. Therefore, we recorded neural activity using the short 64-channel µSEEG in three different settings: (1) from the somatosensory cortex in an anesthetized pig, (2) in an anesthetized NHP in the operating room, and (3) in the operating room from human cortex preceding tumor resection (Fig. 3, Supplementary Fig. 12, and Supplementary Table 2).

Intraoperative clinical mapping often involves the use of stimulation to delineate functional (eloquent) tissue and connectivity. To model this paradigm, we stimulated the pig spinal cord with a bipolar stimulator and recorded with the µSEEG in the pig cortex to map responsiveness and connectivity. We recorded changes in neural activity across cortical layers in the somatosensory cortex induced by direct electrical stimulation in the spine using the short 64-channel µSEEG electrode, stimulating with currents ranging from 200 to 6000 µA (Fig. 3a–c). We found significantly increasing voltage responses with increasing injected current per channel ($p < 0.001$;

Kruskal–Wallis test; Fig. 3a and Supplementary Fig. 23). The response waveforms varied between the different contacts, including a field reversal approximately in the middle of the implanted electrode. This field reversal was most obvious at stimulation currents >800 µA (green dots, Fig. 3b, significantly above a baseline taken 0.5 s before stimulation, Wilcoxon rank-sum test). When we plotted the largest absolute voltage deflections from baseline, we found a clear division in the voltage between the deeper contacts and the superficial contacts. This high-resolution laminar distribution of the responses was also reflected in differences in oscillatory power across the cortical layers. We found gamma (30–55 Hz; $p = 0.0056$ for 6000 µA) power in the more superficial contacts (Kruskal–Wallis multiple comparisons test; Fig. 3c and Supplementary Fig. 23). We also found response timing differences, with the time to voltage peak and HGP peak shorter in middle and superficial contacts (resulting in a Pearson's linear correlation between peak timing and channel number: voltage- rho = $-0.11$; $p < 0.0001$; HGP- rho = $-0.04$; $p = 0.03$) with the trend reversed with the peak beta power (beta power- rho = $0.04$; $p = 0.03$; Fig. 3c and Supplementary Fig. 23). In other words, the µSEEG electrode recorded stimulation-induced activity with cortical layer-specificity at a spatial resolution (60 µm contact to contact pitch) not possible with current clinical leads (resolution on the scale of millimeters; Supplementary Fig. 12).

In a second intraoperative paradigm, we recorded neural activity across cortical layers in the visual cortex of an anesthetized NHP using the short 64-channel µSEEG electrode. We found ongoing anesthesia-related burst suppression, which could be detected using automatic approaches along the depth electrode[40] (Fig. 3d and Supplementary Fig. 24). As shown in previous preparations and other species, we found a gradient of activity across the array, with more detected bursts early in the recording and even relative to suppression epochs (as represented by the burst-suppression ratio) in more superficial contacts[40] (Supplementary Fig. 24). This resulted in high negative correlation values between electrode depth into the tissue and detected bursts over 300 s ($r = -0.73$; $p = 0.0021$; Pearson's linear correlation; Supplementary Fig. 24).

Finally, in a true test of the translational feasibility of the µSEEG, we acutely implanted short 64-channel µSEEG electrodes in the left middle temporal gyrus in two separate human patient participants (Figs. 1f and 3e–g and Supplementary Video 1) undergoing temporal lobe resection for clinical reasons. With each participant, we inserted a single 64-channel short µSEEG devices into the tissue, which the clinical team determined would be resected. The recordings were brief (10 min) yet we were able to record ongoing spontaneous activity. In one case, the participant was under general anesthesia and there was clear evidence of anesthesia-induced burst suppression in the recordings (HS1), also detected through an automatic algorithm[40] (Fig. 3e). Like in the NHP, the number of detected bursts was increased in more superficial contacts, resulting in a correlation between depth (into the tissue) and burst detections of $r = -0.5927$ ($p = 0.0023$; Pearson's linear correlation, over 300 s of recording). Notably, we also found the thin film component of the electrode device, once implanted, would move with the brain tissue movement, indicating the device

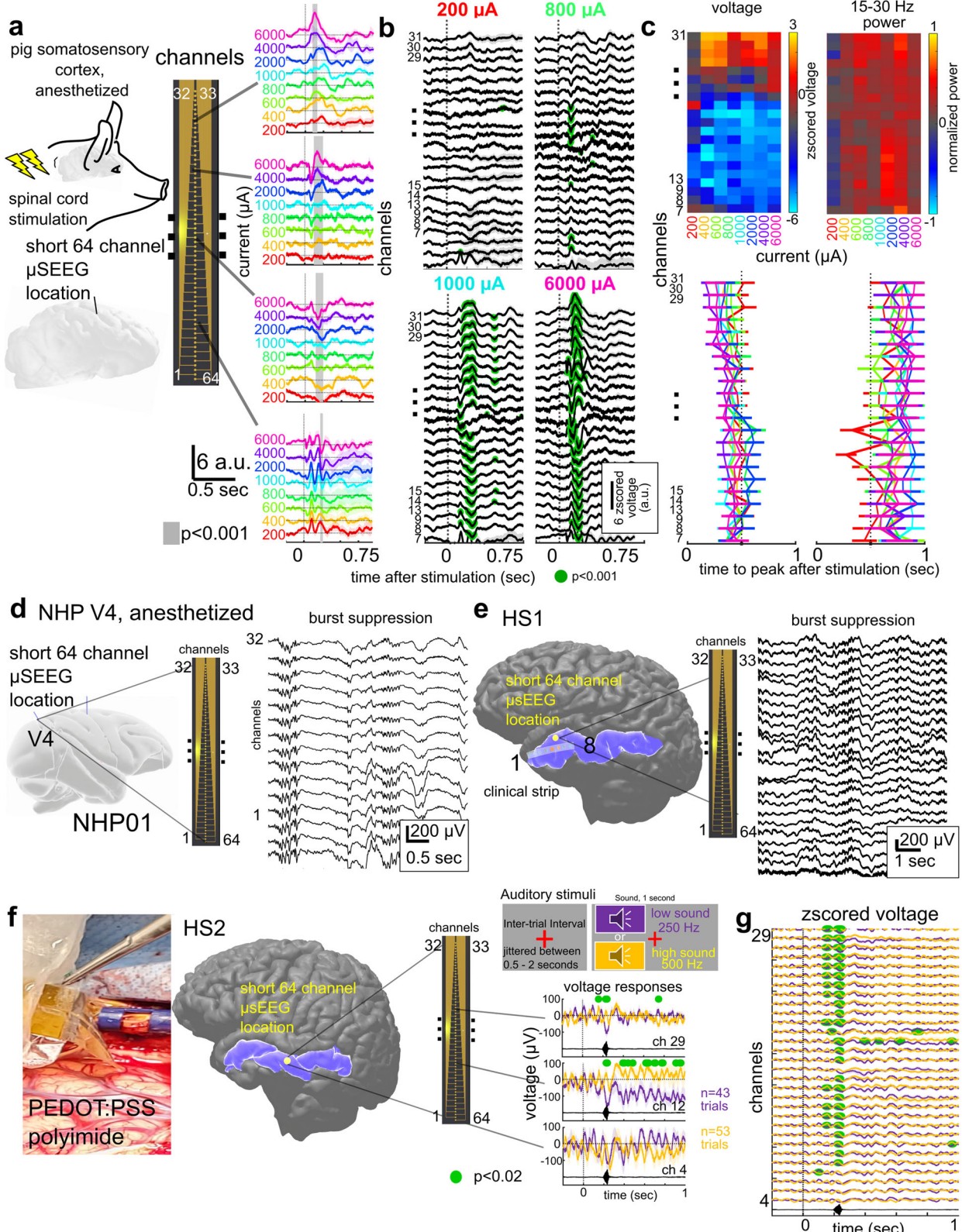

is light enough to move with the recording medium (Supplementary Video 1).

In the second recording, the participant was awake under monitored anesthesia care (MAC) and listened to low and high auditory cues (HS2; see Materials and Methods; Fig. 3f, g). The neural responses were significantly different between low and high tones in the z-scored voltage values and in HGP at the onset of the sound ($p < 0.02$,

corrected for multiple comparisons; Wilcoxon rank-sum test). Furthermore, there were more significant differences in the responses between low and high sounds in superficial array contacts (Fig. 3g).

**μSEEG electrodes detect single-unit cortical activity**

A key purpose of the μSEEG electrode is to offer advantages over current clinical depth electrodes including increased spatial resolution

**Fig. 3 | Stimulated neural activity and ongoing clinically relevant neural dynamics can be recorded acutely using the short 64-channel µSEEG electrode. a** Direct electrical stimulation of the spinal cord during acute short µSEEG recording from the pig cortex. Gray bar indicates significantly different between current steps, Wilcoxon rank-sum test. **b** Voltage responses along the electrode depth with more responses significantly different to baseline (0.5 s before stimulation) occurring more with higher current levels (green dots, Wilcoxon rank-sum test). **c** Top: two-dimensional heat map of the largest voltage deflection from baseline per channel and per current step (left) and the maximum peak in beta (15–30 Hz) power oscillations (right). Bottom: time to peak voltage (left) and peak beta band power (right) after stimulation per channel and current step. Gray dotted line indicating 0.5 s after stimulation. **d** Acute implant of the short µSEEG electrode into V4 in an anesthetized NHP and ongoing evidence of burst suppression. **e**–**g** Acute implantation of short µSEEG electrodes into left anterior temporal lobe middle temporal gyrus (highlighted in blue) to be resected in the course of clinical treatment in two participants, HS1 and HS2 with a photograph of the implant, a three-dimensional reconstruction of each participants' brain and the relative location of the µSEEG electrode (yellow dot) with a zoomed in inset view of the 64 channels as implanted. **e** Spontaneous ongoing activity with burst suppression along the electrode depth. **f** Auditory responses to low and high tones presented at random in sequence with varying jitter times while recording activity in the lateral temporal lobe. Green dots indicating $p < 0.02$ significant difference between low and high tones, Wilcoxon rank-sum test. **g** Differences in the responses varied across the depth of the electrode. $Z$-scored voltage responses at multiple channels at different depths averaged across trials. Green dots indicating $p < 0.02$ significant difference between low and high tones, Wilcoxon rank-sum test. For (**a**) and (**b**), a.u. arbitrary units in $z$-scored voltage for the LFP and normalized High Gamma Power (HGP).

as well as increased neural resolution. To test whether the µSEEG device can record neural activity at multiple depths in the brain closer to the scale of the human brain, we designed and built the long µSEEG electrode (Figs. 1 and 4 and Supplementary Fig. 12). The 128-channel long µSEEG electrode was built to most resemble clinical depth electrodes with a working recording length of 7.65 mm at the tip of the electrode that is, overall, 28 cm long, 1.2 mm wide, and 15 µm thick which would allow insertion and recording from deeper brain structures. The electrode contacts in this design are concentrated at the tip of the device with inter-contact distances of 60 µm. To test if we could record single neuron activity at depth, we recorded neural dynamics in an NHP that was awake but resting and viewing flickering light-emitting diodes (LEDs)[29] to test for visual responses. The long µSEEG electrode was held by a microelectrode microdrive (see Materials and Methods; Supplementary Fig. 25) to drive the microelectrode to multiple depths from the surface of the cortex within an implanted chamber (Fig. 4a). Along the trajectory moving toward the thalamus, we stopped and recorded at three different depths to examine spiking activity in the cortex as well as in white matter (Fig. 4b). At depths 1 and 2, we found we could record spikes which clustered into single-unit and multi-unit activity (MUA) (depth 1: 1 MUA cluster, 4 single-unit clusters; depth 2: 5 MUA clusters, 31 single-unit clusters; Fig. 4c–e), which we determined by examining the autocorrelation of the spike times and the waveform consistently through time using Kilosort[41]. We did not find any identifiable single-unit activity at depth 3, we were likely mostly in white matter at that depth (depth 3: 13 MUA clusters; Fig. 4f, g). The units and MUA clusters were distributed at different distances and locations along the 128 contacts of the long µSEEG with a range of spike rates, most of which were around 2 Hz (Fig. 4h). Finally, we found the waveform measures show that the units sampled at depths 1 and 2 were clustered in amplitude, the peak-trough ratio, and spike duration measures compared to the MUA clusters found at depth 3 (Fig. 4i). In other words, these clusters are more likely single-unit activity or putative neurons since they were detected while the recording contacts were in cortex but not while in white matter.

## Discussion

We developed a µSEEG electrode that is implanted with a stylet inserter similar to clinical SEEG electrodes, but can be tailored in its range of depth to sample cortical and or deep structures in the brain (or both). This advanced µSEEG electrode can record broadband spontaneous and evoked neurophysiological activity including LFP, CSD and single/multi-unit activity across a variety of species including humans and across entire depths of the brain. While the µSEEG construction is robust, it induced less apparent tissue damage than clinical SEEG electrodes. The layout, shape, and size of the µSEEG electrode could be generated with customizable designs (Fig. 1k–n) by leveraging established display screen fabrication techniques on large glass wafers. Fabrication on glass wafers also promises excellent scalability. Glass panels used in the manufacture of displays use plates a few

square meters in area indicating that large number of arrays can be manufactured even if the arrays are long. Furthermore, the high resolution of lithographic capability can achieve 1.2 µm for both metal line and space (L/S) of flat panel displays[42]. Therefore, the number of contacts can be increased well beyond 128 channels presented here importantly afforded at a much lower manufacturing cost than clinical and other research depth electrodes. A 240 sq. in. monitor has a retail price of nearly $100 with active transistors and light-emitting diodes. The same area can be used to manufacture at least 20 µSEEG electrodes, pointing to significantly lower costs than current clinical SEEG electrodes (>$1000 per electrode) when manufactured at scale. This cheaper advanced manufacturing approach and the added spatial resolution and sensitivity to cellular activity in a smaller form factor can advance our ability to study and treat the human brain and will help broaden the access of the technology to underserved communities and other brain diseases.

One potential limitation of these designs is cross-talk among the channels. While we have not definitively quantified cross-talk in the recordings, we observed a strong common-mode signal on all contacts that we subtracted in order to delineate CSD dynamics. One of the possible reasons for the common-mode signal would be recording the same neural activities, given the narrow spacing of the electrodes (60 µm). In addition, we observed that the location of the reference contact also affects the amount of common signal captured. If inter-channel cross-talk is a substantial issue, future designs could involve distributing the metal traces in separate polymer layers. Another limitation concerns connectorization. Current connectors do not match typical clinical standards. Improving the back-end of the devices is an area of active development. Furthermore, the current design includes contacts facing only one direction along the electrode length. Future designs can involve developing multiple directional contact sampling. Finally, additional optimizations regarding the stylet diameter and the placement of the microcontacts could enhance the capabilities of single-unit recording even after the stylet is retracted. This would prevent the electrode from deflating, ensuring that the microcontacts remain in close proximity to the neural tissues. This can be achieved either by reducing the diameter of the stylet or by positioning the microcontacts along the edge of the electrode.

Nevertheless, advantages include the size and shape of the electrodes as well as the capabilities of the devices. For instance, the width of the µSEEG was still destructive to brain tissue, unlike ultraflexible nanoelectronic probes[43]. However, a human-grade electrode that can reach 10 cm deep into the brain with 128 contacts necessitated the stylet-guided µSEEG design, especially with stimulation considerations where the metal traces need to be sufficiently wide to reduce serial resistances and associated potential drops that compromise the long-term electrode stability. Lastly, the µSEEG can also offer stimulation with favorable stimulation characteristics with PtNRs compared to clinical SEEG electrodes. We prepared PtNR electrode contacts with 1 mm diameter to test how PtNR compares with clinical electrodes in

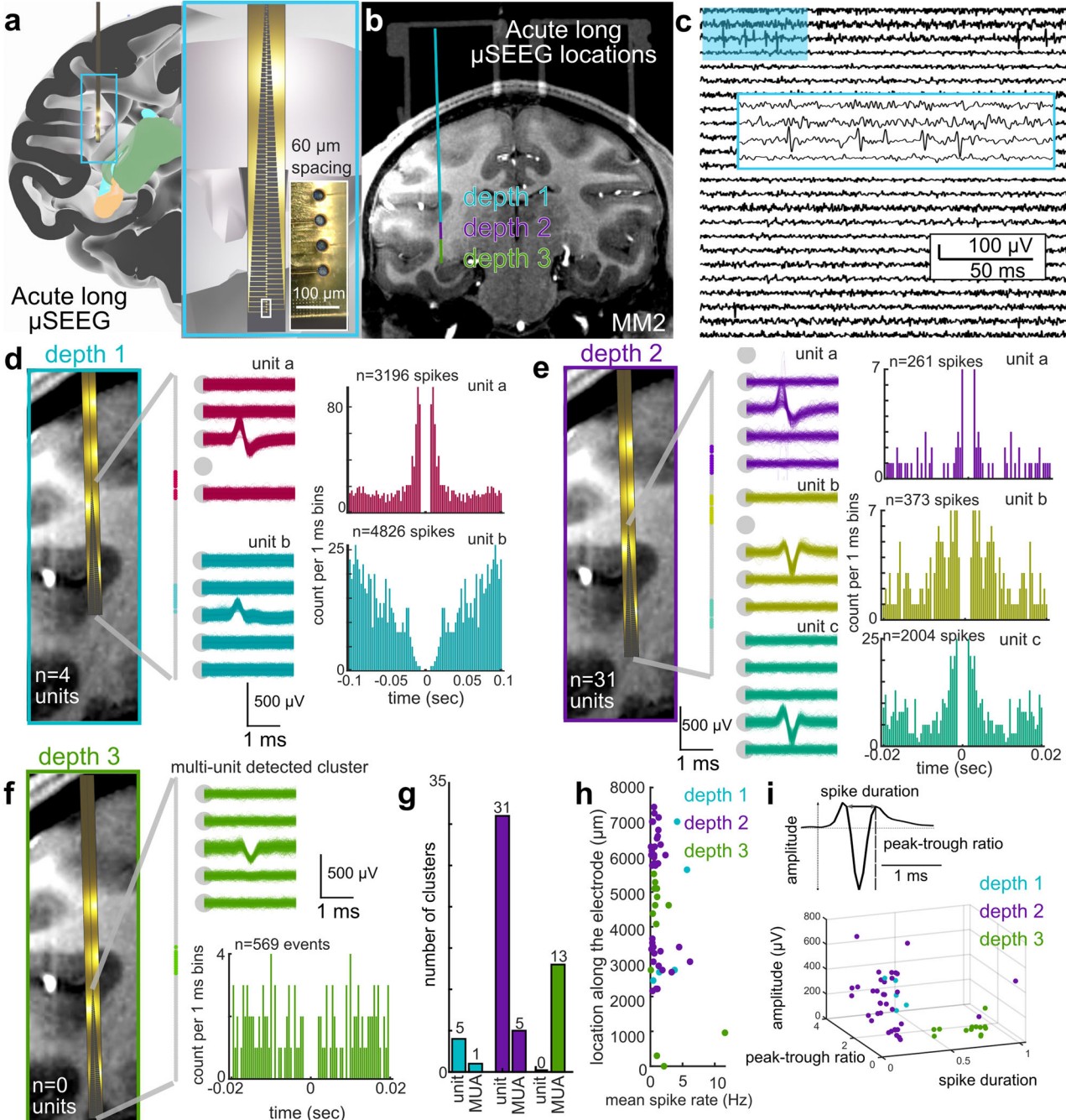

**Fig. 4 | Single-unit activity could be recorded using long μSEEG electrodes.**
**a** Three-dimensional reconstruction of the locations of the long μSEEG inserted at multiple depths into the parietal lobe, temporal lobe, and deeper into the tissue, with a zoomed-in view of the microelectrodes in the long μSEEG[49,50]. **b** MRI with the overlaid CT (chambers above) and the three putative long μSEEG depths in the brain. **c** Example recording from the second depth to show single-unit spiking activity (filtered to between 300 and 6000 Hz). **d**, **e** Example single units recorded at electrode depth 1 (**d**) and depth 2 (**e**) showing overlaid waveforms and the autocorrelation of the spike times. **f** Single-unit activity was not observed at depth 3 which seemed to be in white matter, but possible MUA was recorded at depth 3. **g** Numbers of detected single units and MUA clusters. **h** Location-detected clusters relative to the mean spike rates for the different depths. Each dot is a cluster (which can represent single units or MUA). **i** Spike waveform measures of single units and MUA waveforms showing separation of events detected at depths 1 and 2 versus 3.

delivering stimulation in saline (Supplementary Fig. 26). In addition to manufacturing variable contact sizes with this approach, the PtNR contacts at a 1 mm diameter offer smaller voltage transients and higher electrochemical safety limits when delivering direct electrical stimulation than clinical electrodes of similar surface areas (Supplementary Fig. 26). Stimulation and recording contacts can be distributed uniformly or in clusters across the length of the μSEEG. Thus, the μSEEG can offer micro- and macro-stimulation capability for use in future deep brain stimulation (DBS) or direct electrical stimulation application.

In conclusion, these μSEEG electrodes provide an innovative approach enabling recording across the entire depth of the human brain with greater resolution than ever achieved before. The smaller volume of the μSEEG electrode and its compatibility with procedures used in clinical practice paves the way to increasing our understanding of brain diseases and offering unique and clinical interventions.

## Methods

### μSEEG fabrication

Polished and cleaned soda lime glass plates were used as substrates. A release layer of Micro-90 diluted with deionized water was spin-coated onto the glass. A sacrificial 5-μm-thick polyimide layer (PI-2611, HD MicroSystems) was then deposited. This layer would later separate the device layers from the glass plate. A Ti hard mask was formed for net layer formation, followed by standard lithography, descum, metal deposition, and lift-off processes. The Ti hard mask contained via patterns for hole arrays and a rectangular shape for a sheath. Next, another polyimide layer was applied, serving as the sacrificial bottom layer with holes. A Ti sacrificial layer was deposited to act as an etch-stop layer. Adhesion between the layers was increased by patterning hole arrays. The 1st PI layer was then selectively etched, and the photoresist layer was removed. Afterward, the glass substrate underwent baking, and a 2nd PI layer was applied. Metal traces were formed on the 2nd PI layer, composed of Cr/Au (10/250 nm). This process was repeated to create double-layered metal leads. A PtAg alloy was selectively formed on the micro-contact recording sites. A Ti capping layer was added to prevent oxidation. A 3rd polyimide layer was applied, followed by a Ti hard mask. Etching processes exposed the Ti layer deposited during Ti hard mask for net layer formation on the sacrificial PI layer. Another photolithography and etching process was applied to open via holes for recording sites and contact pads. The exposed PI layers were etched, and a Ti passivation layer and parylene C were deposited to protect PI layers against dealloying. The Ti passivation layer and parylene C layers were then patterned and etched selectively on the recording site regions to expose the PtAg alloys. Dealloying of PtAg alloys was performed. Then, the parylene C and Ti passivation layers were removed. The electrodes were delaminated from the glass substrate. The delaminated electrode was transferred onto another carrier glass wafer. Hole arrays were formed on the 1st PI layer by etching through the sacrificial PI layer and the Ti hard mask for net layer formation. The Ti hard mask for net layer formation and Ti sacrificial layers were dissolved in BOE, and the electrode was rinsed with DI water.

### Rat experiments

The rat experiments were conducted with approval from the University of California San Diego Institutional Animal Care and Use Committee (protocol # S19030). All rats (Sprague-Dawley) were all male, and 3 months old. Acute in vivo electrophysiological recordings were performed on the rat primary somatosensory "barrel" cortex (S1) with the μECoG electrode and the μSEEG electrode. The rat was anesthetized and craniotomy was performed under isofluorane anesthesia. The body temperature of the rat was maintained at 37 °C with a heating pad. Craniotomy and dura removal were performed over the right barrel and surrounding cortical region. Following electrode placement, the rat was transitioned to ketamine/xylazine anesthesia for recording. Tactile stimulation was performed by delivering air puffs to the whisker pad. Air puffs were pressure-injected through a glass micropipette using a PV830 pneumatic picopump (World Precision Instruments, Inc.) with 1 s pulses ($n > 10$ trials per location). The contralateral (left) whiskers with respect to the recording sites were deflected by air puff (±2 mm). First, the whole contralateral whiskers (multi-whisker) were stimulated. Then single whiskers (C1-3, D1-3, E1-4) were stimulated by placing the pipette as close as possible to each whisker to avoid deflection of the neighboring whiskers. Recording data were collected for 60 s for each whisker.

### Pig experiments

The pig experiments were conducted with approval from the University of California San Diego Institutional Animal Care and Use Committee. The pigs (Yucatan) were all female, and 7-month-old. The pigs were induced with isoflurane and intubated. After anesthesia,

the pig was positioned in a stereotaxic frame in a prone position. Vital signs were closely monitored, including heart rate, blood pressure, EtCO$_2$, respiratory rate, and blood oxygenation. The surgical site on the cranium was focused on the motor cortex of the frontal lobe and the somatosensory cortex of the parietal lobe. A skin incision (2–4 inches long) was made, followed by a unilateral craniotomy using a high-speed surgical drill. The removed bone created a window of approximately 25 mm × 15 mm. The underlying dura was cut and moved towards the sagittal sinus. The cortex was hydrated with a normal saline solution. A sterile multielectrode implant was then placed on the surface of the exposed cortex. After placement, the electrode ribbon (less than 1 cm in width) was connected to an Intan recording controller. For spinal exposure, a midline incision and laminectomy were performed to sufficiently expose the target thoracolumbar enlargement of the spinal cord. Following laminectomy, a longitudinal incision was made in the dura, allowing for the exposure of the spinal cord. Spinal cord stimulation was carried out using a handheld Ojemann stimulator with two ball tips (Radionics Inc., Burlington, MA) spaced 0.5 cm apart. Isoflurane anesthesia was discontinued after completion of surgical procedures and replaced with IV Propofol for the duration of the stimulation testing.

### NHP experiments

The experimental procedures on rhesus macaques were conducted in compliance with the Guide to the Care and Use of Laboratory Animals. Measures were taken to minimize discomfort, and the Institutional Animal Care and Use Committee at Massachusetts General Hospital oversaw and approved all procedures. The study involved testing μSEEG electrodes in two scenarios: (1) recording from a short μSEEG in an anesthetized NHP in the operating room for visual cortical dynamics, and (2) recording from a long μSEEG in an awake NHP with the electrode lowered through an implanted chamber using a standard Microdrive.

In the first setting, recordings were obtained from an adult male rhesus macaque (age 11) under general endotracheal anesthesia with isoflurane. A craniotomy was performed over the visual cortex, and the short μSEEG electrode was carefully implanted. Signals were recorded using a custom Intan Recording System.

For the awake NHP preparation, another adult male rhesus macaque (age 14) was implanted with two recording chambers to access different brain regions. A Microdrive was attached to allow insertion through the dura. Trajectories were determined by mapping Microdrive depths to preoperative magnetic resonance imaging (MRI) and postoperative CT scans. The electrode was lowered to record neural activity at three different depths. No noticeable adverse behavioral effects were observed before or after electrode implantation or removal. Recordings of the long μSEEG utilized a 1024-channel Intan Recording system with a specialized device for recording thin film microelectrodes.

### Human tests

The study involved intraoperative recordings on two participants undergoing neurosurgery at Massachusetts General Hospital (MGH). The research had received approval from the Massachusetts General Brigham (formerly Partners) Institutional Review Board. These participants, aged 28 and 46, one female and one male, were already scheduled for a craniotomy for various clinical neurophysiological monitoring purposes, including mapping motor, language, and sensory regions, as well as tissue removal for epilepsy treatment. Both individuals provided voluntary, informed consent, understanding that their clinical care would remain unaffected and they could withdraw from the study at any time without impacting their treatment. These patients were originally scheduled for left anterior temporal lobe surgery for epilepsy or tumor treatment. The option for research recordings was considered only after the decision to proceed with

surgery was confirmed. Neither participant was in a medically unstable condition or in need of urgent surgery. The participants were not monetarily compensated for their involvement. All decisions were made in consultation with the treating neurosurgeon and clinical team. Patients lacking decision-making capacity were not included, as determined by the primary clinical team or physician. The time allocated for research recording for each subject was limited to minimize risk. Participants also consented to the sharing of de-identified neural data.

### Reporting summary

Further information on research design is available in the Nature Portfolio Reporting Summary linked to this article.

## Data availability

All data obtained in this study are either presented in the paper and the Supplementary Materials or deposited in open database. Animal brain recording data is available on OpenNeuro (https://openneuro.org/ at Accession Number ds004819) and human brain recording data could be accessed at Data Archive BRAIN Initiative (DABI) (https://dabi.loni. usc.edu/ at https://doi.org/10.18120/dn61-9y73) using the iEEG BIDS format[44]. To visualize the locations in 3D in the non-human primate and rodent brains, we used the Scalable Brain Atlas with the Calabrese atlas (https://scalablebrainatlas.incf.org/ ; exported into Blender (https://www.blender.org/). Source Data are provided with this paper.

## Code availability

Data were acquired using OpenEphys (http://www.open-ephys.org/) and Intan software (https://intantech.com/RHX_software.html). Most of the data were extracted and processed using MATLAB (Mathworks, Natick, MA). Custom MATLAB code (version R2021a) are available in GitHub (https://github.com/Center-For-Neurotechnology/MicrosEEG_ Data_Analysis, Zenodo https://doi.org/10.5281/zenodo.10042080). Spike sorting was performed using kilosort 2.5 (https://github.com/ MouseLand/Kilosort) with further determination of single units versus multi-unit activity performed in post-processing using Phy (https:// github.com/cortex-lab/phy) and then manually curated using in-house MATLAB code to visually inspect the template as well as the waveforms assigned to each cluster. We detected bursts and calculated the burst suppression ratio (BSR) using an automated method (https://github. com/drasros/bs_detector_icueeg). For all clusters, we measured the spike duration, peak-trough ratio, and amplitude measures (Fig. 4; code adapted from https://github.com/jiaxx/waveform_classification).

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

## Acknowledgements
We are grateful for the technical support from the nano3 cleanroom facilities at UCSD's Qualcomm Institute where the depth electrode fabrication was conducted. This work was performed, in part, at the San Diego Nanotechnology Infrastructure (SDNI) of UCSD, a member of the National Nanotechnology Coordinated Infrastructure, which is supported by the NSF (grant ECCS1542148). We thank Yangling Chou, Aaron Tripp, Fausto Minidio, Daniel J. Soper, and Alexandra O'Donnell for their help in collecting the data.

## Author contributions
K.L., A.C.P., Y.G.R., E.H., S.S.C., and S.A.D. initiated the concept and designed the studies. K.L., A.C.P., Y.G.R., D.R.C., K.T., Y.K., J.P., Y.T., J.L., A.M.B., R.V., J.R.M., S.M.R., J.C.Y., A.B., R.M.R., Z.M.W., S.I.F., A.M.R., S.B.H., E.H., S.S.C, and S.A.D. contributed to methodology. K.L., A.C.P., Y.G.R., D.R.C., K.T., Y.K., J.P., Y.T., J.L., A.M.B., R.V., S.M.R., J.C.Y., A.B., R.M.R., Z.M.W., S.I.F., H.S.U., E.H., S.S.C., and S.A.D. contributed to the experiments. K.L., A.C.P., and Y.G.R. led the experiments and collected the overall data. S.A.D., E.H., and S.S.C. contributed to the funding acquisition. S.A.D. and S.S.C. administrated the project. S.A.D., E.H., S.S.C., J.P., M.R., Z.M.W., and S.I.F. supervised the work. A.C.P., K.L., Y.G.R., and S.A.D. co-wrote the paper. K.L., A.C.P., Y.G.R., D.R.C., K.T., Y.K., J.P., Y.T., J.L., A.M.B., R.V., J.R.M., S.M.R., J.C.Y., A.B., R.M.R., Z.M.W., S.I.F., H.S.U., A.M.R., S.B.H., E.H., S.S.C., and S.A.D. provided feedback on the manuscript.

## Funding
National Institutes of Health BRAIN® Initiative UG3NS123723-01 (S.A.D.). National Institutes of Health BRAIN® Initiative R01NS123655-01 (S.A.D.). National Institutes of Health NBIB DP2-EB029757 (S.A.D.). National Institutes of Health F32 postdoctoral fellowship MH120886-01 (D.R.C). National Science Foundation Award no. 1728497 (S.A.D.). National Science Foundation CAREER no. 1351980 (S.A.D.). National Science Foundation Graduate Research Fellowship Program no. DGE-1650112 (A.M.B.). MGH—ECOR (S.S.C.). K24-NS088568, R01-NS062092 (S.S.C.). Tiny Blue Dot Foundation (to S.S.C. and A.C.P.). National Institutes of Health BRAIN® Initiative K99 NS119291 (K.J.T.). National Eye Institute R01EY027888 (J.S.P.), William M. Wood Foundation, Bank of America Trustee (J.S.P.)

## Competing interests
The authors declare the following competing interests: K.L., Y.G.R., and S.A.D. and the University of California San Diego filed a patent application (#63/584,578, pending) for the manufacture of the novel depth electrodes. A.C.P., D.R.C., Y. T., A.M.R., S.B.H., E.H., S.S.C., and S.A.D. have competing interests not related to this work including equity in Intelecterra Inc. S.A.D. was a paid consultant to MaXentric Technologies. A.M.R. has equity and is a cofounder of CerebroAI. A.M.R. received consulting fees from Abbott Inc and Biotronik Inc. The MGH Translational Research Center has clinical research support agreements with Neuralink, Paradromics, and Synchron, for which S.S.C. provides consultative input. The other authors declare that they have no competing interests.
