## [Peer Review File · Nature Communications]

Flexible, scalable, high channel count stereo-electrode for recording in the human brainREVIEWER COMMENTS

Reviewer #1 (Remarks to the Author):

In this work the authors describe a thin-film sEEG alternative/replacement with up to 128 channels termed uSEEG. The devices are fabricated on glass wafers to allow for device sizes compatible with use in humans. They use a Ti sacrificial layer to generate a sheath in which a stylet is used for temporary stiffening/insertion of the device. There are multiple versions of the device in number of electrodes, layout, substrate, and of electrode surface - platinum nano rods vs. PEDOT. Using these variety of devices, including a 128-ch device with 60 micron center-to-center electrode distances, the authors demonstrate that the uSEEG is able to record physiologic signals in acute and chronic experiments in rodent barrel cortex, as well as acute experiments in porcine S1, multiple cortical structures in non-human primate, and human anterior temporal lobe. In each of these recording sites and subjects, they demonstrate expected physiologic findings, including auditory responses in one of the human subjects and acutely recorded single units in non-human primates during PEDOT coated device insertion.

There is a heroic amount of engineering and demonstration of the feasibility of the technology, which represents the first advancement to a longstanding clinical technology in decades, however, the claims of single-unit recording and chronic recording are with significant caveats based on what is shown here. As it stands now, it is not clear whether the promise of being able to replace clinical sEEG electrodes with uSEEG for days-weeks long implantations is possible in large animals, or if single unit recording is possible other than transiently during insertion with PEDOT devices, which importantly, were abandoned in favor of the PtNR devices due to PEDOT delamination - a long-standing problem with PEDOT. Further, while the devices are convincingly able to detect true physiologic signals, the technologic demonstration would be greatly strengthened if the greater spatial resolution of the uSEEG over SEEG enabled observation of any phenomena at scales previously impossible.

Altogether, the manuscript could be suitable for publication in Nature Communications with either new experiments or further analyses to demonstrate the potential advancements of the technology. The work requires either further demonstration of chronic recording in large animals and chronic single-unit recording in any model system, or the language of the claims must be clarified to include the significant aforementioned caveats. Even with these the caveats of the claims and without further recordings, the work still is a significant step toward advancing a decades-old established clinical tool and would be suitable for publication if there was some demonstration of what the technology was capable of through further analysis of the existing data.

Major Issues

1) Chronic recording - demonstrated in rodents

Compared to rodents, large animal or human chronic recording will have devices encounter orders of magnitude larger forces and motion. While the tensile strength is shown on page 4/Supplemental fig 4, and address any concerns of device fracture, at what point do you have electrical device failure? Given the fine metal traces of thin-film arrays compared to larger robust wires in SEEG, it would be expected that the uSEEG traces may crack at relatively small forces.

Similarly, the thin traces may be more susceptible to bending-related fatigue and electrical failures. Has accelerated lifetime testing including repeat stressing been carried out in these devices?

2) Single-unit recording - demonstrated in NHP during device insertion

The claims about single unit activity need to be more clearly stated, as there is no demonstration of single unit activity in humans, and also no demonstration of chronic recording of single unit activity. Moreover, it is not clear but seems as though all putative single unit activity was recorded with the stylet in place. It is possible that the act of stylet removal can stun/further damage surrounding tissue or the reduced thickness of the device after stylet removal may lead to loss of single unit recording. Was there any single unit activity in any recordings after the stylet was removed? Any single units in the chronic rodent data? In HS2?

Minor Issues

3) Common-mode signal - mentioned in the discussion pg 15

Which dataset(s) does this refer to?

Perhaps this is due to the decreased diameter of the device after stylet removal. The cavity created by device insertion may be larger than the uSEEG to the point where the device, especially acutely in large animals, may be partially sitting in a single electrically-coupled fluid. Does the degree of common mode signal change after stylet removal? To what degree is the signal coordinated in comparison to similarly spaced electrodes on a clinical/commercially-available device?

4) Device yield

What were the device and channel yields? Were there any device losses from loading the stylet? Supplemental Fig 6H - is this data from 6 devices?

5) PEDOT delamination

When was the PEDOT delamination seen? Was there ever delamination at time of insertion? This could be a safety concern in humans especially for penetrating devices.

6) Motion in humans

Are the uSEEG flexible enough to move with the brain or are there still significant motion artifacts? Can human cortical lamina be resolved? If yes this would be a significant contribution/capability of the device.

Reviewer #2 (Remarks to the Author):

Review Nature Communication 406735

The manuscript describes a new probe concept based on a polymeric substrate comprising an open channel / sheath realized using a sacrificial layer. The probe is inserted into deeper brain areas using a stylet inserted into the sheath. This stylet renders the probe temporarily stiff facilitating insertion into deeper brain areas. The stiffening is attributed to an increased probe cross section which might collapse to the initial thickness of the polymeric substrate layers after stylet retraction (which has not been demonstrated or better which is not obvious from the figures presented, e.g. Suppl. Fig. 6). In contrast to insertion procedures which rely on a stiffer wire hooked to the probe (see among others Luan et al. doi 10.1126/sciadv.1601966), the probe shape described in this manuscript is smooth without any wire tip protruding from the implantation hole at the probe tip as used by Luan et al. It has however to be taken into account that the stylet sheath represents an open lumen connecting the brain surface to deeper brain areas (see Fig. 1(f,g) and Suppl. Fig 2). The authors provide an impressive amount of experimental data recorded in different species from rat, pigs, non-human primates and human cortex (due to clinical approval of limited implantation depth and only as acute recording experiments). It seems that really deep recordings have not been demonstrated. Further, the title of the manuscript suggests that mainly recordings in human cortex have been done. This is however a single experiment out of a variety of tests. I recommend to highlight the new probe that has merit for clinical applications.

The paper lacks long-term data (experiments have been done for up to 2 weeks only). I would be interesting to see which effects a fairly wide probe (from Suppl Fig 2f estimated to 1 mm in width) has on its long-term performance. In addition, the probe has an anisotropic stiffness and might even be stiffer than expected given the fact that the stylet causes a permanent deformation of the probe sheath (see Suppl Fig. 6).

The authors need to discuss the number of electrodes per probe as this is to a certain degree limited by the density of interconnection lines to be integrated. Please compare the probe presented in the manuscript with rolled solutions among others by Pothof et al. 2016 (F. Pothof, L. Bonini, M. Lanzilotto, A. Livi, L. Fogassi, G. Orban, O. Paul, and P. Ruther, "Chronic neural probe for simultaneous recording of single-unit, multi-unit, and local field potential activity from multiple brain sites," *Journal of neural engineering*, vol. 13, no. 4, p. 046006, 2016), Kang et al. 2015 (X.

Kang, J.-Q. Liu, H. Tian, B. Yang, Y. Nuli, and C. Yang, "Self-closed parylene cuff electrode for peripheral nerve recording," *Journal of Microelectromechanical Systems*, vol. 24, no. 2, pp. 319–332, 2015), and van der Puije et al. 1989 (van der Puije P D, Pon C R and Robillard H 1989 Cylindrical cochlear electrode array for use in humans *Ann. Otology, Rhinology, Laryngology* 98 466–71).

In view of probe development and performance, the reviewers observed a couple of aspects that would request a further in-depth analysis and discussion – as further detailed below.

- Abstract

- o The term reconfigurability needs to be defined. I would rather mention that probe can be used across several species with adequate design modifications. This aspect is however questionable at least for rat experiments given the large probe width of 1 mm).

- o The authors mention clinical neurotechnology. Potential difficulties in the approval process should be discussed as the probe deviates completely from clinically approved cylindrical probes; how likely is it that this specific probe is approved taking into account that it might act like a blade when forces act parallel to the surface of the fabrication while the probe is highly bendable for forces in the orthogonal direction; this anisotropic bending compliance needs to be discussed.

- Specific comments

- Line 60: the SEEG probes by DIXI, Adtech, PMT Corporation have cylindrical electrodes with heights between 1.5 mm and 3 mm

- Line 79 – NeuroPixels currently plans for probes as long as 40 mm; ATLAS and Cambridge NeuroTech presented silicon-based probe beyond 40 mm in length

- Line 84 – Pothof et al presented small electrodes (30 μm in dia.) on their cylindrical SEEG probe capable of recording single unit activity over ca. 50 days

- Manufacturing

- o Line 124 – Here, I would already explain the purpose of these circular openings in the Ti-based hard mask.

- o Line 133 – just mention that the holes introduced into the 3rd PI layer define the electrodes

- o Line 137 – how is the probe temporarily adhered to another glass substrate? How good is the thermal contact between polyimide probe and the glass to avoid excessive heating during plasma etching.

- o Line 141 – Grammar – ... openings are drilled through the 1st PI layer ** to expose ** the titanium ...

- o Line 146 – Suppl Figure 2 does not really illustrate the insertion of the stylet; a schematic representation would be good to have; the photographs should be made at higher magnification

- o Line 150 – I like the clever concept of openings in the 1st PI layer to interlock PI layers 1 and 2

- o Line 153 – rounded shape of the stylet – how do you insert the rounded stylet into the narrow gap defined by the sacrificial layer

- Probe design

- o Authors should discuss the anisotropic stiffness of these probe, i.e. the probes are fairly flexible orthogonal to the wafer plane but act like a knife when bent parallel to the wafer plane of probe fabrication. This is even more relevant (see next point) when the probe shape is permanently deformed due to the stylet insertion.

- o Authors should further illustrate whether the stylet insertion into the probe results in a permanent plastic deformation of the polyimide substrate – an aspect that is frequently observed with polyimide probes using a wire inserted into a hole at the probe tip (compare probes by Lan Luan et al doi 10.1126/sciadv.1601966). Looking at Figure 1, it is unclear whether the probe shape is again flat when the stylet is removed. Suppl Fig 2 shows the probe with stylet ; will the eye-shaped probe geometry collapse after stylet retraction?

- o Line 207: the figure caption of Fig 1 mentions holes to enable probe integrity. Is the adhesion between the finally applied polyimide layer that weak that this rivet-like structures are needed?

- o Line 228: Please motivate the different electrode diameters for the PEDOT-PSS and PtNR electrodes.

- o Please mention the design specs, among other minimal line and width spacing of the metal tracks. It would be helpful to have these numbers in the main text and not just in the supplementary information.

- o The interface technology and the external instrumentation needs to be discussed. A reasonable approach has been presented by the Yoon Lab in Michigan (Park et al., doi 10.1109/TBME.2021.3093542)

- Probe performance

- o Line 177/178: The authors mention an increase in electrode impedance upon insertion of the stylet. It would be interesting whether the increase is related to the mechanical stress caused by the stylet insertion or whether this would happen anyhow exposing the electrodes to the saline solution. Please provide control experiments.
- o Line 188: GFPA staining: the conclusion on biocompatibility is based on a single experiment in a mouse implanted for 14 days. In particular the increased probe width might cause increased long-term trauma.
- o Line 195: PtNR surface seems to be unaffected by insertion into brain tissue. Is that stability potentially related to the recessed character of the electrodes? Please compare the electrode performance to electrodes based on Pt nanograss by Boehler et al. (doi 10.1021/acsami.9b22798)
- o Lines 217-227: Cracks are seen in the PEDOT-PSS as well as parylene C layers. Are control experiments available that indicate the electrode stability upon immersion into saline solution? Or are the delamination and crack formation solely correlated to the stylet insertion into the probe? Please provide FEM analysis investigating the mechanical stress in the different probe layers caused by stylet insertion.
- o Please explain the large discrepancy in electrode impedance between ca. 35 kOhm (line 204) and up to 268 kOhm (line 280)
- o Cross-talk between different electrodes / interconnection lines needs to be validated (line 489). It would be interesting whether this is due to a capacitive coupling inside the probe or caused by a certain degree of signal spreading inside brain tissue.
- Figures
 - o Figure 1: Fig 1(c) shows the layer composition during probe fabrication not of the finally realized probe; this needs to be clarified.
- Supplementary information
 - o Probe fabrication – information is somehow doubled; please rewrite to streamline the information that needs to be delivered; examples are the description of the sacrificial layer and the Ti-based hard mask (lines 107-117).
 - o Probe fabrication – naming of the different PI layers should be streamlined, i.e. there is an initial sacrificial PI layer on which a hard mask is deposited and patterned; this is followed by a 1st probe PI layer on which a sacrificial Ti layer is deposited and patterned; the Ti layer also serves as an etch stop when patterning etch holes into the 1st PI layer, to ultimately etch the sacrificial Ti layer ...
 - o Probe fabrication – is the sacrificial Ti layer really just 60 nm thick? How long do you need to etch this layer?
 - o Probe fabrication – line 132 – why is the Ti cap layer needed as the Ag has already been etched
 - o Stylet insertion – Suppl Fig 2 requests more details zooming into the position of the threading hole
 - o Suppl line 245 – why is the ECoG electrode array described here at all?
 - o Stylet retraction – Suppl Fig 5 impressively shows to stability in implantation depth after stylet retraction. Suppl Fig 6 indicates that the probe sheath is plastically deformed which renders the probe mechanically stiff. Will the open lumen (the sheath and the respective etch holes to remove the sacrificial layer) between the 1st and 2nd PI layer represent a potential path of infection into deeper brain structures? In any case, the fact that the probe is plastically deformed needs to be mentioned in the main part of the manuscript.

Reviewer #3 (Remarks to the Author):

In this manuscript, Lee et al. introduce a flexible micro-stereo-EEG (uSEEG) electrode for possible human subject recording. They describe the fabrication process and use of electrode materials that permit miniaturization of the neural interface without compromising impedance. This is a very interesting and promising project that has significant potential for future applications as a combined human clinical and research tool. However, there are many areas to be clarified by the authors:

The authors employ a fabrication process that employs a sacrificial Ti interlayer to allow creation of a channel between two polyimide layers. Figure SI2F demonstrates a SEM image that clearly

shows the two PI layers are not merged to form a single block at the edges. What is the cause of this and how does it affect the yield and reproducibility of the devices?

It is important to establish a scalable fabrication process. Currently the process spans more than 1 mm (Figure SI2F). Ability to miniaturize would likely improve the device biocompatibility and signal quality. What is the smallest bonding width requiring for such processes and what dictates this limit?

Regarding the pull tests performed, were all sections of the device tested, or just the tip? The geometry of the full device contains several right angles that may be more sensitive to disruption. In the quantitative comparison, the force required to disrupt the uSEEG electrode was 4 times less than for the PMT electrode. Could the authors please comment on the practical implications of this decreased resistance to pulling forces?

(line 169) Also please check the units – why does 1 MPa correspond to 16 mN and 14 kPa correspond to 48 mN? Figure SI4 does not contain any tensile strength data.

(line 223) "We transitioned to all polyimide PtNR μ SEEG electrodes after we observed that parylene C PEDOT:PSS μ SEEG develop cracks in the parylene C layers and in the PEDOT:PSS layers after ..."

We were not able to find any data in the manuscript supporting this claim. We urge authors to be more careful with such blanket statements. Often times, both the material as well as processes employed to produce such layers are critical for evaluation of the quality and stability of the overall layer.

Parylene C is a chemically vapor deposited polymer, and therefore as a soft polymer it does not "crack". Incompatible thermal cycles or chemistry during the fabrication process as well as poor deposition control of the material are likely the main source of these defects. This material has been approved and extensively used clinically for long-term chronic implants and components. The same can be said in regards to PEDOT:PSS. The authors employed a simplified electro-polymerization process that allows deposition of free PEDOT:PSS onto a conducting surface. From our understanding, no chemistry or crosslinkers were employed or investigated to improve this interface. Thus, the issues observed (though no data is presented in the manuscript) are potentially related to the authors' fabrication process rather than intrinsic properties of the material itself. Indeed, there are several publications that systemically and quantitatively evaluate this topic (e.g. Oldroyd et al Adv Func Mat 2022) and show, using appropriate controls and processes, that the yield of PEDOT:PSS layers surpasses metal interfaces in stability.

The electrophysiology traces in several of the figures (Figure 2, 3 and several of the supplementary figures) are very difficult to interpret due to small size, incorporation of too many traces, and overlaid green highlighting. Clarity of data presentation could be overall improved.

The authors point out successful recordings in 3/9 rat whisker barrel experiments. Greater discussion of why only 1/3 of recordings were successful would be useful.

The animal experiments demonstrate patterns consistent with recording across cortical layers, but in no case was the quality of recording compared to any conventional device. The lack of this comparison makes it difficult to evaluate whether there are any limitations of the data generated by these devices. As the authors mention, they had to remove a strong common-mode signal on all contacts in order to derive CSD dynamics. Side by side comparison of the uSEEG with a conventional silicon probe in animal experiments would delineate this issue and determine the extent to which cross-talk is present across the channels. The utility of having >100 channels is severely limited if the channels are contaminated by cross-talk.

The authors also demonstrate that their technology allows contacts within 7mm at the tip of a long electrode. One of the fundamental properties of SEEG is the ability to have contacts along the entire length of the electrode, from the superficial cortex to the deep gyri to deeper brain targets of gyri. The authors demonstrate the ability to advance an electrode with tip contacts and sequential record from targets. Is this technology compatible with contacts along the length of the electrode, particularly with concerns over common-mode signal cross talk.

The ability to record single units with SEEG is currently a research advantage, without as yet documented clinical benefit. It is important to point out that while there are many potential benefits of being able to clinically record LFPs and single units, this merged technology is currently research and not clinical in the human domain of epilepsy SEEG, the only currently approved use of SEEG. Additionally, the goal of SEEG is not to create a 3-D grid, as this is impossible, and even more so with smaller unidirectional contacts as described here. The value of SEEG in treating human disease is only as good as the anatomo-electro-clinical hypothesis being investigated. It is important that newly developed uSEEG electrodes do not become a vehicle for scientific "fishing expeditions" because the technology allows access of single unit recordings to new areas of the brain.

A critical component of SEEG is the ability to target specific structures in the dorsoventral axis. What is the accuracy and precision of targeting with these uSEEG devices? Given that a stylet is used to insert the device, and the device itself is quite flexible and light, does the device shift along the insertion tract when the stylet is removed? The targeting capacity of the devices should be delineated.

What is the yield of single unit activity compared to number of channels for these devices? Were units detected across multiple channels?

It takes some searching to figure out the actual number of experiments done. For instance, the histology data between a conventional SEEG electrode tip and the uSEEG electrode appears to be based on 1 rat using one hemisphere for each electrode, at one time point, with limited GFAP and neuronal staining. Based on this, the authors concluded that the new uSEEG technology "induced less apparent tissue damage than clinical SEEG electrodes". In the same data section (lines 194-197), the authors contend that an apparently acute (as there were no chronic NHP experiments) implant into a NHP did not result in imaging changes to the electrode as demonstrating the "stability of the μ SEEG electrode in tissue (supplementary Fig. 8, N = 3)". This conclusion is an overreach of the the data.

Response Letter

Author Remark

We express our sincere gratitude to the reviewers for their positive evaluation and valuable feedback on our manuscript. In response to the reviewers' comments, we have carefully revised our manuscript to address their suggestions and concerns. In the following section, we respond to the reviewers' comments (displayed in red), addressing each point individually. Our responses are marked in black font, and the revised sections from the updated manuscript are copied and pasted in blue. The revisions are also appropriately highlighted within the updated manuscript.

Response to the Reviewer's comments

Reviewer: 1

In this work the authors describe a thin-film sEEG alternative/replacement with up to 128 channels termed uSEEG. The devices are fabricated on glass wafers to allow for device sizes compatible with use in humans. They use a Ti sacrificial layer to generate a sheath in which a stylet is used for temporary stiffening/insertion of the device. There are multiple versions of the device in number of electrodes, layout, substrate, and of electrode surface - platinum nano rods vs. PEDOT. Using these variety of devices, including a 128-ch device with 60 micron center-to-center electrode distances, the authors demonstrate that the uSEEG is able to record physiologic signals in acute and chronic experiments in rodent barrel cortex, as well as acute experiments in porcine S1, multiple cortical structures in non-human primate, and human anterior temporal lobe. In each of these recording sites and subjects, they demonstrate expected physiologic findings, including auditory responses in one of the human subjects and acutely recorded single units in non-human primates during PEDOT coated device insertion.

There is a heroic amount of engineering and demonstration of the feasibility of the technology, which represents the first advancement to a longstanding clinical technology in decades, however, the claims of single-unit recording and chronic recording are with significant caveats based on what is shown here. As it stands now, it is not clear whether the promise of being able to replace clinical sEEG electrodes with uSEEG for days-weeks long implantations is possible in large animals, or if single unit recording is possible other than transiently during insertion with PEDOT devices, which importantly, were abandoned in favor of the PtNR devices due to PEDOT delamination - a long-standing problem with PEDOT. Further, while the devices are convincingly able to detect true physiologic signals, the technologic demonstration would be greatly strengthened if the greater spatial resolution of the uSEEG over SEEG enabled observation of any phenomena at scales previously impossible.

Altogether, the manuscript could be suitable for publication in Nature Communications with either new experiments or further analyses to demonstrate the potential advancements of the technology. The work requires either further demonstration of chronic recording in large animals and chronic single-unit recording in any model system, or the language of the claims must be clarified to include the significant aforementioned caveats. Even with these the caveats of the claims and without further recordings, the work still is a significant step toward advancing a decades-old established clinical tool and would be suitable for publication if there was some demonstration of what the technology was capable of through further analysis of the existing data.

Response:

We thank the reviewer for their positive and detailed summary of the work. We would like to clarify one point: that the single unit activity we report here in an NHP was captured with the PtNR depth device, not PEDOT.

Major concerns

Comment R1-1) Chronic recording - demonstrated in rodents

Compared to rodents, large animal or human chronic recording will have devices encounter orders of magnitude larger forces and motion. While the tensile strength is shown on page 4/Supplemental fig 4, and address any concerns of device fracture, at what point do you have electrical device failure? Given the fine metal traces of thin-film arrays compared to larger robust wires in SEEG, it would be expected that the uSEEG traces may crack at relatively small forces.

Similarly, the thin traces may be more susceptible to bending-related fatigue and electrical failures. Has accelerated lifetime testing including repeat stressing been carried out in these devices?

Response R1-1:

We appreciate the reviewer's comment. In response to the suggestion, we have performed an accelerated aging test and bending tests and added results as follows (**supplementary Fig. 6**).

Changes to the manuscript:

Updated discussion, main text (Results), Page 5

To evaluate reliability of the μ SEEG electrodes for longer-term implant periods, we performed an accelerated aging test and a bending test showing negligible degradation over 150 days with 84,000 cycles of lead bending (**supplementary Fig. 6**).

Updated discussion and figure, Supplementary Information (supplementary Fig. 6), Page 9

The reliability of μ SEEG was evaluated through an accelerated aging test to validate the use of our electrode in semi-chronic applications. The electrode was immersed in 50 °C saline solutions for over 74 days to assess its yield after >150 days of implantation⁸. OM images of the μ SEEG before and after the aging test showed negligible changes (**supplementary Fig 6a and 6b**), indicating good structural integrity. During the test, impedance measurements at 1 kHz exhibited minor fluctuations in both mean and standard deviation values (**supplementary Fig 6c**). Prior to the accelerated aging test, we measured 49 functional contacts with an average impedance of 94 ± 26 k Ω . After the test, we observed 52 functional contacts with an average impedance of 96 ± 28 k Ω (**supplementary Fig 6d**). These results demonstrate the robustness of the μ SEEG electrode even under accelerated aging conditions. In addition, we performed 84,000 cycles of lead bend testing, exceeding 90° bends as per EN 45502 standards, to validate the capability of our electrode for general application in active implantable medical devices⁵. A robotic gripper model 2F-140 (Robotiq) was used for the bending tests. After bending, 50 contacts (average impedance of 93 ± 27 k Ω) were functional. A few changes in number of functional channels are attributed to different contact latching between the PCB and the custom ironwood electronics socket that requires latching to form robust electrical connections between PCBs and pins in the socket on the acquisition board and is minimal indicating resilience of our electrodes to bending cycles.

Supplementary Fig. 6. Reliability of our μ SEEG for long term implantation. OM images of our μ SEEG (a) before and (b) after the accelerated aging test. (c) Impedance of our μ SEEG at 1 kHz during the accelerated aging test. (d) Impedance histogram of our μ SEEG before, after the aging test, and after the bending tests.

Comment R1-2) Single-unit recording - demonstrated in NHP during device insertion. The claims about single unit activity need to be more clearly stated, as there is no demonstration of single unit activity in humans, and also no demonstration of chronic recording of single unit activity. Moreover, it is not clear but seems as though all putative single unit activity was recorded with the stylet in place. It is possible that the act of stylet removal can stun/further damage surrounding tissue or the reduced thickness of the device after stylet removal may lead to loss of single unit recording. Was there any single unit activity in any recordings after the stylet was removed? Any single units in the chronic rodent data? In HS2?

Response R1-2:

We thank the reviewer for their excellent questions. We agree there could be confounds in capturing single unit activity following stylet removal as the device will deflate in the process. We did not do this experiment at the time in the NHP recording (and will not be able to collect further data in an NHP) but agree that this step could have demonstrated a possible strength in this technology. The rodent recordings, for technical reasons, were sampled at lower

frequencies (15 or 20 kHz) which made single unit identification not possible. Finally, we examined the N=2 data set of human data for single units. We used both WaveClust and Kilosort to attempt to sort single units but the noise levels in these recordings with these early versions of this device did not allow for detection of single units. We found some changes in the high gamma power which could reflect underlying single unit activity but we do not report this result here. We believe, however, that the improvements in the manufacture in the latest version of the μ SEEG electrodes (which is represented by the long μ SEEG device) allowed for single unit recording.

Minor concerns

Comment R1-3) Common-mode signal - mentioned in the discussion pg 15 Which dataset(s) does this refer to?

Perhaps this is due to the decreased diameter of the device after stylet removal. The cavity created by device insertion may be larger than the μ SEEG to the point where the device, especially acutely in large animals, may be partially sitting in a single electrically-coupled fluid. Does the degree of common mode signal change after stylet removal? To what degree is the signal coordinated in comparison to similarly spaced electrodes on a clinical/commercially-available device?

Response 3:

We appreciate the reviewer's comment. The recording sessions were conducted after the stylet was retracted, and we agree that the cavity created by the device insertion is larger than that of the μ SEEG. However, it is still possible that we measured the same neuronal activities due to the narrow spacing of the electrode contacts (60 μ m). Therefore, a confound here is that the contacts are so close to one another that they are recoding volume-conducted signal in addition to the issue of recording along an electrically-coupled fluid chamber. Future work to expand the use of these devices could be to use large devices with wider contact spacing. Additionally, we observed that the location of the reference contact also affects the amount of common signal captured. Moving forward, we will continue to optimize our recording setup to improve the recording quality.

Comment R1-4) Device yield

What were the device and channel yields? Were there any device losses from loading the stylet?

Supplemental Fig 6H - is this data from 6 devices?

Response R1-4:

Supplemental Fig 6H is data from a single electrode. Your comment has helped us realize that we did not provide enough data to show the device yield. To show the channel yields from multiple electrodes, we have updated the **supplementary Fig. 6h** as shown in the following figure, by including data from 7 PEDOT:PSS-PI μ SEEG electrodes (**Fig. R1-1**). The channel yield before and after stylet insertion is 99.8 ± 0.5 % and 99.3 ± 0.7 %, respectively. The channel yields below reflect the electrode losses during the stylet insertion.

Fig. R1-1. Average electrochemical impedance magnitude at 1 kHz, before and post stylet insertion. The average and standard deviation were obtained from the number of channels of each device denoted in the bar. Channels that have impedance value larger than 500 kΩ were considered failed channel and excluded from the mean calculation. The channel yield before and post stylet insertion is $99.8 \pm 0.5 \%$ and $99.3 \pm 0.7 \%$, respectively.

To provide the averaged impedance magnitudes of multiple PEDOT:PSS μ SEEG electrodes, we updated the manuscript as follows (**supplementary Fig. 6**)

Note that **supplementary Fig. 6** has changed to **supplementary Fig. 8** due to the newly added figures in the revised manuscript.

Changes to the manuscript:

Updated discussion and figure, Supplementary Information (supplementary Fig. 8), Page 7

Supplementary Fig. 8. Manufacture and validation of the short PEDOT:PSS μ SEEG in polyimide (PI). (a) A schematic illustration of a prototype 64 channel μ SEEG electrode layout. (b) Exploded view of the layout at the electrode sites region. (c), (d) Optical images of the μ SEEG electrode after stylet insertion. (e), (f) SEM image of μ SEEG electrode showing PEDOT:PSS on Cr/Pt after stylet insertion. (g) Electrochemical impedance spectra of Pt/PEDOT:PSS recording electrodes. (h) Electrochemical impedance magnitude at 1 kHz of each channel before and after stylet insertion and comparison with Pt. (i) Averaged impedance magnitude at 1 kHz for 7 PEDOT:PSS μ SEEG electrodes, before and after stylet insertion.

Electrochemical impedance spectroscopy (EIS) of the electro-deposited PEDOT:PSS was performed, in 1X phosphate buffer saline (PBS) solution (**supplementary Fig. 8g**). Three electrode configurations i.e., PEDOT:PSS electrodes as the working electrode, Ag/AgCl electrode as a reference electrode, a large platinum electrode as a counter electrode was used. 10 mV root mean square (RMS) sinusoidal signal with zero DC bias were applied and the frequency was swept from 1 Hz to 10 kHz using a Gamry potentiostat (Gamry Interface 1000E; Gamry Instruments). Electrochemical impedance at 1 kHz is commonly used as the benchmark for the characterization of neural electrodes, as this frequency corresponds to spiking activity². The average impedance magnitude of PEDOT:PSS electrodes across 64 channels was 33.0 ± 2.47 kilohms at 1 kHz and their phase indicates their faradaic electrochemical interfaces. After stylet insertion, the average impedance magnitude maintained similar values of 35.0 ± 3.67 kilohms on 64 channels, still lower and consistent with Pt electrodes (**supplementary Fig. 8h**), indicating that the stylet was successfully inserted between the polyimide layers without any damage to the channel or PEDOT:PSS separation from Pt electrodes. **Supplementary Fig. 8i** displays averaged impedance magnitude at 1 kHz for 7 PEDOT:PSS μ SEEG electrodes, before and after stylet insertion. The channel yield before and after stylet insertion is 99.8 ± 0.5 % and 99.3 ± 0.7 %, respectively. The channel yields below reflect the negligible electrode losses during the stylet insertion.

Comment R1-5) PEDOT delamination

When was the PEDOT delamination seen? Was there ever delamination at time of insertion? This could be a safety concern in humans especially for penetrating devices.

Response R1-5:

We understand the reviewer's concern regarding the PEDOT:PSS delamination and its use in human cases. The delamination or apparent crack of PEDOT:PSS was observed from the parylene C devices after inserting a stylet in between two parylene C layers. This occurred due to mechanical damage applied to the parylene C by the stylet, when the stylet was not properly inserted (**Fig. R1-2**). The devices that had delamination of PEDOT:PSS were abandoned and not used for the recording. For the recording, we only used the devices that do not have tears or apparent cracks or delamination of PEDOT:PSS, which was confirmed by both optical microscope and impedance measurements done in PBS. However, to ensure reproducible fabrication of the device, we have decided to use polyimide only to fabricate the electrode.

Fig. R1-2. OM images of PEDOT:PSS μ SEEG electrodes with Parylene C (top, a-c) and polyimide (bottom, d-f). Parylene C electrodes (a) before and (b) after stylet insertion and (c) its zoomed-in image. Polyimide electrodes (d) before and (e) after stylet insertion and (f) its zoomed-in image.

Because our electrode sites possess a recessed wall, wherein the thickness of PEDOT:PSS is smaller than that of the via etched holes (supplementary Fig. 6f), PEDOT:PSS remained confined within the via holes even after extraction of the μ SEEG from the pig brain after recordings, as we can see from **Fig. R1-3** showing OM and SEM images. This indicates the safety of our electrodes for acute recording.

Fig. R1-3. An OM (a) and an SEM (b) image of a PEDOT:PSS μ SEEG electrode after explantation from pig brain showing PEDOT:PSS covered by biomaterials.

To clarify the reason of apparent crack in parylene C PEDOT:PSS μ SEEG electrodes, we newly generated figures in the Supplementary Information including **Fig. R1-2** and updated sentences in **supplementary Fig. 9** and the main text as follows.

Changes to the manuscript:

Added figures, supplementary Fig. 9, Page 25

Supplementary Fig. 9. FEM analysis results showing (a) external strength and (b) deformation introduced by stylet insertion. OM images of the PEDOT:PSS/polyimide (c – e) and (f – h) PEDOT:PSS/parylene C electrodes before and after stylet insertion.

Added sentences, Supplementary Information, Page 8-9

Mechanical stress introduced by stylet insertion was investigated using finite element method (FEM) analysis (**supplementary Fig. 9a** and **9b**). The analysis indicates that the maximum strength applied to the films during the stylet insertion (0.11 MPa) are lower than the critical strength of the polyimide (1 – 2.5 MPa) and parylene C (0.6 MPa), as shown in **supplementary Fig. 9a**. Deformation of the films is lower than 4 μm, which can be considered as negligible from the scale of the electrodes, as shown in **supplementary Fig. 9b**. In theory, according to the simulation, the electrode should not sustain damage during stylet insertion. However, in practical scenarios, there are additional variables that can introduce stress to the electrode, such as misalignment of the electrodes in DI water (twisted or curved) and the angle at which the stylet is inserted, where stylet insertion may not be perfectly horizontal to the surface. While the polyimide layers are resilient and can withstand external stress arising from real-world conditions (**supplementary Fig 9c – e**), the parylene C layers appear to be approaching their critical limits (**supplementary Fig 9f – h**).

Updated and added sentences, Main text (Results), Page 7

We transitioned to all polyimide **PtNR** μ SEEG electrodes after we observed that parylene C **PEDOT:PSS** μ SEEG develop cracks in the parylene C layers and in the PEDOT:PSS layers after stylet insertion whereas polyimide μ SEEG did not suffer from any cracks. **The crack was caused by mechanical damage to the parylene C layers applied by the stylet during insertion, which then propagated to the PEDOT:PSS layer (supplementary Fig. 9).** Additionally, **PtNRs** contacts did not suffer any delamination from the μ SEEG whereas PEDOT:PSS suffered from delamination after stylet insertion in a substantial subset of electrodes, therefore reducing product yield.

Comment R1-6) Motion in humans

Are the μ SEEG flexible enough to move with the brain or are there still significant motion artifacts? Can human cortical lamina be resolved? If yes this would be a significant contribution/capability of the device.

Response R1-6:

We appreciate the reviewer's comment. We have not observed significant motion artifacts in our recordings with different species, including rat, pig, NHP, and human. This is because our device is not only flexible but also lighter (most of the weight comes from the stylet, which was removed following implantation and prior to recording) than clinical electrodes. For the depth μ SEEG, as we observed single unit activity on the same channels for the duration of the recording, we were confident the brain movement was minimal relative to the electrode. In the case of the shorter μ SEEG recorded in humans, we were able to place the electrode and observed the lighter material move with the brain tissue (which we also confirmed with recorded video; **supplementary Video 1**). Therefore, the device appears to move with the brain tissue movement.

Changes to the manuscript:

Added Supplementary Video 1 and text in Supplementary Information Page 51

Supplementary Video 1. Movement of the implanted short thin film μ SEEG relative to the brain. The electrode is shown before and after insertion, where the short depth was inserted into brain tissue and activity was recorded for a short period of time. The gauze and other stabilizing features are largely for the ribbon cables which form the connections, but the thin film components (which are largely transparent and flexible) move with the brain tissue movements in the craniotomy.

Reviewer: 2

The manuscript describes a new probe concept based on a polymeric substrate comprising an open channel / sheath realized using a sacrificial layer. The probe is inserted into deeper brain areas using a stylet inserted into the sheath. This stylet renders the probe temporarily stiff facilitating insertion into deeper brain areas. The stiffening is attributed to an increased probe cross section which might collapse to the initial thickness of the polymeric substrate layers after stylet retraction (which has not been demonstrated or better which is not obvious from the figures presented, e.g. Suppl. Fig. 6). In contrast to insertion procedures which rely on a stiffer wire hooked to the probe (see among others Luan et al. doi 10.1126/sciadv.1601966), the probe shape described in this manuscript is smooth without any wire tip protruding from the implantation hole at the probe tip as used by Luan et al. It has however to be taken into account that the stylet sheath represents an open lumen connecting the brain surface to deeper brain areas (see Fig. 1(f,g) and Suppl. Fig 2).

The authors provide an impressive amount of experimental data recorded in different species from rat, pigs, non-human primates and human cortex (due to clinical approval of limited implantation depth and only as acute recording experiments). It seems that really deep recordings have not been demonstrated. Further, the title of the manuscript suggests that mainly recordings in human cortex have been done. This is however a single experiment out of a variety of tests. I recommend to highlight the new probe that has merit for clinical applications.

The paper lacks long-term data (experiments have been done for up to 2 weeks only). I would be interesting to see which effects a fairly wide probe (from Suppl Fig 2f estimated to 1 mm in width) has on its long-term performance. In addition, the probe has an anisotropic stiffness and might even be stiffer than expected given the fact that the stylet causes a permanent deformation of the probe sheath (see Suppl Fig. 6).

The authors need to discuss the number of electrodes per probe as this is to a certain degree limited by the density of interconnection lines to be integrated. Please compare the probe presented in the manuscript with rolled solutions among others by Pothof et al. 2016 (F. Pothof, L. Bonini, M. Lanzilotto, A. Livi, L. Fogassi, G. Orban, O. Paul, and P. Ruther, "Chronic neural probe for simultaneous recording of single-unit, multi-unit, and local field potential activity from multiple brain sites," *Journal of neural engineering*, vol. 13, no. 4, p. 046006, 2016), Kang et al. 2015 (X. Kang, J.-Q. Liu, H. Tian, B. Yang, Y. Nuli, and C. Yang, "Self-closed parylene cuff electrode for peripheral nerve recording," *Journal of Microelectromechanical Systems*, vol. 24, no. 2, pp. 319–332, 2015), and van der Puije et al. 1989 (van der Puije P D, Pon C R and Robillard H 1989 Cylindrical cochlear electrode array for use in humans *Ann. Otolaryngology, Rhinology, Laryngology* 98 466–71).

In view of probe development and performance, the reviewers observed a couple of aspects that would request a further in-depth analysis and discussion – as further detailed below.

Response:

We thank the reviewer for the thoughtful comments and tremendously useful suggestions regarding papers, different types of probes, and helpful critiques. We will endeavor to discuss and compare this probe to the devices listed by the reviewer.

Comment R2-1: "The paper lacks long-term data (experiments have been done for up to 2 weeks only). I would be interesting to see which effects a fairly wide probe (from Suppl Fig 2f estimated to 1 mm in width) has on its long-term performance. In addition, the probe has an anisotropic stiffness and might even be stiffer than expected given the fact that the stylet causes a permanent deformation of the probe sheath (see Suppl Fig. 6)."

Response R2-1:

The chronic experiments were conducted for more than three weeks, as shown in Fig. 2k, using an electrode with a width of 1 mm. Additionally, we have attached a scanning electron microscope (SEM) image below (**Fig. R2-1**), showing our μ SEEG electrode after ten repetitions of stylet insertion and retraction. The image indicates negligible deformation after the stylet retraction, suggesting that the stylet insertion/retraction does not cause permanent deformation of the probe sheath. When the stylet is inserted through the sheath, the flexibility of the polyimide allows the μ SEEG electrode to adjust its shape, inflating it from two straight lines with a length of 1.24 mm (left image) to an oval shape with a longer length of 1.02 mm and a shorter length of 0.25 mm, which corresponds to the diameter of the stylet (right image). This suggests that the stress to the polyimide is relaxed as the electrode shape changes during stylet insertion.

Fig. R2-1. Scanning electron microscope (SEM) of the μ SEEG (left) after stylet retraction and (right) with the stylet inserted (same with **supplementary Fig. 2f**).

Moreover, DI water plays a critical role in reducing stress to the polyimide during stylet insertion. DI water not only inflates the sheath, but also reduces friction between the polyimide films and the stylet.

Changes to the manuscript:

To clarify the mechanical deformation issue from stylet insertion and retraction, we updated the manuscript as follows (**supplementary Fig. 2h & 2i**)

Added panels to Fig. 2, Supplementary Information, Page 17

Supplementary Fig. 2. Stylet insertion into the μ SEEG electrode. (a) Before insertion. ~~(b) Stylet aligned to the sheath.~~ (b) Stylet aligned to the sheath. Inset shows stitch holes with the stylet interweaved. (c) Zoom-in image of the sheath and the stylet inserted with support from tungsten probe for opening the sheath. (d) During insertion. (e) Zoomed-in image of the μ SEEG during stylet insertion. (f) Stylet insertion all the way to the tip. (g) Scale comparison with conventional SEEG. Cross-sectional SEM image of the (h) μ SEEG and (i) μ SEEG after stylet retraction. The tip of the μ SEEG electrode was deliberately opened by a razor blade to observe cross-sectional details of the μ SEEG electrode with bio-materials remained on. In (i), the μ SEEG deflated after stylet retraction. The dotted surface corresponds to the net layer on the back of the μ SEEG.

Added discussion, Supplementary Information, pages 5-6

The tip of the μ SEEG electrode was deliberately opened by a razor blade to observe cross-sectional details of the μ SEEG electrode with bio-materials remained on. **Supplementary Fig. 2i** displays a cross-section SEM image of the μ SEEG electrode after the stylet retraction, showing negligible deformation and suggesting that the stylet insertion and retraction does not cause permanent deformation of the probe sheath. During the stylet insertion, the flexibility of the polyimide allows the μ SEEG electrode to adjust its shape, inflating it from two straight lines with a length of 1.24 mm to an oval shape with a longer length of 1.02 mm and a shorter length of 0.25 mm, which corresponds to the diameter of the stylet. This indicates that the stress on the polyimide is relaxed as the electrode shape changes during stylet insertion. This is also observed under optical microscope (OM) image (**supplementary Fig. 2e**) showing different light reflection from the edge of the electrode depending on whether the stylet has already been implanted or is yet to be implanted.

Comment R2-2: “The authors need to discuss the number of electrodes per probe as this is to a certain degree limited by the density of interconnection lines to be integrated. Please compare the probe presented in the manuscript with rolled solutions among others by Pothof et al. 2016 (F. Pothof, L. Bonini, M. Lanzilotto, A. Livi, L. Fogassi, G. Orban, O. Paul, and P. Ruther, “Chronic neural probe for simultaneous recording of single-unit, multi-unit, and local field potential activity from multiple brain sites,” *Journal of neural engineering*, vol. 13, no. 4, p. 046006, 2016), Kang et al. 2015 (X. Kang, J.-Q. Liu, H. Tian, B. Yang, Y. Nuli, and C. Yang, “Self-closed parylene cuff electrode for peripheral nerve recording,” *Journal of Microelectromechanical Systems*, vol. 24, no. 2, pp. 319–332, 2015), and van der Puije et al. 1989 (van der Puije P D, Pon C R and Robillard H 1989 Cylindrical cochlear electrode array for use in humans *Ann. Otolaryngology, Rhinology, Laryngology* 98 466–71).”

Response R2-2:

We appreciate the reviewer for suggesting comparing our electrode with previously reported electrodes with rolled solutions and for providing a list of previous reports. We have made a table to discuss the electrode form factors and its application to the subject as you can see in **Table R1**.

	Total channel count	Metal lead spacing	Electrode site size & material	Impedance (kΩ) @ 1 kHz	Probe material	Probe shape	Subject, implant site

van der Puije et al. (1989)	9	N/A	890 μm x (1370 ~ 2180 μm), Pt	~ 1	Polyimide/silicone rubber	Rolled (cylindrical)	N/A (benchtop)
Kang et al. (2015)	20	N/A	D: 100 μm , EIROF on Pt	1.941 - 2.026	Parylene	Rolled (self-closed)	Rat. sciatic nerve
Pothof et al. (2016)	32, 64 (prototype: 128)	N/A	D: 35 μm , Pt	354 \pm 31	Polyimide	Rolled (cylindrical)	NHP, cortex
This work (2023), PEDOT	32, 64	3 μm	D: 20 μm , PEDOT on Pt	35.0 \pm 3.7	Parylene, polyimide	Inflatable sheath	Rat/NHP/pig/human, cortex
This work (2023), PtNR	32, 128	3 μm	D: 30 μm , PtNR	94.0 \pm 2.6	Polyimide	Inflatable sheath	Rat/NHP, cortex

Table R1. Comparison of the electrode form factor with previously reported studies using rolled solutions. (D: diameter, Parylene: Parylene C, EIROF: electrodeposited iridium oxide film, PEDOT: PEDOT:PSS, PtNR: platinum nanorods.)

To provide better understanding in novelty of our electrode, we updated the manuscript as follow (Table R1).

Changes to the manuscript:

Added discussion, Main text (Results) Page 4

Above the second polyimide layer, we deposited and patterned the metal trace layer with 10 nm/250 nm chromium/gold stack (deposited and patterned twice for a total trace thickness of 520 nm) followed by the deposition of a film of platinum-silver alloy used for the formation of PtNR contacts. Both the width and spacing of the metal trace are 3 μm . As it approaches the connectorization where the PCB was attached, the width of the metal lead expands to 20 μm and its spacing becomes 5 μm . After metallization, a film of platinum-silver alloy was deposited to form the PtNR contacts. The μSEEG electrode has advantages in its form factors including channel count, contact size, and impedance over the previously reported electrodes due to advanced MEMS technique employed in the fabrication process (supplementary table 1)²⁶⁻²⁹.

Added Supplementary Table 1, Supplementary Information Page 52

Supplementary Table 1. Comparison of the electrode form factor with previously reported studies using rolled solutions. (D: diameter, Parylene: Parylene C, EIROF: electrodeposited iridium oxide film, PEDOT: PEDOT:PSS, PtNR: platinum nanorods.)

	Total channel count	Metal lead spacing	Electrode site size & material	Impedance (kΩ) @ 1 kHz	Probe material	Probe shape	Subject, implant site
van der Puije et al. (1989)	9	N/A	890 μm x (1370 ~ 2180 μm), Pt	~ 1	Polyimide/silicone rubber	Rolled (cylindrical)	N/A (benchtop)
Kang et al. (2015)	20	N/A	D: 100 μm , EIROF on Pt	1.941 - 2.026	Parylene	Rolled (self-closed)	Rat. sciatic nerve

Pothof et al. (2016)	32, 64 (prototype: 128)	N/A	D: 35 μm, Pt	354 \pm 31	Polyimide	Rolled (cylindrical)	NHP, cortex
This work (2023), PEDOT	32, 64	3 μm	D: 20 μm, PEDOT on Pt	35.0 \pm 3.7	Parylene, polyimide	Inflatable sheath	Rat/NHP/pig/human, cortex
This work (2023), PtNR	32, 128	3 μm	D: 30 μm, PtNR	94.0 \pm 2.6	Polyimide	Inflatable sheath	Rat/NHP, cortex

Comment R2-3: The term reconfigurability needs to be defined. I would rather mention that probe can be used across several species with adequate design modifications. This aspect is however questionable at least for rat experiments given the large probe width of 1 mm.

Response R2-3:

We appreciate the reviewer's comment. We have modified the term "reconfigurable" in the abstract to "customizable". While we can reduce the probe width to less than 1 mm, we used a 1 mm wide probe to include 64 channels with some margins to ensure proper adherence of the polyimide films at the interface, in contrast to the sheath region. Additionally, we demonstrated chronic implantation and recording sessions from rats for more than 3 weeks with the 1 mm wide probe. Some previous studies have also shown successful implantations into the rat brain using a 1 mm wide probe (Pimenta et al. "Double-layer flexible neural probe with closely spaced electrodes for high-density in vivo brain recordings." *Frontiers in Neuroscience* 15 (2021): 663174/ Goshi et al. "Glassy carbon MEMS for novel origami-styled 3D integrated intracortical and epicortical neural probes." *Journal of Micromechanics and Microengineering* 28.6 (2018): 065009).

Changes to the manuscript:

Modified sentence, Abstract, Page 1

Abstract: Over the past decade, stereotactically placed electrodes have become the gold standard for deep brain recording and stimulation for a wide variety of neurological and psychiatric diseases. Current electrodes, however, are limited in their spatial resolution and ability to record from small populations of neurons, let alone individual neurons. Here, we report on a novel, **reconfigurable customizable**, monolithically integrated human-grade flexible depth electrode capable of recording from up to 128 channels and able to record at a depth of 10 cm in brain tissue.

Comment R2-4: The authors mention clinical neurotechnology. Potential difficulties in the approval process should be discussed as the probe deviates completely from clinically approved cylindrical probes; how likely is it that this specific probe is approved taking into account that it might act like a blade when forces act parallel to the surface of the fabrication while the probe is highly bendable for forces in the orthogonal direction; this anisotropic bending compliance needs to be discussed.

Response R2-4:

We appreciate the valuable feedback provided by the reviewer. Our team has taken great care in designing the μ SEEG device with smooth edges to prevent unintentional cutting of the brain and instead introduce rotation when shear stress is applied. We believe that due to the flexibility of the electrode, the device outside of the brain will be rotated, resulting in minimal damage to the brain. We have not observed such an effect or any complications during the use of the μ SEEG across species.

Furthermore, we are actively developing customized anchor bolts to further enhance the stability of the device and minimize any potential damage caused by unintentional electrode movement during clinical trials. Our goal is to ensure that the device is properly fixed in position during diagnosis and treatment, similar to clinical electrodes, in order to maximize the accuracy and safety of our device. As is deemed necessary for clinical trials, we have a plan to do implantation tests in a large animal model for up to 30 days as part of our anticipated regulatory approval pathway for the μ SEEG.

Specific comments

Comment R2-5: Line 60: the SEEG probes by DIXI, Adtech, PMT Corporation have cylindrical electrodes with heights between 1.5 mm and 3 mm

Response R2-5:

We appreciate the reviewer's comment and apologize for the wrong information. We modified the numbers in the manuscript as you can see below.

Changes to the manuscript:

Added discussion, Main text (Introduction), Page 2

Currently, arrays of electrodes are hand-assembled 0.8-1.27 mm diameter cylinders comprised of 8-16 contacts each ~~3~~1.5-5 mm in length.

Comment R2-6: Line 79 – NeuroPixels currently plans for probes as long as 40 mm; ATLAS and Cambridge NeuroTech presented silicon-based probe beyond 40 mm in length.

Response R2-6:

We have edited the statement and included a recent preprint referring to the longer Neuropixels probe.

Changes to the manuscript:

Added discussion and reference, Main text (Introduction), Page 2

They are also currently only able to access superficial cortical layers of the brain in humans, though there are advances in these devices enabling recording from deeper structures used in non-human primates²⁵.

Comment R2-7: Line 84 – Pothof et al presented small electrodes (30 μ m in dia.) on their cylindrical SEEG probe capable of recording single unit activity over ca. 50 days

Response R2-7:

We appreciate the reviewer's comment. The paper presented by Pothof et al. shows a well-established rolling technique to transform a thin-film electrode into a depth electrode and successfully demonstrated long-term and high-quality chronic recording data from microelectrodes. However, due to the nature of its fabrication, it still has limitations in customizing the electrode reproducibly, including the total diameter or length of the electrode, because the starting diameter is predetermined.

To highlight the significant achievements of previous papers, we updated the manuscript as follows:

Changes to the manuscript:

Added discussion, Main text (Introduction), Page 2

To increase the spatial resolution and channel count of electrodes that can record from either the lateral grey matter or deep brain structures, recent engineering approaches have focused on rolling or adhering conformable and photolithographically defined polyimide electrodes around or on medical-grade tubing used in clinical depth electrodes²⁵⁻²⁸. Previous research has well-established the transformation of thin-film electrodes to depth electrodes and demonstrated successful high-quality recording chronically²⁸. However, these hybrid integration approaches impose a limitation on the size of the electrode such that the starting diameter is pre-determined by the clinical depth electrode diameter.

Manufacturing

Comment R2-8: Line 124 – Here, I would already explain the purpose of these circular openings in the Ti-based hard mask.

Response:

We appreciate the reviewer's comment and suggestion. The Ti hard mask allows us to etch the net layer on the back of the electrode without doing additional lithography step once the top layers of the electrode fabrication are nearly completed and the electrode is released from the substrate and flipped so that this net layer can be etched using the Ti hard mask.

To facilitate a better understanding for the fabrication flow, we updated the manuscript as follows, as suggested by the reviewer.

Changes to the manuscript:

Added sentences, Main text (Results) Page 3

To fabricate μ SEEG electrodes, we first coated the glass substrate with a sacrificial polyimide layer and followed by the deposition of titanium etch-mask layer that was patterned with circular openings (**supplementary Fig. 1**). The circular openings were used to etch the 1st PI layer that were deposited on top of this layer. At the end of the fabrication process, the entire device layer was flipped, and O₂ plasma was used to create an array of holes in the 1st PI layer without the need for an additional lithography step. During this process, the titanium etch-mask layer protected the first polyimide layer, except for the circular openings. We then coated the glass substrate with two polyimide layers (1st and 2nd PI layer) and an interleaved Ti sacrificial layer (**Fig. 1c**).

Comment R2-9: Line 133 – just mention that the holes introduced into the 3rd PI layer define the electrodes

Response R2-9:

We appreciate the reviewer's comment and suggestion.

To facilitate a better understanding for the fabrication flow, we updated the manuscript as follows, as suggested by the reviewer.

Changes to the manuscript:

Updated sentences, Main text (Results) Page 4

This was followed by a top-most polyimide layer (3rd PI layer, **Fig. 1c**) coating. The next step involved exposing the microcontacts and defining the electrodes. To achieve this, we induced holes in the 3rd PI layer to expose the platinum-silver alloy films. Simultaneously, we etched

the shape of the electrode and additional larger holes (**Fig. 1d**) into the polyimide layer for mechanical stabilization of the stylet. ~~We next induced holes in the 3rd PI layer to expose the platinum-silver alloy films. The shape of the electrode and additional larger holes (Fig. 1d) were then etched into the polyimide layer for mechanical stabilization of the stylet.~~

Comment R2-10: Line 137 – how is the probe temporarily adhered to another glass substrate? How good is the thermal contact between polyimide probe and the glass to avoid excessive heating during plasma etching.

Response R2-10:

We temporarily adhered the flipped electrode onto another glass substrate with DI water, then removed the excess DI water using cleanroom wipes. To prevent electrode delamination, we then attached 2-mm-wide and 5-mm-long Kapton tape pieces along the edge of the electrode (one piece for every 3-4 cm) and performed O₂ plasma etching. After completing the plasma etching, we immersed the electrodes in an acetone bath and gently removed the Kapton tape pieces. Since the bonding of the probe onto another glass substrate is temporal, we didn't investigate the thermal contact, but the polyimide layers were strong enough to endure heating during plasma etching because the glass transition temperature of polyimide (> 350 °C) should be higher than the heat generated during O₂ plasma. We have had no failures in the fabrication during this step.

Comment R2-11: Line 141 – Grammar – ... openings are drilled through the 1st PI layer ** to expose ** the titanium ...

Response R2-11:

We appreciate the reviewer's comment and apologize for any confusion.

To clarify what we intend to describe, we modified the sentence as you can see below.

Changes to the manuscript:

Updated sentences, Result, Page 4

Continuation of the O₂ plasma etching through the circular openings drilled through the 1st PI layer ~~and exposed the titanium sacrificial layer underneath, which exposed the titanium sacrificial layer underneath.~~

Comment R2-12: Line 146 – Suppl Figure 2 does not really illustrate the insertion of the stylet; a schematic representation would be good to have; the photographs should be made at higher magnification

Response R2-12:

We agree with the reviewer that a higher magnification image will provide better information about the stylet insertion. We prepared higher magnification optical microscope images of the insertion of the stylet as you can see in **Fig. R2-2**.

Fig. R2-2. Optical microscope image of the μ SEEG electrode during stylet insertion.

To provide better understanding of the stylet insertion procedure to the readers, we have updated the manuscript as follows (**supplementary Fig. 2e**)

Changes to the Manuscript

Added panels to supplementary Fig. 2, Supplementary Information, Page 17

supplementary Fig. 2. Stylet insertion into the μ SEEG electrode. (a) Before insertion. ~~(b) Stylet aligned to the sheath.~~ (b) Stylet aligned to the sheath. Inset shows stitch holes with the stylet interweaved. (c) Zoom-in image of the sheath and the stylet inserted with support from tungsten probe for opening the sheath. (d) During insertion. (e) Zoomed-in image of the μ SEEG during stylet insertion. (f) Stylet insertion all the way to the tip. (g) Scale comparison with conventional SEEG. Cross-sectional SEM image of the (h) μ SEEG and (i) μ SEEG after stylet retraction. The tip of the μ SEEG electrode was deliberately opened by a razor blade to observe cross-sectional details of the μ SEEG electrode with bio-materials remained on. In (i), the μ SEEG deflated after stylet retraction. The dotted surface corresponds to the net layer on the back of the μ SEEG.

Added further discussion to Supplementary Information, Pages 5-6

Subsequently, DI water was sprayed along the electrode strips to open the space between the stylet upper and lower layers. In a few seconds after DI water spraying, the stylet could be travelled along the space (**supplementary Fig. 2d and 2e**) and finally reached to the tip of

the electrodes (**supplementary Fig. 2f**). **supplementary Fig. 2g** shows scale of the μ SEEG electrode after the stylet insertion and comparison with the clinical electrode indicating that a length of the stylet inserted part of the μ SEEG electrode is long enough to access to the deep brain structure. A cross-section view SEM image (**supplementary Fig. 2h**) shows the stylet upper and bottom layers with the stylet in between after recordings from non-human primate (NHP). The tip of the μ SEEG electrode was deliberately opened by a razor blade to observe cross-sectional details of the μ SEEG electrode with bio-materials remained on. **supplementary Fig. 2i** displays a cross-section SEM image of the μ SEEG electrode after the stylet retraction, showing negligible deformation and suggesting that the stylet insertion and retraction does not cause permanent deformation of the probe sheath. During the stylet insertion, the flexibility of the polyimide allows the μ SEEG electrode to adjust its shape, inflating it from two straight lines with a length of 1.24 mm to an oval shape with a longer length of 1.02 mm and a shorter length of 0.25 mm, which corresponds to the diameter of the stylet. This indicates that the stress on the polyimide is relaxed as the electrode shape changes during stylet insertion. This is also observed under optical microscope (OM) image (**supplementary Fig. 2e**) showing different light reflection from the edge of the electrode depending on whether the stylet has already been implanted or is yet to be implanted. The dotted surface in (i) corresponds to the net layer on the back of the μ SEEG.

Comment R2-13: Line 150 – I like the clever concept of openings in the 1st PI layer to interlock PI layers 1 and 2

Response R2-13:

We appreciate the reviewer for the comment and for acknowledging the significance of the interlocking arrays.

Comment R2-14: Line 153 – rounded shape of the stylet – how do you insert the rounded stylet into the narrow gap defined by the sacrificial layer

Response R2-14:

We prepared an optical microscope image of the μ SEEG electrode and a tungsten probe (**Fig. R2-3**) to provide information on how the rounded stylet was inserted. The insertion process was initiated by opening the sheath with support from a tungsten probe. Once the sheath was opened, the stylet was slid into the sheath, and the tungsten probe was gently removed.

Fig. R2-3. OM image of the μ SEEG electrode during stylet insertion. Inset shows OM image of the tungsten probe.

To provide better understanding of the stylet insertion procedure to the readers, we have updated the manuscript as follows

Changes to the manuscript:

Added panel for Fig. 2c, Supplementary Information, Page 17

Supplementary Fig. 2. Stylet insertion into the μ SEEG electrode. (a) Before insertion. ~~(b) Stylet aligned to the sheath.~~ (b) Stylet aligned to the sheath. Inset shows stitch holes with the stylet interweaved. (c) Zoom-in image of the sheath and the stylet inserted with support from tungsten probe for opening the sheath. (d) During insertion. (e) Zoomed-in image of the μ SEEG during stylet insertion. (f) Stylet insertion all the way to the tip. (g) Scale comparison with conventional SEEG. Cross-sectional SEM image of the (h) μ SEEG and (i) μ SEEG after stylet retraction. The tip of the μ SEEG electrode was deliberately opened by a razor blade to observe cross-sectional details of the μ SEEG electrode with bio-materials remained on. In (i), the μ SEEG deflated after stylet retraction. The dotted surface corresponds to the net layer on the back of the μ SEEG.

Added text to Supplementary Information, Page 5

Then, the stylet was inserted through the stitch holes to restrict lateral movement of the stylet and the sheath of the electrode (**supplementary Fig. 2b**). **Supplementary Fig. 2c** shows a zoomed-in image of the sheath during stylet insertion. The insertion process was initiated by opening the sheath with support from a tungsten probe. Once the sheath was opened, the stylet was slid into the sheath, and the tungsten probe was gently removed.

Probe design

Comment R2-15: Authors should discuss the anisotropic stiffness of these probe, i.e. the probes are fairly flexible orthogonal to the wafer plane but act like a knife when bent parallel to the wafer plane of probe fabrication. This is even more relevant (see next point) when the probe shape is permanently deformed due to the stylet insertion.

Response R2-15:

We appreciate the reviewer's comment. We have taken your suggestion into consideration and made some changes to address the concerns raised. As mentioned earlier, we are currently developing customized anchor bolts to secure the electrode's position during clinical trials and minimize any potential damage due to unintentional electrode movement. These anchor bolts will be similar to those used in clinical electrodes.

Additionally, we have included SEM images of the μ SEEG after stylet retraction in the supplementary material (revised **supplementary Fig. 2h and 2i** as shown in above) to demonstrate that there is negligible permanent deformation due to the stylet insertion and retraction. We hope that these changes have adequately addressed your concerns and improved the clarity of our manuscript.

Comment R2-16: Authors should further illustrate whether the stylet insertion into the probe results in a permanent plastic deformation of the polyimide substrate – an aspect that is frequently observed with polyimide probes using a wire inserted into a hole at the probe tip (compare probes by Lan Luan et al doi 10.1126/sciadv.1601966). Looking at Figure 1, it is unclear whether the probe shape is again flat when the stylet is removed. Suppl Fig 2 shows the probe with stylet ; will the eye-shaped probe geometry collapse after stylet retraction?

Response R2-16:

We appreciate the reviewer's comment. Unlike the method described by Lan Luan et al., our method does not significantly alter the shape of the polyimide films during or after stylet insertion and retraction. As demonstrated in the SEM images included above, the geometry of our μ SEEG only collapses (return) from an eye-shape to a thin-film structure after stylet retraction.

Comment R2-17: Line 207: the figure caption of Fig 1 mentions holes to enable probe integrity. Is the adhesion between the finally applied polyimide layer that weak that this rivet-like structures are needed?

Response R2-17:

We thank the reviewer for the comment which made us recognize that we did not sufficiently emphasize the importance of interlocking holes. The sentence in the figure caption of Fig. 1 "Red arrow indicates holes to enable electrode integrity." was used to describe a concept of openings in the 1st PI layer to interlock 1st and 2nd PI layers. Without the interlocking hole arrays, the 1st and the 2nd PI layer was easily separated when a stylet was inserted in between (**Fig.**

R2-3a). With help of the interlocking hole arrays, strong adhesion between the 1st and the 2nd PI layers was formed, preventing layer separation during the stylet insertion (**Fig. R2-3b**).

Fig. R2-3. OM image of μ SEEG electrode tip after stylet insertion. (a) Without interlocking hole arrays and (b) with interlocking hole arrays. Without interlocking hole arrays, the PI layers were separated, and the stylet pierced the tip. Scale bars are 500 μ m.

Changes to the manuscript:

To provide a better description of the figure, we have updated the manuscript as follows.

Note that **supplementary Fig. 3** has changed to **supplementary Fig. 4** due to the newly added figures in the revised manuscript.

Updated figure caption, Main text (Results), Page 7

(**h** and **i**) Magnified OM images at the tip **f** front and **g** back layers. ~~Red arrow indicates holes to enable electrode integrity.~~ The red arrow indicates the micro-hole arrays that interlock the 1st and 2nd PI layers.

Updated sentences, Main text (Results), Page 4

As the 2nd PI layer is coated to fill these holes, the interface between the 1st and 2nd PI layers has effectively a larger surface area than a planar one and as a result, better adhesion between the 1st PI layer and the 2nd PI layer is established. The greater mechanical stability afforded by the array of holes prevents the stylet from piercing through the tip when the stylet reaches this interface (**supplementary Fig. 3**). The tip of the stylet is mechanically polished to a rounded shape to minimize damage during insertion (**supplementary Fig. 34**).

Added figure, Supplementary Information, Page 18

Supplementary Fig. 3. OM image of μ SEEG electrode tip after stylet insertion. (a) Without interlocking hole arrays and (b) with interlocking hole arrays. Without interlocking hole arrays, the PI layers were separated and the stylet pierced the tip. Scale bars are 500 μ m.

Comment R2-18: Line 228: Please motivate the different electrode diameters for the PEDOT:PSS and PtNR electrodes.

Response R2-18:

We appreciate the reviewer's comment. The different electrode diameters for PEDOT:PSS and PtNR originate from different fabrication methods. For PEDOT:PSS, electrodeposition of PEDOT:PSS was performed on the via-opened microcontacts. On the other hand, to fabricate the PtNR microcontacts, we first needed to deposit PtAg thin film alloys onto the photoresist pre-patterned using conventional photolithography techniques, and then lift-off the rest of the film alloys except for the microcontacts. For the photolithography process, we used thick negative resist ($\sim 6 \mu\text{m}$ thick, NR9-6000 PY) to ease the lift-off and introduce undercut as much as possible. After process optimization, we found that the minimum diameter of the PtAg that can be lifted-off quickly, easily, and reproducibly is between 20 – 30 μm . To ensure reproducible fabrication of the electrodes, we decided to use the diameter of the PtNR contacts to be 30 μm .

Changes to the manuscript:

To motivate the different electrode diameters for both materials, we have updated the manuscript as follows.

Note that **supplementary Fig. 9** and **supplementary Table 2** have changed to **supplementary Fig. 11** and **supplementary Table 3** due to the newly added figures and tables in the revised manuscript.

Added sentences, Main text (Results), Page 7

All designs used have microelectrode contacts (also called channels, each 30 μm contact diameter for PtNRs and 20 μm contact diameter for PEDOT:PSS) with a center to center spacing of 60 μm (**Fig. 1k, supplementary Fig. 912; supplementary table-2 3**). While the diameter of the microcontacts can be adjusted, we have found that a contact diameter of 30 μm provides the most reproducible and optimal results for PtNRs based on our most recent optimization efforts. Therefore, we decided to use 30 μm as a diameter of the PtNR electrodes.

Comment R2-19: Please mention the design specs, among other minimal line and width spacing of the metal tracks. It would be helpful to have these numbers in the main text and not just in the Supplementary Information.

Response R2-19:

We appreciate the reviewer's comment.

To introduce the design specs about metal leads, we have updated the manuscript as follows.

Added sentences, Main text (Results), Page 4

Above the second polyimide layer, we deposited and patterned the metal trace layer with 10 nm/250 nm chromium/gold stack (deposited and patterned twice for a total trace thickness of 520 nm). Both the width and spacing of the metal traces are 3 μm . As it approaches the connectorization where the PCB were attached, the width of the metal lead expands to 20 μm and its spacing becomes 5 μm . After metallization, a film of platinum-silver alloy was deposited to form the PtNR contacts.

Comment R2-20: The interface technology and the external instrumentation needs to be discussed. A reasonable approach has been presented by the Yoon Lab in Michigan (Park et al., doi 10.1109/TBME.2021.3093542)

Response R2-20:

We appreciate the reviewer's comments. The interface technology and the external instrumentation are the same with our previous paper³⁰ as you can see in **Fig. R2-4**. Depending on the channel counts of the electrode, we utilized 1 to 8 RHD2164 chips for impedance measurement and recordings.

Fig. R2-4. 1024 channels amplifier board populated with BGA connector and Intan RHD2164 chips. 8 SPI cables connect the amplifier board to the recording controller.

To provide the interface technology and the external instrumentation information, we updated the main text as below.

Added sentences, main text (Results), page 8

We also made a longer version designed for accessing deeper structures (simultaneously with lateral cortex) in larger animals. This long μ SEEG includes 128 microelectrode contacts at a 60 μ m center to center spacing along a recording length of 7.65 mm at the tip of the entire array. This configuration most closely resembles clinical depth electrodes (**Fig. 1, j and o; supplementary Fig. 9**) although the spacing of the contacts or the incorporation of contacts with diameters larger than 100 μ m in a future μ SEEG design can be varied for specific end use. Our custom acquisition board connects to a 1024-channel electrophysiology control system, provided by Intan Technologies LLC³⁰. Depending on the channel counts of the electrode, we utilized 1 to 8 RHD2164 chips for impedance measurements and for recordings.

Probe performance

Response R2-21: Line 177/178: The authors mention an increase in electrode impedance upon insertion of the stylet. It would be interesting whether the increase is related to the mechanical stress caused by the stylet insertion or whether this would happen anyhow exposing the electrodes to the saline solution. Please provide control experiments.

Response R2-21:

We appreciate the reviewer's comments. For PEDOT:PSS μ SEEG, it is possible that the PEDOT:PSS is slightly peeled off from Au pad during stylet insertion. This can be addressed if PEDOT:PSS deposition procedure is further optimized to enhance an adhesion between Au and PEDOT:PSS interface. However, for PtNR μ SEEG, we found very negligible changes during 10 times of repetitive stylet insertion and retraction (**Fig. 2-6c**) when PtNR μ SEEG was connected to most recent connectorization setup that we have developed for next studies. Impedance fluctuation seems to be attributed to the connectorization part, less likely from stylet insertion. (Please recall that PEDOT:PSS μ SEEG and PtNR μ SEEG have different connectorizations – anisotropic conductive films & ribbon cables for PEDOT:PSS μ SEEG, and Ag epoxy, PCB, and customized ironwood socket for PtNR μ SEEG). In contrast to ACF & ribbon cable connectorization, a connection interface can be slightly different every time the connection was made. This results in a few k Ω changes in impedance of functional channels. By latching between the bonding interface and the custom ironwood electronics socket (**Fig. R2-5**) to form robust electrical connections between PCBs and pins in the socket (Inset of **Fig. R2-5**), the Ag epoxy between PCB and pad of μ SEEG can be damaged. To show how the impedance fluctuated during each measurement, we measured impedance value of μ SEEG contacts at 1 kHz during 10 times of repetitive latching and unlatching. The measurements exhibited a few k Ω fluctuations in both mean and standard deviation values as well as functional channel counts varied from 110 to 120 channels between different measurements (**Fig. R2-6a**). In our newest configuration of the μ SEEG which is not used for experiments reported in this manuscript, we have modified connector setup to a one that doesn't require latching mechanism to form robust electrical connections with recording setup. We manufactured a new headstage PCB with two RHD2164 chips for 128 recording capabilities and 16 pads for stimulation (**Fig. R2-5**). The PCB also has two Omnetics connectors for stimulation and recording, respectively (the stimulation SPI cable will be connected to separate stimulation system). We also measured impedance of μ SEEG connected to the new connector setup at 1 kHz during 10 times of repetitive cable connection and disconnection (**Fig. R2-6b**) and 10 times of stylet retraction and insertion (**Fig. R2-6c**). The results exhibited only 1 k Ω change at 5th attempt (51 k Ω , 64 channels) during repetitive cable connections whereas other attempts show no changes at all (52 k Ω , 64 channels). However, there are negligible fluctuations in the standard deviation ranged from 18 to 20 k Ω (**Fig. R2-6b**). Meanwhile, we observed no fluctuations in both mean and standard deviation values during repetitive stylet

retraction and insertion studies ($52 \text{ k}\Omega \pm 16 \text{ k}\Omega$, 64 channels). These results indicate that stylet insertion doesn't introduce damage to the electrodes and minor impedance fluctuation seems to be attributed to the latching of the socket.

Fig. R2-5. Image of old and new connectors.

The old connector set up uses custom ironwood electronics socket that requires latching to form robust electrical connections between PCBs and pins in the socket.

Fig. R2-6. Impedance of μ SEEG at 1 kHz during 10 times of (a) repetitive latching & unlatching (old connector set up), (b) repetitive SPI cable connection, and (c) repetitive stylet retraction & insertion.

Response R2-22: Line 188: GFPA staining: the conclusion on biocompatibility is based on a single experiment in a mouse implanted for 14 days. In particular the increased probe width might cause increased long-term trauma.

Response R2-22:

We appreciated the reviewer's comment. In our revision, we include data from a new experiment with 4 rats per group, and 1-4 sections analyzed per animal, after implantation of the clinical and μ SEEG electrodes for 14 days. In these data, we demonstrate modest improvement over the clinical electrode in the amount of GFAP, and a trend towards an increase in the proximity of neurons to the lesion site.

Fig. R2-7. (a) average GFAP area, normalized to each animal (2-way ANOVA, $F(1,72) = 8.749$, $P=0.0042$; $N=4$ /electrode, 1-4 sections per animal). (b) Normalized NeuN+ density (2-way ANOVA, $F(1, 72) = 3.290$, $P=0.0739$, $N=4$ /electrode, 1-4 sections per animal).

Changes to the manuscript:

To provide additional information, we updated the main text and **supplementary Fig. 7** as shown below.

Note that **Supplementary Fig. 7** has changed to **supplementary Fig. 10** due to the newly added figures in the revised manuscript.

Updated figure, supplementary Fig. 10, Page 26 as follows:

Supplementary Fig. 10. GFAP and NeuN reactivity following two weeks of electrode implantation. Wide-field view of horizontal cortical section following (a) clinical or (b) μ SEEG electrode implantation. NeuN is pseudocolored green, GFAP magenta, and DAPI blue. (c) and (d) are magnifications of the boxed areas in (a) and (b) to show NeuN reactivity adjacent to electrode placement. Representative NeuN-positive cells are indicated by arrows. (e) Average number of NeuN+ cells per $100\ \mu\text{m}^2$ sampled randomly at 0–100, 100–200, and 200–300 μm from the electrode. (f) Average GFAP intensity of $100\ \mu\text{m}^2$ boxes sampled randomly at 0–100, 100–200, and 200–300 μm from the electrode. GFAP area, normalized to each animal (2-way ANOVA, $F(1,72) = 8.749$, $P=0.0042$; $N=4/\text{electrode}$, 1–4 sections per animal). (f) Normalized NeuN+ density (2-way ANOVA, $F(1, 72) = 3.290$, $P=0.0739$, $N=4/\text{electrode}$, 1–4 sections per animal).

Data are plotted as mean \pm SEM, **** $P < 0.0001$, ** $P < 0.01$ by 2-way ANOVA followed by Sidak's multiple comparisons test. Scale bars are $100\ \mu\text{m}$.

Updated sentences, Supplementary Information, Page 11

At the termination of the chronic recordings (by four weeks after implant), brains were removed and briefly (2 hrs) fixed in 4% PFA. The cortices were removed and flattened between glass slides in PBS, and then the flattened cortex further fixed in 4% PFA overnight, and then allowed to sink in 30% sucrose. For measuring implantation scar, rats were implanted with a dummy clinical electrode or μ SEEG and allowed to recover for two weeks ($n = 1 / \text{electrode}$). Animals were perfused with 4% PFA and then sunk in 30% sucrose. Whole brains or flattened cortices were sectioned on a cryostat in the horizontal plane from at $40\ \mu\text{m}$. Sections were washed with PBS and stained overnight with one or more of the following: chicken anti-NeuN (1:500, Sigma ABN91), Cy3-conjugated mouse anti-GFAP (1:1000, Sigma C9205), rabbit anti-VGLUT2 (1:500, Abcam ab216463). After wash, slices were incubated in secondaries of goat anti-chicken 488 (Sigma SAB4600039), goat anti-chicken 647 (Sigma SAB2600184), or goat anti-rabbit 488 (ThermoFisher A-11008) at 1:300 for 1 hour, rinsed, and mounted with ProLong with DAPI. Slides were imaged on a Nikon Eclipse Ti2-E with a DS-Q12 CMOS camera at the UCSD Nikon Imaging Core Images were analyzed using ImageJ (**supplementary Figs. 10a**

to d). In our observations from ~~one four~~ chronic (2 weeks) placement, ~~immunochemistry revealed NeuN-positive cells in close (< 20 μm) proximity to the μSEEG electrode and lower GFAP intensity and spread~~ we observed no significant difference in the number of Neun-positive cells surrounding the lesion between the μSEEG and the clinical grid, but there was a small non-significant improvement with the μSEEG electrode (2-way ANOVA, $F(1, 72) = 3.290$, $P = 0.0739$) (**supplementary Figs. 10e and 10f**).

Updated sentences, Main text (Results), Page 5

Finally, to test the amount of tissue damage caused by these devices, we implanted rats with one chronic μSEEG electrode with 1.89 mm recording length on one hemisphere and a clinical electrode on the other hemisphere for 14 days ($N = 4$ 4 electrode). Insertion of the μSEEG electrode resulted in decreased astrocyte scarring, as measured by significantly lower GFAP positive area intensity as compared to the clinical electrode as shown in (**supplementary Fig. 10**) (2-way ANOVA, $F(1,72) = 8.749$, $P=0.0042$). ~~Within 100 μm from the electrode, for example, GFAP intensity in five randomly placed 100 μm² boxes was significantly lower for the μSEEG electrode (1842 +/- 53 a.u.) compared to the clinical electrode (3840 +/- 339 a.u.; two-way ANOVA, $F(1,24) = 85.93$; $P < 0.0001$; Sidak's multiple comparisons posthoc, $p < 0.001$).~~ We observed no significant difference in the number of Neun-positive cells surrounding the lesion between the μSEEG and the clinical grid, but there was a small non-significant improvement with the μSEEG electrode (2-way ANOVA, $F(1, 72) = 3.290$, $P = 0.0739$).~~In addition, significantly fewer neurons were observed between 0-100 μm from the electrode for the clinical electrode compared to the μSEEG (2.4 +/- 1.2 vs. 19 +/- 2.3 cells; 2-way ANOVA, $F(1,24) = 23.26$, $P < 0.001$; Sidak's multiple comparisons posthoc, $p < 0.0001$).~~

Response R2-23: Line 195: PtNR surface seems to be unaffected by insertion into brain tissue. Is that stability potentially related to the recessed character of the electrodes? Please compare the electrode performance to electrodes based on Pt nanograss by Boehler et al. (doi 10.1021/acsami.9b22798)

Response R2-23:

We appreciate the reviewer for pointing out the recessed character of our electrodes and kind suggestion to compare our data with a previously reported paper. Similar to our previous work, (Ganji et al. "Selective formation of porous Pt nanorods for highly electrochemically efficient neural electrode interfaces." Nano letters 19.9 (2019): 6244-6254), our electrode design includes a recessed side wall by the virtue of the process: we need passivation layers for our metal traces and regardless of when the PtAg allow is deposited in the process, the height of the PtNRs will always be lower than the height of the sidewall opening in the passivation layer. This also protects the PtNRs during implantation. However, the morphology of Pt nanograss in a previous paper suggested by the reviewer appeared to be stable after 5 weeks of implantation despite not having a recessed wall, potentially due to the low mechanical stress of brain tissue on the nanograss. The recessed wall in this paper was only applied to the planar electrode, which was not used for the implantation. While comparing the mechanical stability of PtNRs and Pt nanograss is difficult due to their different morphology and crystallinity, it appears from our results that biomaterials are deposited on Pt nanostructures which helps to maintain their functionalities during long-term implantation, indicating their biocompatibility, robustness, and suitability for clinical translation.

Response R2-24: Lines 217-227: Cracks are seen in the PEDOT-PSS as well as parylene C layers. Are control experiments available that indicate the electrode stability upon immersion into saline solution? Or are the delamination and crack formation solely correlated to the stylet

insertion into the probe? Please provide FEM analysis investigating the mechanical stress in the different probe layers caused by stylet insertion.

Response R2-24:

We appreciate the reviewer for suggesting FEM analysis to investigate the mechanical stress caused by the stylet insertion. The delamination or crack of PEDOT:PSS was observed from parylene C electrodes after inserting a stylet in between two parylene C layers, which occurred due to mechanical damage applied to the parylene C by the stylet, when it was not properly inserted. The electrodes that had delamination or crack of PEDOT:PSS were abandoned and not used for the recording. For the recording, we only used the devices that did not have delamination or crack of PEDOT:PSS, which was confirmed by both optical microscope image and impedance measurement done in PBS (**Fig. R2-8**). In contrast, PEDOT:PSS was stable on polyimide electrodes showing no mechanical failures after the stylet insertion because the polyimide is more robust than parylene C. According to our additional study to measure the critical tensile strength of the parylene C, it was measured as 0.6 Mpa, which is smaller than that of the polyimide (1 – 2.5 Mpa). We also provide FEM analysis to investigate the mechanical stress caused by the stylet insertion, as the reviewer requested (**Fig. R2-9**). The analysis indicates that the maximum strength applied to the films during the stylet insertion (0.11 Mpa) are lower than the critical strength of the polyimide (1 – 2.5 Mpa) and parylene C (0.6 Mpa), and deformation of the films are lower than 4 μm , which can be considered as negligible from the scale of the electrodes. In theory, according to the simulation, the electrode should not sustain damage during stylet insertion. However, in practical scenarios, there are additional variables that can introduce stress to the electrode, such as misalignment of the electrodes in DI water (twisted or curved) and the angle at which the stylet is inserted, which may not be perfectly horizontal to the surface. While the polyimide layers are resilient and can withstand external stress arising from real-world conditions, the parylene C layers appear to be approaching their critical limits.

Fig. R2-8. OM images of PEDOT:PSS μ SEEG electrodes. Parylene C electrodes (a) before and (b) after stylet insertion and (c) its zoomed-in image. Polyimide electrodes (d) before and (e) after stylet insertion and (f) its zoomed-in image.

Fig. R2-9. FEM analysis showing the mechanical stress and deformation caused by the stylet insertion.

To clarify the reason for crack in parylene C PEDOT:PSS μ SEEG electrodes, we newly added the above figure in the Supplementary Information and updated sentences in the main text as follows.

Changes to the Manuscript:

Added figures, supplementary Fig. 9, Page 25

supplementary Fig. 9. FEM analysis results showing (a) external strength and (b) deformation introduced by stilet insertion. OM images of the PEDOT:PSS/polyimide (c – e) and (f – h) PEDOT:PSS/parylene C electrodes before and after stilet insertion.

Updated sentences, Supplementary Information, Page 8

We investigated tensile stress of μ SEEG using pull measurement to exam mechanical stability of our electrode and compare it with clinical depth electrode (PMT electrode). The μ SEEG was integrated on the pressure sensor system (voice coil-powered linear actuator system with internally integrated force sensor; V-275 PIMag Voice Coil Linear Actuator). A customized 3D printed sample mount was attached onto the actuator (**supplementary Fig. 45a**) and a tip of the μ SEEG was placed on the sample mount followed by strong fixation using epoxy. Rest of the electrode was attached on a main body of the system and fixed by epoxy as well (**supplementary Fig. 45b**). We applied force to the actuator to exam tensile strength upon the μ SEEG-electrodes break (indicated by red arrow) and obtained that the critical tensile strength of the μ SEEG is 1 Mpa (16 mN)–2.5 MPa (37 mN). The critical strength of parylene C μ SEEG and polyimide μ SEEG are 0.6Mpa (24 mN with an area of the stress of $4e-8$ m²) and 1 – 2.5 Mpa (16 – 37 mN with an area of the stress of $1.5e-8$ m²), respectively. Meanwhile, the PMT electrode that was anchored on two polyurethane tube regions around a Pt contact was damaged when 14 kPa (48 mN with an area of the stress of $3.5e^{-7}$ m²) was applied to the

PMT electrode (**supplementary Fig. 45c**). The illustration of the area of the stress of electrodes is illustrated in **supplementary Fig. 5d**.

Added sentences, Supplementary Information, Page 8

Mechanical stress introduced by stylet insertion was investigated using finite element method (FEM) analysis (**supplementary Fig. 9a** and **9b**). The analysis indicates that the maximum strength applied to the films during the stylet insertion (0.11 Mpa) are lower than the critical strength of the polyimide (1 – 2.5 Mpa) and parylene C (0.6 Mpa), as shown in **supplementary Fig. 9a**. Deformation of the films are lower than 4 μm , which can be considered as negligible from the scale of the electrodes, as shown in **supplementary Fig. 9b**. In theory, according to the simulation, the electrode should not sustain damage during stylet insertion. However, in practical scenarios, there are additional variables that can introduce stress to the electrode, such as misalignment of the electrodes in DI water (twisted or curved) and the angle at which the stylet is inserted, which may not be perfectly horizontal to the surface. While the polyimide layers are resilient and can withstand external stress arising from real-world conditions (**supplementary Fig. 9c – e**), the parylene C layers appear to be approaching their critical limits (**supplementary Fig. 9f – h**).

Updated and added sentences, Main text (Results), Page 7

We transitioned to all polyimide **PtNR** μSEEG electrodes after we observed that parylene C **PEDOT:PSS** μSEEG develop cracks in the parylene C layers and in the PEDOT:PSS layers after stylet insertion whereas polyimide μSEEG did not suffer from any cracks. **The crack was caused by mechanical damage to the parylene C layers applied by the stylet during insertion, which then propagated to the PEDOT:PSS layer (supplementary Fig. 9)**. Additionally, PtNRs contacts did not suffer any delamination from the μSEEG whereas PEDOT:PSS suffered from delamination after stylet insertion in a substantial subset of electrodes, therefore reducing product yield.

Comment R2-25: Please explain the large discrepancy in electrode impedance between ca. 35 kOhm (line 204) and up to 268 kOhm (line 280)

Response R2-25:

We appreciate the feedback provided by the reviewer. 35 kOhm is the impedance of the PEDOT:PSS device which was measured in PBS, where 268 kOhm is the impedance of the PtNR device which was measured post implant. Though the large discrepancy in the electrode impedance may be due to a significant coverage of biomaterials on the PtNRs, it is expected that the impedance in vivo to be significantly higher than that in saline as reported in our systematic studies elsewhere, primarily due to the difference of conductivity in the media and spreading resistances at the electrode-media interface [Frontiers in Neuroscience, 2022972252-1 - 2022972252-17, 2022]. Our additional experiments have shown that the impedance of explanted electrodes when measured in saline do not significantly increased when biomaterials do not cover the entire PtNRs, as confirmed by SEM measurements (please see **Fig. R2-10**).

Fig. R2-10. SEM image of the PtNRs after μ SEEG retraction from pig's brain.

Comment R2-26: Cross-talk between different electrodes / interconnection lines needs to be validated (line 489). It would be interesting whether this is due to a capacitive coupling inside the probe or caused by a certain degree of signal spreading inside brain tissue.

Response R2-26:

We appreciate the reviewer's comment. It is possible that we measured the same neuronal activities, given the narrow spacing of the electrodes (60 μ m). Additionally, we observed that the location of the reference contact also affects the amount of common signal captured. Moving forward, we will continue to optimize our recording setup to improve the recording quality.

To validate the cross-talk issue of our electrode, we newly added discussions in the main text as follows.

Updated and added sentences, Main text (Results), Page 17

One potential limitation of these designs is cross-talk amongst the channels. While we have not definitively quantified cross-talk in the recordings, we observed a strong common-mode signal on all contacts that we subtracted in order to delineate CSD dynamics. **One of possible reasons for the common-mode signal would be recording the same neural activities, given the narrow spacing of the electrodes (60 μ m). Additionally, we observed that the location of the reference contact also affects the amount of common signal captured.** If inter-channel cross-talk is a substantial issue, future designs could involve distributing the metal traces in separate polymer layers. Another limitation concerns connectorization. Current connectors do not match typical clinical standards. Improving the back-end of the devices is an area of active development. Further, the current design includes contacts facing only one direction along the electrode length. Future designs can involve developing multiply directional contact sampling.

Comment R2-27: Figure 1: Fig 1l shows the layer composition during probe fabrication not of the finally realized probe; this needs to be clarified.

Response R2-27:

We appreciate the reviewer's comment. We added 'will be removed' in Fig. 1c to clarify that this is not a final composition of the probe, and the sacrificial layer will be removed to make a hollow space between PI layers for stylet insertion. We updated our manuscript as follows.

Changes to the manuscript:

Updated figure, Fig. 1c, Page 6

· Supplementary information

Comment R2-28: Probe fabrication – information is somehow doubled; please rewrite to streamline the information that needs to be delivered; examples are the description of the sacrificial layer and the Ti-based hard mask (lines 107-117).

Response R2-28:

We apologize for the confusion. We corrected our manuscript to be clearer in the fabrication flow description.

Changes to the manuscript:

Updated sentences, Supplementary Information, Page 3

Supplementary Fig. 1 summarizes the fabrication process of the μ SEEG electrode. Polished and cleaned photomask-grade soda lime glass plates (Nanofilm) with dimensions of 7" \times 7" \times 0.06" were used as substrates for the fabrication. First, a Micro-90 (International Products Corporation) diluted with deionized (DI) water (0.1%) layer was spin-coated on the glass substrate as a release layer for the polyimide electrodes in the last step of the fabrication processes. Subsequently, a sacrificial 5- μ m-thick-polyimide (PI-2611 from HD Microsystems) was initially deposited by conventional spin-coating, soft-baking, and curing (340 °C for 3 hrs, 3 °C/min in N₂ ambient) processes. Then, a 60-nm-thick Ti hard mask for net layer formation was formed on the 4st sacrificial polyimide layer. The Ti hard mask was proceeded by a standard lithography, descum, metal deposition, and lift-off process using AZ5214E-IR photoresist (MicroChemicals), maskless photolithography system (Heidelberg MLA 150), UV flood exposure system (DYMAX), plasma etcher (Oxford Plasmalab 80), and e-beam evaporator (Temescal). The Ti hard mask contains via patterns with hole arrays with a diameter and a spacing of 5 μ m for both, and a 1.5 mm wide and 0.6 mm long rectangular shape for a sheath.

Then, a 1st polyimide layer was spun-coated, soft-baked, and cured at 330 °C for 3 hrs in N₂ ambient following pre-curing of the substrate to dry out any solvent or moisture trapped in the polyimide surface under 340 °C for 3 hrs in N₂ ambient. The 1st polyimide layer will role as the sacrificial bottom layer with net of holes in the final product. Then, a 60-nm-thick Ti sacrificial layer was deposited using same method for the Ti hard mask preparation. The Ti sacrificial layer also acts as etch-stop layer during the final net formation etching step using O₂ plasma. To increase adhesion between the 1st and a 2nd polyimide layer, hole arrays with a diameter and a spacing of 5 μ m for both are patterned along the Ti sacrificial layer using AZ 1518 (MicroChemicals). Then, the 1st polyimide layer was selectively etched by O₂ plasma (50 mTorr, 200 W for 7 min), and the photoresist layer was removed by solvent cleaning processes.

Comment R2-29: Probe fabrication – naming of the different PI layers should be streamlined, i.e. there is an initial sacrificial PI layer on which a hard mask is deposited and patterned; this is followed by a 1st probe PI layer on which a sacrificial Ti layer is deposited and patterned; the Ti layer also serves as an etch stop when patterning etch holes into the 1st PI layer, to ultimately etch the sacrificial Ti layer...

Response R2-29:

We apologize for the confusion. We updated our manuscript as follows as the reviewer suggested.

Updated sentences, Supplementary Information, Page 3

Supplementary Fig. 1 summarizes the fabrication process of the μ SEEG electrode. Polished and cleaned photomask-grade soda lime glass plates (Nanofilm) with dimensions of 7" \times 7" \times 0.06" were used as substrates for the fabrication. First, a Micro-90 (International Products Corporation) diluted with deionized (DI) water (0.1%) layer was spin-coated on the glass substrate as a release layer for the polyimide electrodes in the last step of the fabrication processes. Subsequently, a sacrificial 5- μ m-thick-polyimide (PI-2611 from HD Microsystems) was initially deposited by conventional spin-coating, soft-baking, and curing (340 °C for 3 hrs, 3 °C/min in N₂ ambient) processes. **This sacrificial PI layer will serve to separate the device layers, which will be constructed on top of this layer, from the glass plate.** Then, a 60-nm-thick

Ti hard mask for net layer formation (Ti_{net}) was formed on the sacrificial polyimide layer. The ~~Ti hard mask~~ Ti_{net} was proceeded by a standard lithography, descum, metal deposition, and lift-off process using AZ5214E-IR photoresist (MicroChemicals), maskless photolithography system (Heidelberg MLA 150), UV flood exposure system (DYMAX), plasma etcher (Oxford Plasmalab 80), and e-beam evaporator (Temescal). The Ti hard mask contains via patterns with hole arrays with a diameter and a spacing of 5 μm for both, and a 1.5 mm wide and 0.6 mm long rectangular shape for a sheath.

Then, ~~another a~~ 4^{st} polyimide layer was spun-coated, soft-baked, and cured at 330 $^{\circ}\text{C}$ for 3 hrs in N_2 ambient following pre-curing of the substrate to dry out any solvent or moisture trapped in the polyimide surface under 340 $^{\circ}\text{C}$ for 3 hrs in N_2 ambient. This 4^{st} polyimide layer is the 1^{st} device layer (1^{st} PI layer) which will be role as the sacrificial bottom layer with net of holes in the final product. Then, a 60-nm-thick Ti sacrificial layer ($Ti_{sacrificial}$) was deposited using same method for the ~~Ti hard mask~~ Ti_{net} preparation. The $Ti_{sacrificial}$ ~~sacrificial layer~~ also acts as etch-stop layer during the final net formation etching step using O_2 plasma. To increase adhesion between the 1^{st} PI layer and ~~another upcoming a~~ 2^{nd} polyimide layer (2^{nd} PI layer), hole arrays with a diameter and a spacing of 5 μm for both are patterned along the $Ti_{sacrificial}$ ~~sacrificial layer~~ using AZ 1518 (MicroChemicals). Then, the 1^{st} PI layer ~~polyimide layer~~ was selectively etched by O_2 plasma (50 mTorr, 200 W for 7 min), and the photoresist layer was removed by solvent cleaning processes.

After processes on the 1^{st} PI layer ~~polyimide layer~~, the glass substrate was baked at 330 $^{\circ}\text{C}$ for 2 hrs in N_2 ambient. Then, a 2^{nd} PI layer ~~polyimide layer~~ was coated and cured under 320 $^{\circ}\text{C}$ for 2 hrs in N_2 ambient. Subsequently, metal traces with a width and a spacing of 3 μm for both were formed on the 2^{nd} PI layer ~~polyimide layer~~. Metal traces were composed of Cr/Au (10/250 nm) and the entire lithography, deposition, and lift-off process was repeated on top of the first metal lead layer to form Cr/Au/Cr/Au (10/250/10/250 nm) traces. This double-patterning process was employed to increase yield and reduce risk of photoresist particles from compromising the thin traces.

After the double-layer metal leads formation, a 30 μm -diameter-PtAg alloy was formed selectively on the individual micro-contact recording sites by photolithography, descum, and PtAg alloy co-sputtering using the maskless photolithography system with NR9-6000 (Futurrex) photoresist, plasma etcher, and DC/RF magnetron sputter (Denton Discovery 18), respectively. A 60-nm-thick Ti capping layer (Ti_{cap}) was deposited on top of PtAg alloys to prevent oxidation in air or under oxygen plasma in the following processes. The detailed fabrication methods and characteristics of PtNRs can be found elsewhere^{29, 30}. Notably, this process involves a selective etching of silver in a dealloying process, leaving behind non-toxic platinum.

A 3^{rd} polyimide layer (3^{rd} PI layer) was then coated and cured under 300 $^{\circ}\text{C}$ for 1 hr in N_2 ambient. On top of the 3^{rd} PI layer ~~polyimide layer~~, a 60-nm-thick-Ti hard mask ($Ti_{outline}$) was deposited and AZ5214E-IR photoresist was coated on top of the $Ti_{outline}$ ~~layer~~ and patterned to define the outline of the electrodes. Then, SF_6/Ar plasma was used to selectively etch the $Ti_{outline}$ ~~layer~~ through the photoresist. To accomplish a 1^{st} deep etch step, 3 hrs of O_2 plasma was performed to etch 3^{rd} , 2^{nd} and 1^{st} PI ~~polyimide~~ layers until exposing the Ti_{net} ~~hard mask layer~~ on the sacrificial PI ~~polyimide~~ layer.

Following the 1^{st} deep etch process, a-8- μm -thick AZ12XT-20PL-10 (MicroChemicals) was spun-coated and patterned to open via holes for recording sites and printed circuit board (PCB) contact pads, and the outline of the electrodes. Then, SF_6/Ar plasma was used to etch Ti_{net} ~~hard mask layer~~ and expose the surface of the 3^{rd} PI layer ~~polyimide layer~~ on the recording

sites and the sacrificial PI polyimide layers along the outline of the electrodes. Subsequently, the exposed PI polyimide layers were etched by O₂ plasma for 1 hr.

To protect PI polyimide layers from dealloying process to form PtNRs using nitric acid, a 100-nm-Ti (Ti_{passivation}) was conformally deposited using sputtering system and a 1- μ m-thick parylene C was additionally coated using a parylene deposition system (Specialty Coating Systems 2010 Labcoter). To open via holes for recording sites, a 8- μ m-thick AZ12XT-20PL-10 was spin-coated and patterned again. Then, O₂ plasma and subsequent SF₆/Ar plasma was used to etch the parylene C and Ti layers including the Ti_{passivation} passivation and Ti_{cap} cap layers to expose the PtAg alloys. The glass substrate was immersed into 60 °C nitric acid for 2 min for dealloying of PtAg alloys and rinsed in de-ionized (DI) water. The microscopic morphology of the PtNR is shown in the scanning electron microscope (SEM) images (**Fig. 1b**).

The parylene C and Ti_{passivation} passivation layers were removed by O₂ plasma and 6:1 buffered oxide etchant (BOE), respectively. Then, the glass substrate was immersed in DI water to delaminate the electrodes from the glass substrate with the dissolution of the underlying Micro-90 layer.

The delaminated electrode was then flipped and transferred onto a carrier glass wafer. Following the flip transfer, the net of hole arrays was formed on the 1st PI polyimide layer by O₂ plasma removing the sacrificial PI polyimide layer and selectively etch the 1st PI polyimide layer through the via openings in the Ti_{net} hard-mask. The etching time was set longer to obtain clean opening of hole arrays. Note that the 2nd PI polyimide layer was protected, during the over etching procedure, by the Ti_{sacrificial} sacrificial layer prepared between 1st and 2nd PI polyimide layers. Lastly, the Ti_{net} and Ti_{sacrificial} hard-mask and the sacrificial layers were all dissolved by BOE and the electrode was immersed in DI water for ~6 hrs to rinse any residual BOE.

Updated figure, supplementary Fig. 1, Page 16

Supplementary Fig. 1. Fabrication process of μ SEEG electrode.

Comment R2-30: Probe fabrication – is the sacrificial Ti layer really just 60 nm thick? How long do you need to etch this layer?

Response R2-30:

The thickness of the sacrificial Ti layer we use is 60 nm. We only need a thickness that is enough to avoid bonding of the 1st and the 2nd polyimide layer during the curing process of the 2nd polyimide layer. At the same time, it has to be thin so that it can be easily etched by buffered oxide etchant (BOE). BOE wet etching of the sacrificial layer took ~ 2 hrs to make sure all Ti residues were dissolved. This is facilitated by the presence of a large number of net layer holes on the back of the electrode that facilitates the etching by BOE, which otherwise, would have needed a much thicker Ti layer as the reviewer may be eluding to.

Comment R2-31: Probe fabrication – line 132 – why is the Ti cap layer needed as the Ag has already been etched (Keundong)

Response R2-31:

We apologize for the confusion. Ti cap layer is deposited after the PtAg alloys deposition step. We would like to note that the PtAg alloys are not dealloyed at this step. They are dealloyed after the 2nd deep etching step as introduced in **supplementary Fig.1**.

During the 2nd deep etching process, Ag in PtAg alloys may be oxidized so there is a chance that the dealloying process would not be ideal to remove Ag in PtAg alloys. To prevent any

oxidation of Ag, we deposit Ti cap layer right after PtAg alloys deposition to protect any oxidation in PtAg alloys.

Changes to the manuscript

We recognize our description may give confusion to the readers. To clarify this process, we have updated the manuscript as follows.

Added sentences, Supplementary Information, Page 4

After the double-layer metal leads formation, a 30 μm -diameter-PtAg alloy was formed selectively on the individual micro-contact recording sites by photolithography, descum, and PtAg alloy co-sputtering using the maskless photolithography system with NR9-6000 (Futurrex) photoresist, plasma etcher, and DC/RF magnetron sputter (Denton Discovery 18), respectively. A 60-nm-thick Ti capping layer was deposited on top of PtAg alloys to prevent oxidation in air or under oxygen plasma in the following processes. The detailed fabrication methods and characteristics of PtNRs can be found elsewhere^{29, 30}. Notably, this process involves a selective etching of silver in a dealloying process, leaving behind non-toxic platinum.

Comment R2-32: Stylet insertion – Suppl Fig 2 requests more details zooming into the position of the threading hole

Response R2-32:

We appreciate the reviewer's suggestion for including zoomed-in image of the threading hole. To ensure that the stylet is aligned with the center of the μSEEG , the center of the stitch hole arrays is aligned with the center of the sheath. The diameter of each hole is 800 μm , and they are spaced 2 mm apart. We have included a zoomed-in image to show more details regarding the position of the threading holes (referred as stitch holes in the Supplementary Information) as follows.

Fig. R2-11. OM image of the stitch holes with the stylet interweaved.

Changes to the Manuscript:

Updated figure, Supplementary Information, Page 17

Supplementary Fig. 2. Stylet insertion into the μ SEEG electrode. (a) Before insertion. (b) Stylet aligned to the sheath. Inset shows stitch holes with the stylet interweaved. (c) Zoom-in image of the sheath and the stylet inserted with support from tungsten probe for opening the sheath. (d) During insertion. (e) Zoomed-in image of the μ SEEG during stylet insertion. (f) Stylet insertion all the way to the tip. (g) Scale comparison with conventional SEEG. Cross-sectional SEM image of the (h) μ SEEG and (i) μ SEEG after stylet retraction.

Added sentences, Supplementary Information, Page 5

Then, the stylet was inserted through the stitch holes to restrict lateral movement of the stylet and the sheath of the electrode (**supplementary Fig. 2b**). To ensure that the stylet is aligned with the center of the μ SEEG, the center of the stitch hole arrays is aligned with the center of the sheath. The diameter of each hole is 800 μ m, and they are spaced 2 mm apart.

Comment R2-33: Suppl line 245 – why is the ECoG electrode array described here at all?

Response R2-33:

We fabricated a parylene C based μ ECoG electrode array to use it on the surface of the cortex while the depth recording is being performed, so that we can investigate propagation of the neural signals from the depth to the surface. (**Fig. 2d, supplementary Fig. 13 and supplementary Fig. 14**) We described the μ ECoG electrode in the Supplementary Information to provide a description of the fabrication detail, device design/dimensions.

To clarify the reason for the ECoG electrode array fabrication, we have updated the manuscript as follows.

Changes to the manuscript:

Added sentence, Supplementary Information, Page 7

We fabricated a μ ECoG electrode array to use during depth recordings to address the location of the barrel cortex before μ SEEG implantation. The μ ECoG electrode array was placed on the region where the barrel cortex is expected to be located and we performed recordings with whisker airpuff stimulation. After we confirmed that the μ ECoG electrode array captures activities from the whisker airpuff stimulation, the μ SEEG was implanted between the columns of the μ ECoG electrode array. Parylene C was chosen as the surface μ ECoG electrode material because of its superior conformability and hydrophobic surface which makes stable electrical and mechanical contact with the surface of the cortex³.

Comment R2-34: Stylet retraction – Suppl Fig 5 impressively shows to stability in implantation depth after stylet retraction. Suppl Fig 6 indicates that the probe sheath is plastically deformed which renders the probe mechanically stiff. Will the open lumen (the sheath and the respective etch holes to remove the sacrificial layer) between the 1st and 2nd PI layer represent a potential path of infection into deeper brain structures? In any case, the fact that the probe is plastically deformed needs to be mentioned in the main part of the manuscript.

Response:

We appreciate the reviewer's comment. As mentioned earlier, the geometry of our μ SEEG only returns from an eye-shape to a thin-film structure after stylet retraction, indicating that our μ SEEG is not plastically deformed by stylet insertion and retraction. Moreover, the histology result we have included in this paper (**supplementary Fig. 7** of the submitted version or **supplementary Fig. 10** of the revised version) indicates that there is no sign of potential infection due to stylet retraction. We will conduct further investigations to confirm our argument through additional chronic experiments with large animal brain models including histological investigations before using our electrode for clinical trials.

Reviewer: 3

In this manuscript, Lee et al. introduce a flexible micro-stereo-EEG (μ SEEG) electrode for possible human subject recording. They describe the fabrication process and use of electrode materials that permit miniaturization of the neural interface without compromising impedance. This is a very interesting and promising project that has significant potential for future applications as a combined human clinical and research tool. However, there are many areas to be clarified by the authors:

Response: We thank the reviewer for the positive comments and provide below responses to the reviewer's concerns.

Comment R3-1: The authors employ a fabrication process that employs a sacrificial Ti interlayer to allow creation of a channel between two polyimide layers. Figure SI2F demonstrates a SEM image that clearly shows the two PI layers are not merged to form a single block at the edges. What is the cause of this and how does it affect the yield and reproducibility of the devices?

Response R3-1:

Figure 1c shows that the sacrificial Ti interlayer's width and length are smaller than those of the 1st and 2nd polyimide layers. This design enables the bonding of the 2nd polyimide layer onto the 1st polyimide layer, except for the region where the sacrificial Ti interlayer is deposited. After the removal of the sacrificial Ti interlayer, there exists an unbonded area between the 1st and 2nd polyimide layers, where the sacrificial Ti interlayer was initially patterned. The tip of the electrode was sealed as a single block of PI layer, with help of interlocking hole arrays that strengthens the adhesion between the 1st and 2nd polyimide layer, where the Ti sacrificial layer was not patterned. Figure SI2F is provided to show cross-sectional detail of the tip with the stylet inside, as we wrote in the supplementary note: "The tip of the μ SEEG electrode was deliberately opened by a razor blade to observe cross-sectional details of the μ SEEG electrode with bio-materials remained on." However, we recognize that the Figure SI2F might give confusion that the tip is opened.

Changes to the manuscript:

To clarify this point, we added the following sentence to the caption of Figure SI2F.

Added sentence, Supplementary Information, Page 17

supplementary Fig. 2. Stylet insertion into the μ SEEG electrode. (a) Before insertion. (b) Stylet aligned to the sheath. (c) During insertion. (d) Stylet insertion all the way to the tip. (e) Scale comparison with conventional SEEG. (f) Cross-sectional SEM image of the μ SEEG. The tip of the μ SEEG electrode was deliberately opened by a razor blade to observe cross-sectional details of the μ SEEG electrode with bio-materials remained on.

Comment R3-2: It is important to establish a scalable fabrication process. Currently the process spans more than 1 mm (Figure SI2F). Ability to miniaturize would likely improve the device biocompatibility and signal quality. What is the smallest bonding width requiring for such processes and what dictates this limit?

Response R3-2:

We appreciate the reviewer's comment. Conventionally, what determines the width of the device are 1) number of channels and 2) lithography resolution. Meanwhile, the μ SEEG need more space for adhesive hole arrays (interlocking layer) and stylet insertion, which is the most dominant factor to determine the width. As you can see in the figure below (**Fig. R3-1**), the stylet was not tight to the sheath of the μ SEEG; the sheath still has hollow space after stylet insertion. Given that the μ SEEG adjusts its shape when the stylet is inserted because it is flexible (**Fig. R3-2**), we can expect the minimum width of the electrode can correspond to a circumference of the stylet. As shown in **Fig. R3-3**, when the inner sheath length is same with the circumference of the stylet ($314\ \mu\text{m}$), the total width of the μ SEEG after the stylet retraction can be calculated to be $544\ \mu\text{m}$. This number can be reduced further if we decrease the diameter of the stylet.

Fig. R3-1. Optical microscope (OM) image of the laminar electrode with a-100- μm -wide stylet inserted.

Fig. R3-2. Scanning electron microscope (SEM) of the μ SEEG (left) after stylet retraction and (right) with the stylet inserted (same with supplementary Fig. 6f).

Fig. R3-3. Illustration of the μ SEEG that has a tight interface with the stylet.

Comment R3-3: Regarding the pull tests performed, were all sections of the device tested, or just the tip? The geometry of the full device contains several right angles that may be more sensitive to disruption. In the quantitative comparison, the force required to disrupt the μ SEEG electrode was 4 times less than for the PMT electrode. Could the authors please comment on the practical implications of this decreased resistance to pulling forces?

Response R3-3:

We conducted a pull test specifically for the tip region. In order to eliminate potential weak points and prevent cracking, we designed the electrode with rounded turns rather than sharp turns. This design significantly enhances the electrode's resistance to shear force.

Compared to clinical electrodes, the μ SEEG is much thinner and more flexible, which can make it more difficult to handle. However, in all our acute and semichronic experiments reported in this manuscript and acute experiments in pigs (not shown here), we were able to retract the μ SEEG with relative ease and without deformation or break of the electrode. Our plan is to conduct pre-clinical animal experiments to develop implantation and explanation procedures for these electrodes by experienced neurosurgeons as part of our regulatory application for use of these electrodes in humans, and further strengthen the electrodes by increasing the polyimide thickness if necessary.

Comment R3-4: (line 169) Also please check the units – why does 1 MPa correspond to 16 mN and 14 kPa correspond to 48 mN? Figure SI4 does not contain any tensile strength data. (Keundong)

Response 3-4:

We apologize for the confusion. The applied stress on a material can be determined by dividing the applied force by its area. **Fig. R3-4** shows illustrations of the area of stress of the electrodes. For the polyimide μ SEEG, which has a width of 1 mm and a thickness of 15 μm , the applied stress can be calculated as $16 \times 10^{-3} \text{ N} / 15 \times 10^{-9} \text{ m}^2 = 1.06 \times 10^6 \text{ N/m}^2$, which corresponds to 1 MPa. This is the stress at which the μ SEEG was damaged. Similarly, the parylene C μ SEEG which has a width of 1 mm and a thickness of 4 μm , the applied force when the films are broken was 24 mN in the area of $4 \times 10^{-8} \text{ m}^2$, which the applied stress corresponds to 0.6 MPa.

For the clinical electrode, which has a hollowed structure, its thickness (outer diameter - inner diameter) is 100 μm , and its diameter is 1.2 mm, giving a circumference of 3.52 mm. The area of the stress can be calculated as $3.5 \times 10^{-7} \text{ m}^2$. Given that the force applied was 48 mN, the applied stress can be calculated as $48 \times 10^{-3} \text{ N} / 3.5 \times 10^{-7} \text{ m}^2 = 1.4 \times 10^4 \text{ Pa}$, which is 14 kPa.

Fig. R3-4. Illustration of the area of the stress (cross-sectional illustrations for parylene C μ SEEG, polyimide μ SEEG, and clinical electrode).

Changes to the manuscript:

To provide better understanding, we updated supplementary Fig. 4 as follows.

Note that supplementary Fig. 4 has changed to supplementary Fig. 5 due to the newly added figures in the revised manuscript.

Updated figure, supplementary Fig. 5d, Page 20

Supplementary Fig. 5. Pull measurement setup and result. (a) Measurement setup. (b) Our μ SEEG and (c) 1.2 mm diameter PMT depth electrode integrated on the measurement setup. (d) illustrations of area of stress (cross-sectional illustrations of the tested electrodes)

Updated text of Supplementary Information, Page 8

We investigated tensile stress of μ SEEG using pull measurement to examine mechanical stability of our electrode and compare it with clinical depth electrode (PMT electrode). The μ SEEG was integrated on the pressure sensor system (voice coil-powered linear actuator system with internally integrated force sensor; V-275 PIMag Voice Coil Linear Actuator). A customized 3D printed sample mount was attached onto the actuator (**supplementary Fig.**

45a) and a tip of the μ SEEG was placed on the sample mount followed by strong fixation using epoxy. Rest of the electrode was attached on a main body of the system and fixed by epoxy as well (**supplementary Fig. 45b**). We applied force to the actuator to exam tensile strength upon the μ SEEG-electrodes break (indicated by red arrow) and obtained that the critical tensile strength of the μ SEEG is ~~1 Mpa (16 mN) – 2.5 MPa (37 mN)~~. The critical strength of parylene C μ SEEG and polyimide μ SEEG are 0.6Mpa (24 mN with an area of the stress of $4e-8$ m²) and 1 – 2.5 Mpa (16 – 37 mN with an area of the stress of $1.5e-8$ m²), respectively. Meanwhile, the PMT electrode that was anchored on two polyurethane tube regions around a Pt contact was damaged when 14 kPa (48 mN with an area of the stress of $3.5e^{-7}$ m²) was applied to the PMT electrode (**supplementary Fig. 45c**). The illustration of the area of the stress of electrodes is illustrated in **supplementary Fig. 5d**.

Comment R3-5: (line 223) "We transitioned to all polyimide PtNR μ SEEG electrodes after we observed that parylene C PEDOT:PSS μ SEEG develop cracks in the parylene C layers and in the PEDOT:PSS Parylene C is a chemically vapor deposited polymer, and therefore as a soft polymer it does not "crack". Incompatible thermal cycles or chemistry during the fabrication process as well as poor deposition control of the material are likely the main source of these defects. This material has been approved and extensively used clinically for long-term chronic implants and components. The same can be said in regards to PEDOT:PSS. The authors employed a simplified electro-polymerization process that allows deposition of free PEDOT:PSS onto a conducting surface. From our understanding, no chemistry or crosslinkers were employed or investigated to improve this interface. Thus, the issues observed (though no data is presented in the manuscript) are potentially related to the authors' fabrication process rather than intrinsic properties of the material itself. Indeed, there are several publications that systemically and quantitatively evaluate this topic (e.g. Oldroyd et al Adv Func Mat 2022) and show, using appropriate controls and processes, that the yield of PEDOT:PSS layers surpasses metal interfaces in stability.

Response R3-5:

We appreciate the reviewer's insightful comment. We agree with the reviewer that parylene C is a soft polymer that does not crack intrinsically. We mentioned "crack" since we observed line shaped damages on the parylene C layer after inserting the stylet which most likely occurred due to mechanical damage and stress applied to the parylene C layer by the stylet, when it was not properly inserted. **Fig. R3-5** shows that the crack from the parylene C layers propagates along the region where the stylet is inserted. In contrast, it is clearly shown that PEDOT:PSS was not cracked when the substrate material changes to polyimide layers. We also performed FEM analysis to simulate how much of stress is applied and the layers are deformed when the stylet is inserted. We included the simulation result in the updated manuscript as below.

Fig. R3-5. OM images of PEDOT:PSS μ SEEG electrodes. Parylene C electrodes (a) before and (b) after stylet insertion and (c) its zoomed-in image. Polyimide electrodes (d) before and (e) after stylet insertion and (f) its zoomed-in image.

Changes to the manuscript:

To clarify the reason for crack in parylene C PEDOT:PSS μ SEEG electrodes, we newly added the above figure in the Supplementary Information and updated sentences in the main text as follows.

Added figures, supplementary Fig. 9, Page 25

Supplementary Fig. 9. FEM analysis results showing (a) external strength and (b) deformation introduced by stylet insertion. OM images of the PEDOT:PSS/polyimide (c – e) and (f – h) PEDOT:PSS/parylene C electrodes before and after stylet insertion.

Added sentences, Supplementary information, Pages 8-9

Mechanical stress introduced by stylet insertion was investigated using finite element method (FEM) analysis (**supplementary Fig. 9a** and **9b**). The analysis indicates that the maximum strength applied to the films during the stylet insertion (0.11 Mpa) are lower than the critical strength of the polyimide (1 – 2.5 Mpa) and parylene C (0.6 Mpa), as shown in **supplementary Fig. 9a**. Deformation of the films are lower than 4 μm, which can be considered as negligible from the scale of the electrodes, as shown in **supplementary Fig. 9b**. In theory, according to the simulation, the electrode should not sustain damage during stylet insertion. However, in practical scenarios, there are additional variables that can introduce stress to the electrode, such as misalignment of the electrodes in DI water (twisted or curved) and the angle at which the stylet is inserted, which may not be perfectly horizontal to the surface. While the polyimide layers are resilient and can withstand external stress arising from real-world conditions (**supplementary Fig 9c – e**), the parylene C layers appear to be approaching their critical limits (**supplementary Fig 9f – h**).

Updated and added sentences, Main text (Results), Page 7

We transitioned to all polyimide **PtNR** μ SEEG electrodes after we observed that parylene C **PEDOT:PSS** μ SEEG develop cracks in the parylene C layers and in the PEDOT:PSS layers after stylet insertion whereas polyimide μ SEEG did not suffer from any cracks. The crack was caused by mechanical damage to the parylene C layers applied by the stylet during insertion, which then propagated to the PEDOT:PSS layer (**supplementary Fig. 9**). Additionally, PtNRs contacts did not suffer any delamination from the μ SEEG whereas PEDOT:PSS suffered from delamination after stylet insertion in a substantial subset of electrodes, therefore reducing product yield.

Comment R3-6: The electrophysiology traces in several of the figures (Figure 2, 3 and several of the supplementary figures) are very difficult to interpret due to small size, incorporation of too many traces, and overlaid green highlighting. Clarity of data presentation could be overall improved.

Response R3-6 and changes to the manuscript:

We thank the reviewer for their useful comment. We have attempted to expand traces and improve the clarity of the presentation with the latest figures.

Comment R3-7: The authors point out successful recordings in 3/9 rat whisker barrel experiments. Greater discussion of why only 1/3 of recordings were successful would be useful.

Response R3-7:

We appreciate the reviewer's feedback. During the acute experiment, implanting the μ SEEG electrode in the rat barrel cortex was relatively straightforward as we had the flexibility to create a larger craniotomy and position the μ ECoG arrays to locate the barrel cortex before electrode implantation. However, for chronic implantations, additional preparations are necessary. Firstly, the headstage must be securely fixed to the rat's skull prior to the implantation. Secondly, it is crucial to keep the size of the craniotomy as small as possible to minimize the risk of infections. These requirements increase the complexity of implanting the μ SEEG electrode into the rat barrel cortex and resulted in successful detection of strong whisker stimulation-evoked activities in only 3 out of 9 rats.

To clarify the reason of 1/3 of recordings were successful, we added sentences in the Supplementary Information as follows.

Changes to the manuscript:

Added sentences, Supplementary Information, Page 11

4.3. Chronic rat recordings: implantation of μ SEEG electrode, surgical procedures, and sensory stimulation

The short 32 channel μ SEEG electrode for chronic rat recordings consists of 32 channels of microcontacts that are spread along the 2-mm-scale μ SEEG electrode, which are connected to the metal pads for interfacing with PCB (**supplementary Fig. 22a**). The metal pads and PCB are bonded by Ag epoxy. The back side of the PCB has electrical and mechanical contact with zero-insertion-force (ZIF) connector. The ZIF connector was activated with a lever to close the contact after flat flexible cable (FFC) insertion into the ZIF connector to make secure electrical and mechanical contact.

For implantation of the chronic μ SEEG electrode, first, a C-shape mount was fixed onto the skull near the craniotomy using screws and it was then bonded with the headstage using a super glue. Then, the chronic μ SEEG electrode with PCB and ZIF with FFC inserted was temporarily attached on the glass slide and the glass slide was connected to the micromanipulator (**supplementary Fig. 22b**). During the implantation, position of the chronic μ SEEG electrode was precisely controlled by micromanipulator. After the implantation, FFC was unplugged from the ZIF and the chronic μ SEEG electrode with PCB and ZIF are securely protected by the headstage and fixed onto the skull due to the screw-fixed mount. The craniotomy was sealed by ultraviolet curable medical glue (Tetric EvoFlow). Stimuli to induce neural activity included whisker, paw, and trunk mechanical stimulation including air puffs delivered >10 times/trials and the recording data from each trial were averaged.

During the acute experiment, implanting the μ SEEG electrode in the rat barrel cortex was relatively straightforward as we had the flexibility to create a larger craniotomy and position the μ ECoG arrays to locate the barrel cortex before electrode implantation. However, for chronic implantations, additional preparations are necessary. Firstly, the headstage must be securely fixed to the rat's skull prior to the implantation. Secondly, it is crucial to keep the size of the craniotomy as small as possible to minimize the risk of infections. These requirements increase the complexity of implanting the μ SEEG electrode into the rat barrel cortex and resulted in successful detection of strong whisker stimulation-evoked activities in only 3 out of 9 rats.

Response R3-8: The animal experiments demonstrate patterns consistent with recording across cortical layers, but in no case was the quality of recording compared to any conventional device. The lack of this comparison makes it difficult to evaluate whether there are any limitations of the data generated by these devices. As the authors mention, they had to remove a strong common-mode signal on all contacts in order to derive CSD dynamics. Side by side comparison of the μ SEEG with a conventional silicon probe in animal experiments would delineate this issue and determine the extent to which cross-talk is present across the channels. The utility of having >100 channels is severely limited if the channels are contaminated by cross-talk.

Response R3-8:

We appreciate the critical comment from the reviewer. While a direct side-by-side comparison between μ SEEG and a conventional silicon probe in animal experiments would be ideal to address this issue, it presents certain challenges. Firstly, the implantation locations for both electrodes may be too distant to record the exact same neuronal activities, complicating data comparison. Secondly, the footprint of the implantation will differ as the silicon probe is sharp and rigid compared to the μ SEEG, potentially leading to variations in data quality and making the analysis more complex.

For our data recording, it is possible that we measured the same neuronal activities, given the narrow spacing of the electrodes (60 μ m). Additionally, we observed that the location of the reference contact also affects the amount of common signal captured. Moving forward, we will continue to optimize our recording setup to improve the recording quality.

Changes to the manuscript:

To further discuss the cross-talk issue of our electrode, we newly added discussions in the main text as follows.

Updated and added sentences, Main text (Results), Page 17

One potential limitation of these designs is cross-talk amongst the channels. While we have not definitively quantified cross-talk in the recordings, we observed a strong common-mode signal on all contacts that we subtracted in order to delineate CSD dynamics. One of possible reasons for the common-mode signal would be recording the same neural activities, given the narrow spacing of the electrodes (60 \$\mu\text{m}\$ ). Additionally, we observed that the location of the reference contact also affects the amount of common signal captured. If inter-channel cross-talk is a substantial issue, future designs could involve distributing the metal traces in separate polymer layers. Another limitation concerns connectorization. Current connectors do not match typical clinical standards. Improving the back-end of the devices is an area of active development. Further, the current design includes contacts facing only one direction along the electrode length. Future designs can involve developing multiply directional contact sampling.

Comment R3-9: The authors also demonstrate that their technology allows contacts within 7mm at the tip of a long electrode. One of the fundamental properties of SEEG is the ability to have contacts along the entire length of the electrode, from the superficial cortex to the deep gyri to deeper brain targets of gyri. The authors demonstrate the ability to advance an electrode with tip contacts and sequential record from targets. Is this technology compatible with contacts along the length of the electrode, particularly with concerns over common-mode signal cross talk.

Response R3-9:

We appreciate the feedback from the reviewer. Achieving contact distribution along the entire length of the electrode is a feasible task through modifications to the layout file using software such as AutoCAD or L-edit. In our electrodes prepared for a pre-clinical and clinical work (See Fig. R2-5 above), we have uniform contact distribution of both micro and macro-contacts throughout the array.

Changes to the manuscript:

We added the following sentence in the manuscript, Discussion Section, page 18:

Stimulation and recording contacts can be distributed uniformly or in clusters across the length of the \$\mu\$ SEEG.

Comment R3-10: The ability to record single units with SEEG is currently a research advantage, without as yet documented clinical benefit. It is important to point out that while there are many potential benefits of being able to clinically record LFPs and single units, this merged technology is currently research and not clinical in the human domain of epilepsy SEEG, the only currently approved use of SEEG. Additionally, the goal of SEEG is not to create a 3-D grid, as this is impossible, and even more so with smaller unidirectional contacts as described here. The value of SEEG in treating human disease is only as good as the anatomo-electro-clinical hypothesis being investigated. It is important that newly developed uSEEG electrodes do not become a vehicle for scientific “fishing expeditions” because the technology allows access of single unit recordings to new areas of the brain.

Response R3-10:

The Reviewer makes several excellent points regarding both limited current usefulness of single units in the clinical domain, particularly with epilepsy investigations to identify seizure onset zones in the course of clinical care. Further, we appreciate the Reviewers' point of that using these devices without clear hypotheses related to the clinical questions at hand could

be challenging. However, considering the widespread use of both FDA-approved Benke-Fried and DIXI MicroDEEP electrodes (which are SEEG electrodes but with additional micro-contacts for single-unit activity), patient consent and participation in undergoing these recordings are adding to the growing evidence in the use of single units in investigating the contributions of cell types in epilepsy³⁻⁸. Indeed, a recent publication from our collaborators have demonstrated that high resolution microelectrode information (similar to the scale of the electrode devices presented here) may be useful for localizing epileptiform activity to the scale of cortical layers with possible improvements to identifying the seizure onset zone⁹.

In addition, a major benefit of the PEDOT:PSS and PtNR electrode contacts is that they are low impedance even as microelectrodes, enabling true 'broad band' recordings from single unit activity to local field potential (LFP), the latter which are used to identify epileptiform activity in the clinical domain. Current clinical electrodes are often metal contacts which can limited the type of data they can collect as well as the shape and size of the manufactured electrodes. In contrast, the μ SEEG devices offer customizable flexibility in the design and layout with a wide amount of possible information gathered in the neural recordings. Therefore, we are confident these electrodes and the substrate provides a massive amount of information including clinically relevant information.

Finally, single neuron activity is used to guide brain structure targeting such as the subthalamic nucleus in deep brain stimulation surgery which could be a second example where a device at this scale (which is more narrow than the DBS electrode) could be useful in improving targeting resolution¹⁰.

Comment R3-11: A critical component of SEEG is the ability to target specific structures in the dorsoventral axis. What is the accuracy and precision of targeting with these μ SEEG devices? Given that a stylet is used to insert the device, and the device itself is quite flexible and light, does the device shift along the insertion tract when the stylet is removed? The targeting capacity of the devices should be delineated.

Response R3-11:

The Reviewer asks excellent questions regarding the precision and targeting of key brain structures with these μ SEEG devices. The plan for implanting these devices though, will be guided by the use of an obturator (often a thin steel rod) which can form an initial track to the target. After the obturator is removed, the electrode can then be placed, following that initial track. This is the procedure used for most clinical SEEG electrodes which, without the obturator, may also be deflected on the path to the target (which we have observed in past cases on the clinical monitoring side). However, we agree with the reviewer that we should test further whether or how much this electrode will shift as it goes to target in large animal models, including with stylet removal. This could be a part of future studies of our pre-clinical work.

Comment R3-12: Were units detected across multiple channels?

Response R3-12:

At the depth where we detected the most number of units (depth 2), we found 5 multi-unit activity (MUA) clusters and 31 single unit clusters of waveforms across 128 channels. We did detect units across multiple channels in approximately a third of the detected clusters.

Comment R3-13: It takes some searching to figure out the actual number of experiments done. For instance, the histology data between a conventional SEEG electrode tip and the μ SEEG electrode appears to be based on 1 rat using one hemisphere for each electrode, at one time point, with limited GFAP and neuronal staining. Based on this, the authors concluded that

the new μ SEEG technology “induced less apparent tissue damage than clinical SEEG electrodes”. In the same data section (lines 194-197), the authors contend that an apparently acute (as there were no chronic NHP experiments) implant into a NHP did not result in imaging changes to the electrode as demonstrating the “stability of the μ SEEG electrode in tissue (supplementary Fig. 8, N = 3)”. This conclusion is an overreach of the the data.

Response R3-13:

We appreciated the reviewer’s comment. We agree with the reviewer’s comment on lack of chronic recording data from NHPs. However, to support our argument as much as we can, we include data from a new experiment with 4 rats per group, and 1-4 sections analyzed per animal, after clinical, and μ SEEG electrode insertion for 14 days (**Fig. R3-6**). In these data, we demonstrate modest improvement over the clinical electrode in the amount of GFAP, and a trend towards an increase in the proximity of neurons to the lesion site. We will further work on chronic implantation using a large animal model (pig’s brain) to prove the stability of the μ SEEG electrode in a large animal model.

Fig. R3-6. (a) average GFAP area, normalized to each animal (2-way ANOVA, $F(1,72) = 8.749$, $P=0.0042$; $N=4$ /electrode, 1-4 sections per animal). (b) Normalized NeuN+ density (2-way ANOVA, $F(1, 72) = 3.290$, $P=0.0739$, $N=4$ /electrode, 1-4 sections per animal).

Changes to the Manuscript:

To provide additional information, we updated the main text and **supplementary Fig. 7** as shown below.

Note that **supplementary Fig. 7** has changed to **supplementary Fig. 10** due to the newly added figures in the revised manuscript.

Updated figure, supplementary Fig. 10, Page 26

Supplementary Fig. 10. GFAP and NeuN reactivity following two weeks of electrode implantation. Wide-field view of horizontal cortical section following (a) clinical or (b) μ SEEG electrode implantation. NeuN is pseudocolored green, GFAP magenta, and DAPI blue. (c) and (d) are magnifications of the boxed areas in (a) and (b) to show NeuN reactivity adjacent to electrode placement. Representative NeuN-positive cells are indicated by arrows. (e) Average number of NeuN+ cells per $100 \mu\text{m}^2$ sampled randomly at 0-100, 100-200, and 200-300 μm from the electrode. (f) Average GFAP intensity of $100 \mu\text{m}^2$ boxes sampled randomly at 0-100, 100-200, and 200-300 μm from the electrode. GFAP area, normalized to each animal (2-way ANOVA, $F(1,72) = 8.749$, $P=0.0042$; $N=4/\text{electrode}$, 1-4 sections per animal). (f) Normalized NeuN+ density (2-way ANOVA, $F(1, 72) = 3.290$, $P=0.0739$, $N=4/\text{electrode}$, 1-4 sections per animal). Data are plotted as mean \pm SEM, **** $P < 0.0001$, ** $P < 0.01$ by 2-way ANOVA followed by Sidak's multiple comparisons test. Scale bars are $100 \mu\text{m}$.

Updated sentences, Supplementary Information, Page 11

At the termination of the chronic recordings (by four weeks after implant), brains were removed and briefly (2 hrs) fixed in 4% PFA. The cortices were removed and flattened between glass slides in PBS, and then the flattened cortex further fixed in 4% PFA overnight, and then allowed to sink in 30% sucrose. For measuring implantation scar, rats were implanted with a dummy clinical electrode or μ SEEG and allowed to recover for two weeks ($n = 1 / \text{electrode}$). Animals were perfused with 4% PFA and then sunk in 30% sucrose. Whole brains or flattened cortices were sectioned on a cryostat in the horizontal plane from at $40 \mu\text{m}$. Sections were washed with PBS and stained overnight with one or more of the following: chicken anti-NeuN (1:500, Sigma ABN91), Cy3-conjugated mouse anti-GFAP (1:1000, Sigma C9205), rabbit anti-VGLUT2 (1:500, Abcam ab216463). After wash, slices were incubated in secondaries of goat anti-chicken 488 (Sigma SAB4600039), goat anti-chicken 647 (Sigma SAB2600184), or goat anti-rabbit 488 (ThermoFisher A-11008) at 1:300 for 1 hour, rinsed, and mounted with ProLong with DAPI. Slides were imaged on a Nikon Eclipse Ti2-E with a DS-Q12 CMOS camera at the UCSD Nikon Imaging Core Images were analyzed using ImageJ (**supplementary Figs. 10a**

to d). In our observations from ~~one four~~ chronic (2 weeks) placement, ~~immunohistochemistry revealed NeuN-positive cells in close (< 20 µm) proximity to the µSEEG electrode and lower GFAP intensity and spread~~ we observed no significant difference in the number of Neun-positive cells surrounding the lesion between the µSEEG and the clinical grid, but there was a small non-significant improvement with the µSEEG electrode (2-way ANOVA, $F(1, 72) = 3.290$, $P = 0.0739$) (**supplementary Figs. 10e and 10f**).

Updated sentences, Main text (Results), Page 5

Finally, to test the amount of tissue damage caused by these devices, we implanted rats with one chronic µSEEG electrode with 1.89 mm recording length on one hemisphere and a clinical electrode on the other hemisphere for 14 days ($N = 4$ 4 electrode). Insertion of the µSEEG electrode resulted in decreased astrocyte scarring, as measured by significantly lower GFAP positive area intensity as compared to the clinical electrode as shown in (**supplementary Fig. 10**) (2-way ANOVA, $F(1,72) = 8.749$, $P=0.0042$). ~~Within 100 µm from the electrode, for example, GFAP intensity in five randomly placed 100 µm² boxes was significantly lower for the µSEEG electrode (1842 ± 53 a.u.) compared to the clinical electrode (3840 ± 339 a.u.; two-way ANOVA, $F(1,24) = 85.93$; $P < 0.0001$; Sidak's multiple comparisons posthoc, $p < 0.001$).~~ We observed no significant difference in the number of Neun-positive cells surrounding the lesion between the µSEEG and the clinical grid, but there was a small non-significant improvement with the µSEEG electrode (2-way ANOVA, $F(1, 72) = 3.290$, $P = 0.0739$). ~~In addition, significantly fewer neurons were observed between 0-100 µm from the electrode for the clinical electrode compared to the µSEEG (2.4 ± 1.2 vs. 19 ± 2.3 cells; 2-way ANOVA, $F(1,24) = 23.26$, $P < 0.001$; Sidak's multiple comparisons posthoc, $p < 0.0001$).~~

1. Tchoe, Y. et al. Human brain mapping with multithousand-channel PtNRGrids resolves spatiotemporal dynamics. *Science translational medicine* **14**, eabj1441 (2022).
2. Buzsáki, G., Anastassiou, C.A. & Koch, C. The origin of extracellular fields and currents—EEG, ECoG, LFP and spikes. *Nature reviews neuroscience* **13**, 407-420 (2012).
3. Carlson, A.A., Rutishauser, U. & Mamelak, A.N. Safety and utility of hybrid depth electrodes for seizure localization and single-unit neuronal recording. *Stereotactic and functional neurosurgery* **96**, 311-319 (2018).
4. Cash, S.S. & Hochberg, L.R. The emergence of single neurons in clinical neurology. *Neuron* **86**, 79-91 (2015).
5. Chari, A., Thornton, R.C., Tisdall, M.M. & Scott, R.C. Microelectrode recordings in human epilepsy: a case for clinical translation. *Brain communications* **2**, fcaa082 (2020).
6. Mukamel, R. & Fried, I. Human intracranial recordings and cognitive neuroscience. *Annual review of psychology* **63**, 511-537 (2012).
7. Reed, C.M. et al. Extent of single-neuron activity modulation by hippocampal interictal discharges predicts declarative memory disruption in humans. *Journal of Neuroscience* **40**, 682-693 (2020).
8. Truccolo, W. et al. Single-neuron dynamics in human focal epilepsy. *Nature neuroscience* **14**, 635-641 (2011).
9. Fabo, D. et al. The role of superficial and deep layers in the generation of high frequency oscillations and interictal epileptiform discharges in the human cortex. *Scientific Reports* **13**, 9620 (2023).
10. Amirnovin, R., Williams, Z.M., Cosgrove, G.R. & Eskandar, E.N. Experience with microelectrode guided subthalamic nucleus deep brain stimulation. *Operative Neurosurgery* **58**, ONS-96-ONS-102 (2006).

REVIEWERS' COMMENTS

Reviewer #1 (Remarks to the Author):

The authors have gone through significant lengths to address concerns, especially with accelerated lifetime testing and the additional figures regarding the parylene C devices. There are two areas raised previously that, once addressed, would have the manuscript in a state ready for publication. My remaining concerns are both related to the relative recording quality seen before/after removal of the stylet and if there is significant decline in recording quality with removal of the stylet due to 'deflation' of the device into the larger cavity created by the stylet. It is clear from what is already shown that the device does not break with stylet removal.

1) Single-unit recording (R1-2)

The work here encourages comparison to existing clinical sEEG and depth electrodes, which are used for both intraoperative and sub-chronic implantation. The statements in the abstract (pg 1 lines 37-39)

"This thin, stylet-guided depth electrode is capable of recording local field potentials and single unit neuronal activity (action potentials), validated across species" and also discussion (see pg 15 line 36) are slightly misleading in that the single-unit recording capabilities have only been demonstrated during acute recording with stylet in place. It is understandable that no single-unit activity was able to be recovered from recordings from rodents and humans as the authors have described in their response, but the fact that this device has not yet yielded chronic single-unit recordings nor recordings after the stylet was removed should be made explicit when the single-unit capabilities of the device are discussed in order to be suitable for publication.

Recording with the stylet in place is a significant difference as the device is designed to 'deflate' after stylet removal (See R2-16) and could find the contacts, located in the center of the device where the largest change in device thickness before/after stylet removal, no longer in close approximation to the neural tissues. Supplementary figure 2H, Supplementary figure 10B, and the reported device thickness of 15 microns seem to suggest that the device may indeed be deflating into a larger cavity created by the presence of the stylet.

2) Common-mode signal (R1-3)

As asked previously, is this found in all datasets? Does this change with presence of stylet? This is related to the above concerns about single-unit recording with presence of stylet vs. after removal. It appears that at least some recordings, such as the single-unit recordings in large animals, were completed with stylet in place - did these recordings have large differences in common-mode signal compared to those with stylet removed?

Reviewer #2 (Remarks to the Author):

The authors handled all my comments in a highly satisfactory manner.

The authors mention that lines widths down to 1.2µm are feasible; have you ever thought about the line resistance compared to the electrode impedance using lines of this width? No need to respond, but definitely an issue to think about.

I only have to reply to one of the responses – nothing serious. Further, I have seen a couple of typos and wrong uses of units which might already be handled by the journal.

Reply to responses

- Response R2-30 – it seems doable to remove the Ti-based sacrificial layer as demonstrated. What I meant in my comment is not the etching of the Ti layer inside the etch openings – this should not be an issue at all. It is rather etching Ti in-between the 2 PI layers with a certain limitation in diffusion as the gap between both PI layers is just 60 nm. The transport of fresh buffered oxide etch and removal of the Ti-containing etch solution is limited inside this gap.

Minor typos

- Page 3, line 11: It should read microelectromechanical system (MEMS) devices.
- Page 3, line 15: remove "of" before the brackets
- Page 4, line 28: ... At the very tip "of" the electrode, ... - add "off"

- Page 5, line 12: during implantation “of the” μ SEEG
- Page 7, line 1: ... the micro-hole *array* that interlocks OR the micro-hole arrays that *interlock*
- Mechanical stress ins given in MPa not Mpa – see among other supplementary Information page 8: “The critical strength of parylene C μ SEEG and polyimide μ SEEG are 0.6Mpa (24 mN with an area of the stress of $4e-8$ m²) and 1 – 2.5 Mpa (16 – 37 mN with an area of the stress of $1.5e-8$ m²), respectively.”

Response Letter

Author Remark

We express our sincere gratitude to the reviewers for their positive evaluation and valuable feedback on our manuscript. In response to the reviewers' comments, we have carefully revised our manuscript to address their suggestions and concerns. In the following section, we respond to the reviewers' comments (displayed in red), addressing each point individually. Our responses are marked in black font, and the revised sections from the updated manuscript are copied and pasted in blue. The revisions are also appropriately highlighted within the updated manuscript.

Response to the Reviewer's comments

Reviewer: 1

The authors have gone through significant lengths to address concerns, especially with accelerated lifetime testing and the additional figures regarding the parylene C devices. There are two areas raised previously that, once addressed, would have the manuscript in a state ready for publication. My remaining concerns are both related to the relative recording quality seen before/after removal of the stylet and if there is significant decline in recording quality with removal of the stylet due to 'deflation' of the device into the larger cavity created by the stylet. It is clear from what is already shown that the device does not break with stylet removal.

Response:

We thank the reviewer for their positive summary of the work and also thoughtful comments on remaining concerns.

Minor concerns

R1-1) Comment on R1-2) The work here encourages comparison to existing clinical sEEG and depth electrodes, which are used for both intraoperative and sub-chronic implantation. The statements in the abstract (pg 1 lines 37-39)

"This thin, stylet-guided depth electrode is capable of recording local field potentials and single unit neuronal activity (action potentials), validated across species" and also discussion (see pg 15 line 36) are slightly misleading in that the single-unit recording capabilities have only been demonstrated during acute recording with stylet in place. It is understandable that no single-unit activity was able to be recovered from recordings from rodents and humans as the authors have described in their response, but the fact that this device has not yet yielded chronic single-unit recordings nor recordings after the stylet was removed should be made explicit when the single-unit capabilities of the device are discussed in order to be suitable for publication.

Recording with the stylet in place is a significant difference as the device is designed to 'deflate' after stylet removal (See R2-16) and could find the contacts, located in the center of the device where the largest change in device thickness before/after stylet removal, no longer in close approximation to the neural tissues. Supplementary figure 2H, Supplementary figure 10B, and the reported device thickness of 15 microns seem to suggest that the device may indeed be deflating into a larger cavity created by the presence of the stylet.

Response R1-1:

We thank the reviewer for their excellent feedback. We agree there could be confounds in capturing single unit activity following stylet removal as the device will deflate in the process. To obtain better single unit recording capabilities, we are currently working on reducing the stylet diameter while maintaining the stiffness of the stylet as much as it can by changing the material of the stylet from stainless steel to tungsten. Also, we will test on single unit recording capabilities by modifying location of the microcontact which is currently patterned at the center

of the electrode to the edge of the electrode so that the relative location of the microcontact in the brain is not changed much when the stylet is retracted as illustrated in **Fig. R1-1**.

Fig. R1-1. Illustration of the position of the microcontact before/after stylet retraction depending on 1) stylet diameter, and 2) location.

Changes to the manuscript:

To clarify and address the reviewer's point, we updated the manuscripts as follows.

Added discussion, Main text (Discussion), pages 16

One potential limitation of these designs is cross-talk amongst the channels. While we have not definitively quantified cross-talk in the recordings, we observed a strong common-mode signal on all contacts that we subtracted in order to delineate CSD dynamics. One of possible reasons for the common-mode signal would be recording the same neural activities, given the narrow spacing of the electrodes (60 μm). Additionally, we observed that the location of the reference contact also affects the amount of common signal captured. If inter-channel cross-talk is a substantial issue, future designs could involve distributing the metal traces in separate polymer layers. Another limitation concerns connectorization. Current connectors do not match typical clinical standards. Improving the back-end of the devices is an area of active development. Further, the current design includes contacts facing only one direction along the electrode length. Future designs can involve developing multiply directional contact sampling. Finally, additional optimizations regarding the stylet diameter and the placement of the microcontacts could enhance the capabilities of single-unit recording even after the stylet is retracted. This would prevent the electrode from deflating, ensuring that the microcontacts remain in close proximity to the neural tissues. This can be achieved either by reducing the diameter of the stylet or by positioning the microcontacts along the edge of the electrode.

R1-2) Comment on R1-3) As asked previously, is this found in all datasets? Does this change with presence of stylet? This is related to the above concerns about single-unit recording with presence of stylet vs. after removal. It appears that at least some recordings, such as the

single-unit recordings in large animals, were completed with stylet in place - did these recordings have large differences in common-mode signal compared to those with stylet removed?

Response R1-2:

We appreciate the reviewer's comment and bringing up an excellent point.

Unfortunately, we did not investigate common mode signal issues before and after stylet removal from the same brain tissue and the same setting. However, for the human, pig, and rat data obtained after stylet removal, common median subtraction helped resolve the signal and allowed us to perform CSD/MUA analysis that resembled previous publications with other electrodes. Of course, common mode effects can result from complex origins which include metal lead spacing and stylet retraction, among others. For clinical applications, the electrode will be used after stylet retraction. Therefore, because of our experiences as reported in this manuscript, the next version of this electrode is being developed to minimize common mode signal processing and increase single unit recording capabilities by adapting multilayer structures which 128 metal leads for recording patterned on two polyimide layers with wider spacing and patterning the electrode contacts on the edge of the electrode to minimize recording site movement due to the stylet retraction (as described above).

Reviewer: 2

The authors handled all my comments in a highly satisfactory manner.

The authors mention that lines widths down to 1.2 μ m are feasible; have you ever thought about the line resistance compared to the electrode impedance using lines of this width? No need to respond, but definitely an issue to think about.

I only have to reply to one of the responses – nothing serious. Further, I have seen a couple of typos and wrong uses of units which might already be handled by the journal.

Response:

We thank the reviewer for the thoughtful comments. The electrode impedance can be slightly higher if the metal lead width becomes narrower, but it is also possible to make the metal thickness thicker to recover the impedance. In all our electrodes, the metal traces are designed to provide a total resistance of less than 100ohms, which is usually less than 1% of the electrode impedance at 1kHz.

Minor concerns

R2-1) Comment on R2-30) It seems doable to remove the Ti-based sacrificial layer as demonstrated. What I meant in my comment is not the etching of the Ti layer inside the etch openings – this should not be an issue at all. It is rather etching Ti in-between the 2 PI layers with a certain limitation in diffusion as the gap between both PI layers is just 60 nm. The transport of fresh buffered oxide etch and removal of the Ti-containing etch solution is limited inside this gap.

Response R2-1:

We appreciate the reviewer's comment.

We understand the reviewer's comment on Ti sacrificial layer removal using BOE given that the thickness of the Ti layer is only 60 nm so the gap between 1st and 2nd PI layers would be only 60 nm, which seems to be too thin so the lateral etching would be limited. As we illustrate in **Fig. R2-1**, Ti net layer is 5 μ m wide and 5 μ m spaced, so BOE that leaks through each net opening in 1st PI layer only needs to etch 60 nm vertical, and 10 μ m wide Ti layer. When we had no net openings, we usually immersed the sample more than overnight, or a few days to

fully remove the sacrificial layer, as the reviewer mentioned – BOE etching is limited significantly.

Fig. R2-1. Illustration of the bottom components of the device including Ti net, 1st PI, Ti sacrificial, and 2nd PI layers when the device is immersed in BOE.

R2-2) Minor typos

- Page 3, line 11: It should read microelectromechanical system (MEMS) devices.
- Page 3, line 15: remove “of” before the brackets
- Page 4, line 28: ... At the very tip “of” the electrode, ... - add “off”
- Page 5, line 12: during implantation “of the” μSEEG
- Page 7, line 1: ... the micro-hole *array* that interlocks OR the micro-hole arrays that *interlock*
- Mechanical stress ins given in MPa not Mpa – see among other supplementary Information page 8: “The critical strength of parylene C μSEEG and polyimide μSEEG are 0.6Mpa (24 mN with an area of the stress of 4e-8 m2) and 1 – 2.5 Mpa (16 – 37 mN with an area of the stress of 1.5e-8m2), respectively.”

Response R2-2:

We appreciate the reviewer's comment. We edited minor typos that the reviewer pointed out. We greatly appreciate your correction.